# Monopole operators and bulk-boundary relation in holomorphic topological theories

Keyou Zeng$^\star$

Perimeter Institute for Theoretical Physics, Waterloo, Ontario, Canada N2L 2Y5

$\star$ kzeng@perimeterinstitute.ca

## Abstract

We study the holomorphic twist of $3d$ $\mathcal{N} = 2$ supersymmetric field theories, discuss the perturbative bulk local operators in general, and explicitly construct non perturbative bulk local operators for abelian gauge theories. Our construction is verified by matching the character of the algebra with the superconformal index. We test a conjectural relation between the derived center of boundary algebras and bulk algebras in various cases, including Landau-Ginzburg models with an arbitrary superpotential and some abelian gauge theories. In the latter cases, monopole operators appear in the derived center of a perturbative boundary algebra. We briefly discuss the higher structures in both boundary and bulk algebras.

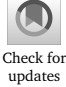

# 1 Introduction

Twisting supersymmetric field theories [1,2] has been a very powerful and successful tool for extracting mathematical structures from physical quantum field theories. Twisted theories, as they were originally discovered, involve fully topological twist, which renders the original field theories into topological field theories. Classical examples include A and B model topological string [2] in $2d$, Rozansky-Witten theory [3] in $3d$, Donaldson-Witten theory [1], Vafa-Witten theory [4] and Kapustin-Witten theory [5] in $4d$.

Fully topological twist depends on the choice of a supercharge $Q$ that transforms as a scalar under a "twisted" Lorentz group action. The existence of such a scalar requires the existence of a relatively large amount of supersymmetry. Actually, the twisting procedure can be modified to adapt to a generic nilpotent supercharge. Early examples include [6–8] and

this technique has been systematically developed in e.g. [9, 10] recently. After such a twist, the resulting theories become holomorphic in most space time directions. In even spacetime dimension $d = 2n$, the choice of the nilpotent supercharge specifies a complex structure on $\mathbb{R}^{2n} = \mathbb{C}^n$. All correlation functions of the twisted theory will depend holomorphically on $\mathbb{C}^n$. In odd spacetime dimension $d = 2n+1$, the choice of nilpotent supercharge specifies a splitting $\mathbb{R}^{2n+1} = \mathbb{C}^n \times \mathbb{R}$. Correlation functions of the twisted theory will depend holomorphically on $\mathbb{C}^n$ and be independent of $\mathbb{R}$.

In this paper, we study twisted 3d $\mathcal{N} = 2$ theories following [11, 12]. The purpose of this paper is twofold. First, we try to give a more detailed description of the bulk algebras of the twisted theories. The study of monopole operators will play an important role here. A monopole operator is defined by prescribed local singular behavior of fields. The strategy we employ for studying them relies on the state-operator correspondence, which has also been used in many previous works [13–15] that successfully identified their spectrum and various quantum numbers. In this paper, we combine state-operator correspondence with the method of geometric quantization to give a description of the full local operator algebras for abelian gauge theories. The algebra is decomposed into different sectors labeled by monopole (topological) charges and each sector represents a family of gauge invariant dressed monopole operators. As a justification of our construction, we compute the characters of the local operator algebras and our results agree nicely with existing literature on superconformal index of 3d $\mathcal{N} = 2$ theory [16–19].

Then, we examine a conjectural relation between the bulk and boundary local operator algebras [12]. This relation is actually a general feature of topological quantum field theories [20], which links boundary information with bulk information. The 2d TQFT version of this conjecture is the isomorphism between the bulk algebra and the Hochschild cohomology of boundary algebra. In 3d, this conjecture becomes the isomorphism between the bulk algebra and the derived center of the boundary chiral algebra, which is computed by self-Ext of the boundary vacuum module. This technique has also been used in [21, 22] to study Higgs and Coulomb branch operators of 3d $\mathcal{N} = 4$ theories.

This paper is organized as follows. In Section 2, we review the twisted 3d $\mathcal{N} = 2$ theories in BV formalism and in AKSZ formalism. In Section 3, we analyze the bulk local operators of the theories. A description of the perturbative algebras will be provided for the general situations. We describe non perturbative algebras for theories with abelian gauge group, by utilizing state operator correspondence. We also discuss characters of the local operator algebras and their relation with the 3d superconformal indices. In Section 4, we study local operator algebras and their characters in some explicit examples. The BRST cohomologies of the local operator algebras will be computed in the first few spin sectors. The implications of 3d mirror symmetry on the local operators will be discussed. In Section 5, we briefly review boundary conditions and the corresponding boundary chiral algebras. In Section 6, we compute the self-Ext of various boundary chiral algebras under different boundary conditions and prove that they agree with bulk algebra in some non-trivial cases. A failure case of this technique will also be provided. Finally, we touch on the algebraic structures on the self-Ext and their relations with the algebraic structures of bulk local operators.

## 1.1 Future directions

We outline several other motivations for this work and a partial list of related open issues.

**Three manifold invariants**  String Theory/M-Theory predicts the existence of the 6d superconformal field theories living on M5 branes. A twisted compactification of a 6d theory labeled by a simply-laced Lie algebra $\mathfrak{g}$ on a manifold $M$ of dimension $d$ defines a $(6-d)$ dimensional supersymmetric field theory $T[M, \mathfrak{g}]$. In the IR, the theories $T[M, \mathfrak{g}]$ only depend on part of

the geometry of $M$. Specifically, they all have a nilpotent supercharge whose cohomology is invariant under the diffeomorphisms of $M$.

For $d = 3$, theories $T[M^3, \mathfrak{g}]$ are a family of 3d $\mathcal{N} = 2$ supersymmetric field theories. For $\mathfrak{g} = \mathfrak{su}(2)$, theories $T[M^3, \mathfrak{su}(2)]$ are explicitly constructed in [23] based on a triangulation of the 3 manifold $M^3$ into tetrahedra, and a gluing of the "building block" theory $T_\Delta$ through the triangulation. This construction leads to the 3d $\mathcal{N} = 2$ abelian Chern-Simons matter theories, whose twisted versions are studied in this paper. Superconformal index of $T[M^3]$ is studied in [19] and gives us an invariant of the three manifold $M^3$. The bulk algebra $\mathrm{Obs}_{T[M^3]}$ then provides a "categorification" of the 3d index in [19] (In the sense that we replace a number by a graded vector space). The techniques in this paper can be applied to describe monopole operators in $T[M^3]$, and to give an explicit construction of the three manifold invariant $\mathrm{Obs}_{T[M^3]}$.

**Line operators**   Twisted 3d $\mathcal{N} = 2$ theories admit a large collection of line operators extending in the real direction. They come in two basic varieties: Wilson lines and vortex lines. Together, they generate a category $\mathcal{C}$ that is expected to be a chiral category [24] and holds a wealth of algebraic information about the theory. For example, in this guise the bulk boundary relation studied in this paper can be adapted to the statement that local operators are the same as (or can be defined as) self-interfaces of the trivial line defect. More explicitly, we have

$$\mathrm{Obs}_{\mathrm{loc}} = \mathrm{Hom}_{\mathcal{C}}(\mathbb{1}_{\mathcal{C}}, \mathbb{1}_{\mathcal{C}}). \tag{1.1}$$

For topological twist of 3d $\mathcal{N} = 4$ theories, line operators are systematically studied in [25]. It will be interesting to identify the category $\mathcal{C}$ in various $\mathcal{N} = 2$ setups.

Mirror symmetry always makes nontrivial predictions for mathematical objects extracted from mirror quantum field theories. For the category of line operators, mirror duality of two 3d $\mathcal{N} = 2$ theories $\mathcal{T}$ and $\mathcal{T}'$ then suggests an equivalence of categories $\mathcal{C}_{\mathcal{T}} \cong \mathcal{C}_{\mathcal{T}'}$. For 3d $\mathcal{N} = 4$ theories, such examples are explored in [25] and recently in [26].

**Integrable system**   We can consider the twisted 3d $\mathcal{N} = 2$ theories defined on a spacetime of the form $\Sigma \times \mathbb{R}$, where $\Sigma$ is a Riemann surface. We can also consider line defects passing through $\Sigma$ and this will lead to a punctured surface with some labels on the punctured points. The twisted theory will assign a Hilbert space $\mathcal{H}(\Sigma)$ to the surface after quantization [27]. Bringing a bulk operator to the surface $\Sigma$ defines an action of the operator on the Hilbert space $\mathcal{H}(\Sigma)$.

$$H^\bullet(\mathrm{Obs}_{\mathrm{loc}}, Q) \to \mathrm{End}(\mathcal{H}(\Sigma)). \tag{1.2}$$

If we place bulk local operators at different points separated from the line defects that we inserted, then as a consequence of the twisted 3d theory being topological along $\mathbb{R}$, the action of the operators on $\mathcal{H}(\Sigma)$ all commute. This is a general pattern of the quantum integrable system, where bulk local operators play the roles of commuting Hamiltonians. For example, for theories with only vector multiplet at critical Chern Simons level, this construction will lead to Hitchin integrable system [28, 29] and specifically the Gaudin integrable system [30, 31] for $\Sigma$ a punctured sphere. It will be interesting to figure out the integrable systems associated with more general 3d $\mathcal{N} = 2$ theories possibly with both vector and chiral multiplets.

**Higher algebra and Deligne's conjecture**   Structures of topological quantum field theories, or quantum field theories in general, impose structures on the space of observables [32]. For the case of $d$ dimensional TQFT, the algebraic structures of the local operators at the chain level are captured by the notion of $E_d$ algebra. At the level of cohomology, an $E_d$ algebra becomes a shifted version of Poisson algebra, known as $P_d$ algebra. The bulk boundary relations then predict that the derived center of the boundary will carry these structures. For example in

$2d$, we have the well known Gerstenhaber structure on the Hochschild cohomology $HH^\bullet(A)$ [33]. If we look at the chain level, this becomes Deligne's conjecture (now a theorem) on the existence of $E_2$ structure on the Hochschild complex [34,35]. In $d \geq 2$ dimension, "higher" Deligne's conjecture states that $E_{(d-1)}$ Hochschild cohomology is furnished with an $E_d$ algebra structure [20,36].

We wish to find analogous statements for the holomorphic topological theories in $3d$. The general structure of the local operators should be captured by a higher analog of chiral algebra that also includes the OPE structures along $\mathbb{R}$. Though such a structure is still mysterious to us, if we take cohomology, this structure becomes the more familiar (shifted) Poisson vertex algebra [38, 39]. Therefore, we hope that the self-Ext $\mathrm{Ext}^\bullet_{V-\mathrm{mod}}(V, V)$ of a vertex algebra $V$ can be endowed with a shifted Poisson vertex algebra structure. If we look at the chain level, $\mathrm{RHom}_{V-\mathrm{mod}}(V, V)$ should have the higher analog of vertex algebra structure. A proper chain complex, namely, a chiral generalization of the Hochschild complex, is needed to elucidate these structures. We believe that the chiral deformation complex constructed in [37] is the right one. Moreover, when $V$ comes equipped with a stress energy tensor, namely $V$ is a conformal vertex algebra, the bulk theory of it becomes topological (see [12] for more detail). In this case, the self-RHom should be endowed with the structure of an $E_3$ algebra. We explain this story in more detail in Section 6.4.

# 2 Review of holomorphic twisted $3d$ $\mathcal{N} = 2$ theory

In this paper, we mainly work in the flat Euclidean spacetime $\mathbb{R}_t \times \mathbb{C}_{z,\bar{z}}$. The $3d$ $\mathcal{N} = 2$ SUSY algebra has generators $Q_\pm, \bar{Q}_\pm$ and commutation relations

$$
\begin{aligned}
\{Q_+, \bar{Q}_+\} &= -2i\partial_{\bar{z}}, & \{Q_-, \bar{Q}_-\} &= 2i\partial_z, \\
\{Q_+, \bar{Q}_-\} &= \{Q_-, \bar{Q}_+\} = i\partial_t.
\end{aligned}
\tag{2.1}
$$

In [11, 12], a class of $3d$ holomorphic topological theories is studied. Such a theory arises from a $3d$ $\mathcal{N} = 2$ supersymmetric field theory after performing a holomorphic twist. This amounts to changing the cohomological degree of fields using the $R$ symmetry and adding the supercharge $\bar{Q}_+$ into the BRST differential. By adding $\bar{Q}_+$ into the BRST differential, many $\bar{Q}_+$-exact terms can be removed and we get a new (and much simpler) quantum field theory. The fact that $\partial_t$ and $\partial_{\bar{z}}$ are $\bar{Q}_+$-exact tells us that all correlation functions of the new theory will be independent of $t$ and $\bar{z}$. We call such a theory holomorphic topological, meaning that it depends topologically in the $t$ direction and holomorphically in the $z$ direction. This explains the name holomorphic twist or, to be more precise, holomorphic topological twist.

In this section, we review the $3d$ $\mathcal{N} = 2$ theory in the twisted formalism introduced in [11, 12]. The advantage of directly working in the twisted formalism is that the field content, the action functional and the equations of motion are greatly simplified, while many structures that are contained in the $\bar{Q}_+$ cohomology of the original physical theory are still preserved in the twisted formalism. We then recast the theory into AKSZ formalism. This encodes classical information of the theory into a geometric structure, which paves the way for performing geometric quantization later.

## 2.1 Twisted theory in BV formalism

A $3d$ $\mathcal{N} = 2$ SUSY theory containing both vector multiplets and chiral multiplets is specified by the following data.

1. A compact gauge group $G$. (In twisted formalism, we always work with its complexification. So we also use $G$ to denote the complexification of the gauge group in the following.)

2. A unitary representation $V$ of $G$. (After complexification, $V$ becomes a complex linear representation.)

3. A $G$ invariant polynomial $W : V \to \mathbb{C}$, called superpotential.

4. An integer (or half-integer) $k$ called Chern-Simons level.

To perform the holomorphic twist we require additional data of a $U(1)_R$ symmetry, under which $V$ decomposes into subspaces with different $R$ charges: $V = \oplus_r V^{(r)}$. Under the $R$ charge assignment, the superpotential $W$ must be a quasi-homogeneous function of $R$-charge 2.

In this paper, we directly work with the twisted theories. These theories are generally defined on a 3-manifold $M$ with a transverse holomorphic foliation(THF). A 3-manifold is equipped with a THF if it has local coordinate patches $(t, z) \in \mathbb{R} \times \mathbb{C}$ and the transition functions take the form

$$\left( f(t, z, \bar{z}), h(z), \bar{h}(\bar{z}) \right),\tag{2.2}$$

where $h(z)$ is a holomorphic function. The above form of coordinate transformation implies that the (anti)holomorphic 1-forms $\Omega^{1,0}(\Omega^{0,1})$ are globally well defined. And we have well defined $\partial, \bar{\partial}$ operators. The $dt$ direction, on the other hand, is not globally defined. However, we can define the quotient spaces $\Omega^1/\Omega^{1,0}$ and $\Omega^1/\Omega^{0,1}$. The projection map $p : \Omega^1 \to \Omega^1/\Omega^{1,0}$ can be written in local coordinates as

$$p : \ f_t dt + f_z dz + f_{\bar{z}} d\bar{z} \to f_t dt + f_{\bar{z}} d\bar{z}.\tag{2.3}$$

Using this projection, we can define a modified differential operator as

$$\hat{d} = p \cdot d : \ f \to \partial_t f \, dt + \partial_{\bar{z}} f \, d\bar{z}.\tag{2.4}$$

We also introduce a dg algebra

$$\mathcal{A}^\bullet = \Omega^\bullet / (\Omega^\bullet \wedge \Omega^{1,0}).\tag{2.5}$$

In local coordinates, it takes the form $C^\infty(U)[dt, d\bar{z}]$. The differential $\hat{d}$ naturally extends to $\mathcal{A}^\bullet$ and we still denote it by $\hat{d}$. There are natural variants of $\mathcal{A}^\bullet$:

$$\mathcal{A}^{\bullet,k} = \mathcal{A}^\bullet \otimes \Omega^{k,0}.\tag{2.6}$$

In local coordinates, it takes the form $C^\infty(U)[dt, d\bar{z}]dz^k$. The operator $\hat{d}$ is still well defined on $\mathcal{A}^{*,k}$ and takes the form $\hat{d} = dt\partial_t + d\bar{z}\partial_{\bar{z}}$ in local coordinates. The wedge product gives us a natural pairing

$$\mathcal{A}^{i,k} \otimes \mathcal{A}^{j,l} \to \mathcal{A}^{i+j,k+l}.\tag{2.7}$$

Integration on $M$ gives us a map

$$\int : \mathcal{A}^{2,1} = \Omega^3(M, \mathbb{C}) \to \mathbb{C}.\tag{2.8}$$

In this paper we study the local properties of these theories, hence it suffices to work in a local coordinate. The cohomology of $(\mathcal{A}^{\bullet,k}, \hat{d})$ in a local coordinate consists of those functions $\mathrm{Hol}(D)$ which are independent of $t$ and holomorphic in $z$.

A convenient way to analyze gauge theories is through BV formalism. By adding ghosts, anti-fields, and anti-ghosts, a gauge theory can be described by a differential graded (dg) Lie algebra equipped with an invariant pairing of degree $-3$. For the holomorphic twisted $3d$ $\mathcal{N} = 2$ theory, the field content in BV formalism can be organized into the following "superfields"

- A gauge field: $\mathbf{A} = c + A + B^* \in \mathcal{A}^\bullet \otimes \mathfrak{g}[1]$.

- A coadjoint valued field: $\mathbf{B} = B + A^* + c^* \in \mathcal{A}^{\bullet,1} \otimes \mathfrak{g}^*$.

- A matter field in representation $V$: $\boldsymbol{\Phi} = \phi + \eta^* + \psi^* \in \oplus_k \mathcal{A}^{\bullet,k} \otimes V^{(k)}$.

- A matter field in the dual representation $V^*$: $\boldsymbol{\Psi} = \psi + \eta + \phi^* \in \oplus_k \mathcal{A}^{\bullet,1-k} \otimes (V^{(k)})^*[1]$.

where the symbol $[1]$ indicates a shift of cohomological degree by 1, so that the gauge field $A$ has degree 0. Here, the fields $\mathbf{A}$, $\mathbf{B}$ are obtained from twisting the vector multiplets of the physical 3d $\mathcal{N} = 2$ theory, and $\boldsymbol{\Phi}$, $\boldsymbol{\Psi}$ are obtained from twisting the chiral multiplets. Thus, we will also use the names "vector multiplets" and "chiral multiplets" to call them in the twisted formalism.

We can write down the full BV action functional in terms of the above data

$$S_{BV} = \int \left\langle \mathbf{B}, \hat{d}\mathbf{A} + \frac{1}{2}[\mathbf{A}, \mathbf{A}] \right\rangle + \left\langle \boldsymbol{\Psi}, (\hat{d} + \mathbf{A})\boldsymbol{\Phi} \right\rangle + W(\boldsymbol{\Phi}) + \frac{k}{8\pi} \left\langle \mathbf{A}, \partial \mathbf{A} \right\rangle . \tag{2.9}$$

Here the inner product $\langle -, - \rangle$ is induced from the dual pairing between $\mathfrak{g}, \mathfrak{g}^*$ and $V, V^*$ and wedge product of differential forms. The integration map $\int$ is defined by 2.8 that picks up the top degree forms.

We mention that in the presence of a Chern-Simons term, the gauge theory part is actually equivalent to a physical Chern-Simons theory. This fact can be obtained via a field redefinition [12]. We provide another explanation in the next section.

Another important structure in the BV formalism of QFT is the degree 1 bracket that pairs fields/ghosts with anti-fields/anti-ghosts, called the BV bracket. In our theory, the BV bracket is given by

$$\{\boldsymbol{\Phi}(x), \boldsymbol{\Psi}(y)\}_{BV} = \{\mathbf{A}(x), \mathbf{B}(y)\}_{BV} = \delta(x - y)\mathrm{dVol} . \tag{2.10}$$

The BRST operator is easy to calculate from the BV bracket using the formula $Q = \{S, -\}_{BV}$. We find

$$
\begin{aligned}
Q\mathbf{A} &= \hat{d}\mathbf{A} + [\mathbf{A}, \mathbf{A}] , \\
Q\mathbf{B} &= \hat{d}_{\mathbf{A}}\mathbf{B} + \mu(\boldsymbol{\Phi}, \boldsymbol{\Psi}) - \frac{k}{4\pi}\partial\mathbf{A} , \\
Q\boldsymbol{\Phi} &= \hat{d}_{\mathbf{A}}\boldsymbol{\Phi} , \\
Q\boldsymbol{\Psi} &= \hat{d}_{\mathbf{A}}\boldsymbol{\Psi} + \frac{\delta W}{\delta \boldsymbol{\Phi}} ,
\end{aligned}
\tag{2.11}
$$

where we used (shifted) moment map $\mu : V \otimes V^*[1] \to \mathfrak{g}^*$ associated with the action of $G$ on $V$.

## 2.2 AKSZ formalism

Roughly speaking, the geometry of the (classical) BV formalism is given by a dg manifold equipped with a $-1$ shifted symplectic form. A BV action functional is an even function $S$ such that $Q = \{S, -\}$. In some cases, the BV field theory can be formulated as an AKSZ sigma model [40]. In the AKSZ construction, we have a dg manifold $N$ equipped with a volume form of degree $l$ and a dg manifold $M$ equipped with a symplectic form of degree $-1-l$. Then we consider the mapping space $\mathrm{Map}(N, M)$, which can be shown to have the structure of a classical BV field theory. The natural $(-1)$-shifted symplectic structure $\omega$ is described as follows. Given $f : N \to M$, the tangent space of $\mathrm{Map}(N, M)$ at $f$ is $T_f \mathrm{Map}(N, M) = \Gamma(N, f^* TM)$. Then the symplectic form on the mapping space $\mathrm{Map}(N, M)$ can be written as

$$\omega(\alpha, \beta) = \int_N \mathrm{dVol}_N(\alpha, \beta)_M , \quad \alpha, \beta \in \Gamma(N, f^* TM) , \tag{2.12}$$

where $(\ ,\ )_M$ denotes the pairing on $T_M$ that comes from the symplectic structure on $M$.

In this section, we show that the holomorphic twisted 3d $\mathcal{N} = 2$ theory described above can also be formulated as an AKSZ sigma model.

We first consider the vector multiplet with gauge group $G$ at Chern Simons level 0, the field content is

$$(\mathbf{A}, \mathbf{B}) \in \mathcal{A}^{\bullet} \otimes \mathfrak{g}[1] \oplus \mathcal{A}^{\bullet,1} \otimes \mathfrak{g}^{*}. \tag{2.13}$$

As a graded manifold, these fields describe maps $\mathbb{R}_{dR} \times \mathbb{C}_{\bar{\partial}} \to \mathfrak{g}[1] \oplus \mathfrak{g}^{*}$. We can observe that the BRST differential is inherited from the Chevalley-Eilenberg differential of $C^{\bullet}(\mathfrak{g}; \operatorname{Sym}\mathfrak{g}^{*}) = \mathbb{C}[T^{*}[1]B\mathfrak{g}]$. Besides, the BV bracket comes from the natural symplectic form on the shifted cotangent bundle $T^{*}[1]B\mathfrak{g}$. Therefore, in the AKSZ formulation, the twisted vector multiplet (without a CS term) can be formulated as the mapping space: $\operatorname{Map}(\mathbb{R}_{dR} \times \Sigma_{\bar{\partial}}, T^{*}[1]B\mathfrak{g})$. We should be careful here. By writing the target as $T^{*}[1]B\mathfrak{g}$, we only defined a perturbation theory. The "global" version of this theory is

$$\operatorname{Map}\left(\mathbb{R}_{dR} \times \Sigma_{\bar{\partial}}, T^{*}[1]BG\right). \tag{2.14}$$

We can also understand the theory with a Chern-Simons term in a similar fashion. First, we rewrite the mapping space as follows

$$\operatorname{Map}(\mathbb{R}_{dR} \times \Sigma_{\bar{\partial}}, T^{*}[1]BG) = \operatorname{Map}(\mathbb{R}_{dR} \times T[1]\Sigma_{\bar{\partial}}, BG). \tag{2.15}$$

This is because $\mathcal{O}(\mathbb{R}_{dR} \times T[1]\Sigma_{\bar{\partial}}) = \mathcal{A}^{\bullet}[\epsilon] = \mathcal{A}^{\bullet} \oplus \mathcal{A}^{1,\bullet}[-1]$. Therefore the right hand side gives the same field content and BRST differential as the left hand side.

Adding a Chern-Simons term has the effect of turning $T[1]\Sigma_{\bar{\partial}}$ into $\Sigma_{dR}$. Thus we obtained the standard description of Chern-Simons theory [40]

$$\operatorname{Map}\left((\mathbb{R} \times \Sigma)_{dR}, BG\right). \tag{2.16}$$

We can also make the dependence on complex structure more explicit. By using the identity $\operatorname{Map}(X \times Y, Z) = \operatorname{Map}(X, \operatorname{Map}(Y, Z))$, we have

$$\operatorname{Map}((\mathbb{R} \times \Sigma)_{dR}, BG) = \operatorname{Map}\left(\mathbb{R}_{dR}, T_{\mathbf{k}}^{*}\operatorname{Map}(\Sigma_{\bar{\partial}}, BG)\right) = \operatorname{Map}\left(\mathbb{R}_{dR}, T_{\mathbf{k}}^{*}\operatorname{Bun}_{G}(\Sigma)\right). \tag{2.17}$$

In this way, we understand Chern-Simons theory as a topological quantum mechanics with target $T_{\mathbf{k}}^{*}\operatorname{Bun}_{G}(\Sigma)$ as a twisted cotangent bundle of $\operatorname{Bun}_{G}(\Sigma)$. Here we actually understand the Chern-Simons level $\mathbf{k}$ as a Cartan-Killing form, which can be further identified as a symplectic structure on $BG$ and integrated to a closed two form on $\operatorname{Bun}_{G}(\Sigma)$. From this perspective, we can regard the theory without Chern-Simons term as a topological quantum mechanics with target $T^{*}\operatorname{Bun}_{G}(\Sigma) = \operatorname{Higgs}_{G}(\Sigma)$, the Hitchin moduli space of $G$ bundle on $\Sigma$. This suggests a close relation between this theory and the Hitchin system.

For the chiral multiplet, the field content is

$$(\boldsymbol{\Phi}, \boldsymbol{\Psi}) \in \bigoplus_{k} \mathcal{A}^{\bullet,k} \otimes V^{(k)} \oplus \mathcal{A}^{\bullet,1-k} \otimes \left(V^{(k)}\right)^{*}[1]. \tag{2.18}$$

They can be described by the mapping space $\operatorname{Map}(\mathbb{R}_{dR} \times \Sigma_{\bar{\partial}}, T^{*}[1]V)$.

We can describe twisted theory of chiral multiplet with a superpotential along the same line. As we can see from 2.11, turning on a superpotential $W$ deforms the BRST differential $Q$ by a term $\frac{\delta W}{\delta \boldsymbol{\Phi}} \frac{\partial}{\partial \boldsymbol{\Psi}}$. As is reviewed in Appendix A, we see the same differential A6 from the algebra of function on $\operatorname{dCrit}(W)$ – the derived critical locus of $W$. This tells us that turning on a superpotential amounts to replacing $T^{*}[1]V$ with $\operatorname{dCrit}(W)$ as the target. Thus we conclude that the AKSZ formulation for chiral multiplet with superpotential $W$ is given by the mapping space:

$$\operatorname{Map}\left(\mathbb{R}_{dR} \times \Sigma_{\bar{\partial}}, \operatorname{dCrit}(W)\right). \tag{2.19}$$

More generally, for theory with both vector and chiral multiplets, the AKSZ formulation is given by the mapping space

$$\mathrm{Map}\left(\mathbb{R}_{dR} \times \Sigma_{\bar{\partial}}, \mathrm{dCrit}(W : V/G \to \mathbb{C})\right), \tag{2.20}$$

or equivalently

$$\mathrm{Map}\left(\mathbb{R}_{dR} \times \Sigma_{\bar{\partial}}, \mathrm{dCrit}(W) /\!\!/ G\right). \tag{2.21}$$

We review the corresponding supermanifolds in Appendix A. We can check that this is compatible with previous special cases. For $G$ trivial, the target automatically becomes dCrit($W$). For $V = 0$, $V/G = BG$, the derived critical locus for trivial superpotential is the 1 shifted cotangent bundle, which gives us $T^*[1]BG$.

# 3 Bulk algebras and their characters

In this section, we study the bulk local operator algebra for the holomorphic twisted 3d $\mathcal{N} = 2$ theory. We study both perturbative and non perturbative algebra. This includes the consideration of monopole operators, which cannot be expressed as polynomial functions of the fields. After appropriately identifying the operators and their quantum numbers, we study the character of the algebra, which reproduces the 3d superconformal index of the corresponding SUSY theory.

At the perturbative level, the local operator algebra Obs$^{\mathrm{per}}$ is easy to describe. Perturbative local operators are polynomial functions in the fields $(\mathbf{\Phi}, \mathbf{\Psi}), (\mathbf{B}, \mathbf{A})$, subject to the BRST differential 2.11. As we have mentioned, locally, the cohomology of $(\mathcal{A}^{\bullet,k}, \hat{d})$ consists of those functions Hol($D$) that are independent in $t$ and holomorphic in $z$. Therefore, by only taking the bottom component of the superfields and restricting to fields that are holomorphic in $z$, we find a smaller but quasi-isomorphic complex. This space consists of functions in the fields $\{\phi(z), \psi(z), b(z), c(z)\}$. The differential can be schematically written as

$$
\begin{aligned}
Qc &= \frac{1}{2}[c, c], \\
Qb &= [c, b] + \mu(\phi, \psi) - \frac{k}{4\pi}\partial c, \\
Q\phi &= [c, \phi], \\
Q\psi &= [c, \psi] + \frac{\delta W(\phi)}{\delta \phi}.
\end{aligned}
\tag{3.1}
$$

We can also explain this result in a geometric manner. We consider a cylinder $C_\epsilon = [-\epsilon, \epsilon] \times D_\epsilon$ where $D_\epsilon$ is a disk of radius $\epsilon$ in $\mathbb{C}$. The perturbative local operators are functions on the (derived) space of solution to the equation of motion on $C_\epsilon$ as $\epsilon \to 0$.

$$\mathrm{Obs}^{\mathrm{per}} = \lim_{\epsilon \to 0} \mathbb{C}[\mathrm{EOM}(C_\epsilon)]. \tag{3.2}$$

The AKSZ formalism of the theory immediately tells us that perturbatively, EOM($C_\epsilon$) is

$$\mathrm{Map}\left([-\epsilon, \epsilon]_{dR} \times (D_\epsilon)_{\bar{\partial}}, \mathrm{dCrit}(W : V/\mathfrak{g} \to \mathbb{C})\right). \tag{3.3}$$

This space describes maps constant along the real direction and holomorphic on $D_\epsilon$. Equivalently, we can ignore $[-\epsilon, \epsilon]_{dR}$ and we replace the disk by the formal disk $\mathbb{D} = \mathrm{Spec}(\mathbb{C}[[z]])$ as $\epsilon \to 0$. Then the space of solutions to the equations of motion can be described by algebraic maps $\{\mathbb{D} \to \mathrm{dCrit}(W : V/\mathfrak{g} \to \mathbb{C})\}$, which can be identified with the infinite jet $J_\infty(\mathrm{dCrit}(W : V/\mathfrak{g} \to \mathbb{C}))$ of the target space dCrit($W : V/\mathfrak{g} \to \mathbb{C}$). Then we find the space of

perturbative local operators to be the space of functions: $\mathbb{C}[J_\infty(\mathrm{dCrit}(W : V/\mathfrak{g} \to \mathbb{C}))]$. Recall that given an affine scheme $X$ with ring of functions $\mathbb{C}[X] = \mathbb{C}[x^i]$, its infinite jet $J_\infty X$ has ring of functions $\mathbb{C}[J_\infty X] = \mathbb{C}[x^i_n, n = 0, 1 \dots]$. For a derived scheme, its infinite jet scheme has a similar interpretation. Given the coordinate description of $\mathrm{dCrit}(W : V/\mathfrak{g} \to \mathbb{C}))]$ as in A11, A14, we find that functions on the infinite jet $J_\infty(\mathrm{dCrit}(W : V/\mathfrak{g} \to \mathbb{C}))$ are the same as functions of the fields $\{\phi(z), \psi(z), b(z), c(z)\}$ with differential 3.1.

However, even at the perturbative level, this answer is not correct, as we need to be more careful about the ghost. In a quantum field theory with gauge symmetry, we can think of introducing ghosts as a homological method to compute gauge invariant local operators. However, for a compact gauge group, taking $G$ invariant is already an exact functor and there is no need to do this homologically. It suffices to introduce only the higher order ghost modes. A similar problem appears in [12] in the discussion of the boundary algebra, where more details about (derived) invariants of Lie algebra and Lie group are provided. The upshot is that, instead of introducing constant ghost mode in the local operator algebra, we impose the $G$ invariant by hand. Therefore, the perturbative operator algebra should be something that looks like $\mathbb{C}[\partial c, \partial^2 c, \dots]^G$ instead of $\mathbb{C}[c, \partial c, \dots]$.

From a physical perspective, the first order derivative of the $c$ ghost is cohomologous to the gaugino in the physical SUSY theory before twisting [12]. In the computation of the superconformal index in physics literature [16, 18], only gaugino and its derivatives contribute to the index and it always involves an integration over the gauge $G$ fugacities. This essentially means that only $G$ invariant polynomials without constant ghost mode $c$ contribute to the bulk algebra.

Accordingly, when $G$ is a compact group, we write the perturbative local operator algebra as

$$\mathrm{Obs}^{\mathrm{per}} = \mathbb{C}[J_\infty(\mathrm{dCrit}(W) /\!\!/ G)]. \tag{3.4}$$

The symplectic quotient here is understood in a derived sense. The derived quotient by $J_\infty G = G[[z]]$ is divided into two parts using the decomposition $G[[z]] = G \ltimes zG[[z]]$. Taking the derived quotient by $zG[[z]]$ amounts to adding ghost valued in the Lie algebra and taking quotient by $G$ amounts to taking the $G$ invariant by hand.

In this paper, all constructions are assumed to be "derived". Mathematically, "derived" means we keep track of all the homological information. Physically, this means that we keep track of all the ghost, anti-field, etc. For the algebra of local operators, we consider the whole complex of local operators of all ghost numbers. Typically, physicists only care about local operators of zero ghost number, as operators of non-zero ghost number do not have direct physically interpretation. However, in our setup, considering the whole complex is necessary because the twisting procedure will mix the ghost numbers and the R charges. In other words, physical fields in the original SUSY theory can become ghosts after the twist. As we will see later, to reproduce the $3d$ $\mathcal{N} = 2$ superconformal index from the twisted theory, we need to take into account of the whole complex of local operators. This provide another evidence why derived construction is necessary in our setup.

A full description of the local operators should also include monopole operators, which are defined by specifying some singular behavior of fields around a point. A general strategy for dealing with these operators is to use state-operator correspondence. For an $n$ dimensional TQFT, state-operator correspondence tells us that the space of local operators can be identified with the Hilbert space of state $Z(S^{n-1})$ on the $n - 1$ sphere surrounding the point. More generally, local operators of an $n$ dimensional CFT are in one-to-one correspondence with states in the radially quantized Hilbert space of the theory. This method is used in [13–15] to construct monopole operators in "physical" 3d theories, and is also closely related to the BFN construction of the Coulomb branch operators [41, 42]. The theory we consider is holomorphic topological. Therefore, instead of the sphere $S^2$, it will be more convenient to consider the

following punctured cylinder:

$$C^\times = D_\epsilon \times [-\epsilon, \epsilon] \backslash \{(0,0)\}. \tag{3.5}$$

We take the limit $\epsilon \to 0$ in the end. Our goal is to construct the Hilbert space associated with this punctured cylinder. The standard procedure for doing this consists of two steps, we first construct the phase space on this cylinder as a symplectic manifold and then perform the geometric quantization.

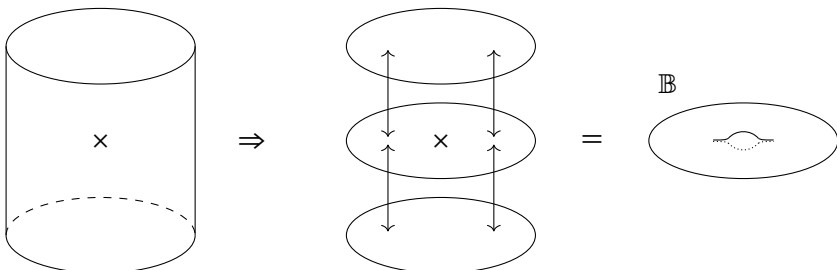

First, we analyze the phase space $\text{EOM}(C^\times)$ as the derived space of solutions to the equations of motion on $C^\times$. It is important to note that this space describes maps that are independent of $t$. Therefore, we can make the following simplification. We take two sets of solutions on $C_\epsilon$, which are determined only by the data on $D_\epsilon$. Then we glue the two sets of solutions together via an isomorphism over the punctured disk $D_\epsilon^\times$. We expect to have an equivalence between the phase space on $C^\times$ and the space constructed via the gluing procedure. Again, taking $\epsilon \to 0$ we replace disk by formal disk $\mathbb{D}$. The above analysis suggests that the phase space can be constructed formally as solutions on the "ravioli", or "formal bubble" $\mathbb{B} = \mathbb{D} \bigsqcup_{\mathbb{D}^\times} \mathbb{D}$ defined by gluing two formal disks through a punctured disk. A more detailed analysis of the phase space is performed in the following sections.

## 3.1 Chiral multiplets

For theory with only chiral multiplets, we do not expect to have any non-perturbative local operators. If we try to solve the equations of motion, any solution will be constant in $t$, and there couldn't exist non-perturbative objects that can localize in the $t$ direction. In this case, the smallest dimensional non-perturbative objects are line defects, instead of local operators. Therefore, the perturbative description of the local operators should suffice, and we get from 3.4 functions on the infinite jet of the derived critical locus of $W$.

$$\text{Obs} = \mathbb{C}[J_\infty \text{dCrit}(W)]. \tag{3.6}$$

Although monopole operators are absent, the state-operator correspondence is still valid to describe the local operators. This provides us with an alternative way to compute the space of local operators, which should give us the same answer as the perturbative analysis. We will do this in the following because this process exhibits us the simplest example of computation on the formal bubble, which we will generalize later.

For chiral multiplets without superpotential, the phase space we need to consider is $\text{Map}(\mathbb{B}_{\bar\partial}, T^*[1]V)$. The "derived" structure of this phase space is essential for our analysis. For example, the ring of functions on $\mathbb{B}_{\bar\partial}$ can be modeled on the Čech cohomology, which gives us the following complex

$$0 \to \mathbb{C}[[z]] \oplus \mathbb{C}[[z]] \xrightarrow{d} \mathbb{C}((z)) \to 0, \tag{3.7}$$

where the differential $d$ is given by $d(f(z), g(z)) = f(z) - g(z)$ for $(f(z), g(z)) \in \mathbb{C}[[z]] \oplus \mathbb{C}[[z]]$. Similarly, the phase space of the free chiral multiplet is modeled on the complex

$$0 \rightarrow (T^*[1]V)[[z]]^{\oplus 2} \xrightarrow{d} (T^*[1]V)((z)) \rightarrow 0\,. \tag{3.8}$$

More precisely, we have direct sum of complexes $0 \rightarrow (V^{(r)}[[z]]dz^r)^{\oplus 2} \rightarrow V^{(r)}((z))dz^r \rightarrow 0$ and $0 \rightarrow (V^{(r)*}[1][[z]]dz^{1-r})^{\oplus 2} \rightarrow V^{(r)*}[1]((z))dz^{1-r} \rightarrow 0$. Here we can absorb the factor $dz^r$ into $V$ and $dz^{1-r}$ into $V^*[1]$. By this notation, $V$ now has twisted spin grading and so does $V^*$. To correctly incorporate the spin grading of them, the shifted cotangent bundle $T^*[1]V$ need to be corrected to $T^*[1](1)V$, where by $(1)$ we shift the twisted spin grading. To simplify the notation we still denote it by $T^*[1]V$. Remembering the shifting of spin grading is important in the computation of the characters of the algebra.

The above complex has cohomology

$$\begin{aligned}
\mathcal{P} &= T^*[1]V[[z]] \oplus (T^*[1]V((z))/T^*[1]V[[z]])[-1] \\
&= V[[z]] \oplus V^*[1][[z]] \oplus (V^*[1][[z]])^* \oplus (V[[z]])^*\,.
\end{aligned} \tag{3.9}$$

The symplectic form is given by the natural pairing between $V[[z]]$ and $(V[[z]])^*$ and the pairing between $V^*[1][[z]]$ and $(V^*[1][[z]])^*$.

To perform geometric quantization, we need to choose a polarization. We can choose the following Lagrangian fibration

$$\pi : \mathcal{P} \rightarrow T^*[1]V \otimes \check{H}^0(\mathbb{B}) = V[[z]] \oplus V^*[1][[z]]\,. \tag{3.10}$$

With this choice of polarization, we consider functions on $\mathcal{P}$ that are constant along the fiber of $\pi$, which is equivalent to functions on $(T^*[1]V)[[z]] = J_\infty T^*[1]V$.

We could, of course, choose other polarizations. For example, we could take the polarization to be the Lagrangian fibration over

$$\mathrm{Map}(\mathbb{B}_{\bar{\partial}}, V) = V[[z]] \otimes (V^*[1][[z]])^*\,. \tag{3.11}$$

This looks like the polarization much often used in other related setups. However, global functions on $(V^*[1][[z]])^* = V((z))/V[[z]][-1]$ are not well defined. Note that we can identify $(V^*[1][[z]])^*$ with $V[z^{-1}]z^{-1}[-1]$ via the residue pairing. If we naively take functions on $V[z^{-1}]z^{-1}[-1]$ we meet the problem of operators of arbitrary negative spin. This is because we assigned $z$ with spin $-1$, so that linear dual of $z^{-n}$ has spin $-n$. Instead, "distributions" on $V \otimes \check{H}^1(\mathbb{B})$ (or global sections of the dualizing sheaf) are well defined. Note that distributions behave like "dual" of functions. So they also have the correct spins. Therefore, in this polarization, we need to define the Hilbert space to be functions on $V \otimes \check{H}^0(\mathbb{B})$ tensoring with "distributions" on $V \otimes \check{H}^1(\mathbb{B})$. A similar phenomenon appears in [12] in studying the boundary chiral algebra, where Dolbeault homology instead of cohomology is used to construct boundary operators, and in [42] where equivariant Borel-Moore homology is used to define Coulomb branch operators. In any case, the space of local operators we get should be the only reasonable answer by appropriate construction, which is the space of functions on $J_\infty T^*[1]V$:

$$\mathrm{Obs} = \mathbb{C}[J_\infty T^*[1]V]\,. \tag{3.12}$$

To turn on a superpotential, we simply replace $T^*[1]V$ by $\mathrm{dCrit}(W)$ in the above analysis. This gives us $\mathrm{Obs} = \mathbb{C}[J_\infty \mathrm{dCrit}(W)]$, which is the same as the perturbative analysis.

It is useful to write down an explicit expression for the local operators. Here we take the usual physical notation. We denote $\partial^n \phi_i$ the $n$-th $z$ derivatives of the bottom component of the field $\Phi_i$ at $z = 0$, and $\partial^n \psi_i$ the $n$-th $z$ derivatives of the bottom component of the field $\Psi_i$ at $z = 0$. They serve as the coordinates on the infinite jet of $V$ and $V^*[1]$ respectively. Then

the space of local operators can be written as $\mathbb{C}[\{\partial^n \phi_i, \partial^n \psi_i\}_{n \geq 0, i}]$. The superpotential $W$ gives us a non-zero differential

$$Q\partial^n \psi = \partial^n \left( \frac{\delta W(\phi)}{\delta \phi} \right). \tag{3.13}$$

This expression will be useful later when we compare the Ext computation with the direct bulk computation.

## 3.2 Theories with gauge fields

In this section, we take vector multiplets into consideration. The existence of monopole operators makes these theories much more complicated than theories with only chiral multiplets. We will not attempt to give a systematic study of vector multiplets. Instead, we focus on theories with only abelian gauge group. Their moduli spaces of equations of motion have simpler structure, which allows us to avoid many technical details but is still able to give us some nontrivial results.

### 3.2.1 Perturbative algebra

Let's first consider pure gauge theories. According to our previous discussion 3, perturbative local operators consist of $G$ invariant functions of fields $b(z), c(z)$ with constant ghost modes removed. This is exactly the cochain complex of relative Lie algebra cohomology. We can also obtain this result through our "definition" 3.4. By taking $V = 0$, we have

$$\text{Obs}^{\text{per}} = \mathbb{C}[J_\infty(\mathfrak{g}^*/G)] = \mathbb{C}[\mathfrak{g}^*[[z]]/G[[z]]]. \tag{3.14}$$

As we have explained, we take $zG[[z]]$ invariant by introducing ghost valued in the Lie algebra of $zG[[z]]$, and the $G$ invariant is taken by hand. Then we get relative Lie algebra cohomology

$$\text{Obs}^{\text{per}} = C^\bullet(\mathfrak{g}[[z]], \mathfrak{g}, \text{Sym}(\mathfrak{g}^*[[z]])^*). \tag{3.15}$$

When $\mathfrak{g}$ is semisimple, we can identify $\mathfrak{g}^*$ with $\mathfrak{g}$ by the Cartan-Killing form. Then the local operator algebra can be equivalently written as

$$\text{Obs}^{\text{per}} = C^\bullet(\mathfrak{g}[[z, \varepsilon]], \mathfrak{g}), \tag{3.16}$$

where $\varepsilon$ is an odd parameter satisfying $\varepsilon^2 = 0$. This relative Lie algebra cohomology is computed in [43]. Here we follow the notation of [44]. Denote $C_{\mathfrak{g}^*} := \mathfrak{g}^*/G = \text{Spec}(\mathbb{C}[\mathfrak{g}^*]^G)$. We choose generators $P_i, i = 1, \ldots l$ of $\mathbb{C}[\mathfrak{g}^*]^G$ of degree $d_i + 1$. Then $\mathbb{C}[C_{\mathfrak{g}^*}] = \mathbb{C}[P_1, \ldots, P_l]$. Define the local Hitchin space as

$$C_{\mathfrak{g}^*, K} = \Gamma(\mathbb{D}, K \times_{\mathbb{C}^\times} C_{\mathfrak{g}^*}), \tag{3.17}$$

where $K$ is the canonical line bundle on the disk. We can identify the ring of functions $\mathbb{C}[C_{\mathfrak{g}^*, K}]$ on the local Hitchin space as the polynomial algebra $\mathbb{C}[\{P_{i,n}\}_{i=1,\ldots,l; n \geq 0}]$. It was proved in [43] that, there is an isomorphism of graded algebras

$$H^\bullet(\mathfrak{g}[[z, \varepsilon]], \mathfrak{g}) = \Omega^\bullet(C_{\mathfrak{g}^*, K}). \tag{3.18}$$

The ghost number zero part $H^0(\mathfrak{g}[[z, \varepsilon]], \mathfrak{g}) = \mathcal{O}(C_{\mathfrak{g}^*, K})$ of this algebra has been extensively studied in the context of vertex algebra, Hitchin system and geometric Langlands. We define the center $\mathfrak{z}(\hat{\mathfrak{g}}) = \text{End}_{\hat{\mathfrak{g}}_{\kappa_c}} V_{\kappa_c}$ of the vacuum Verma module $V_{\kappa_c}$ at the critical level. $\mathfrak{z}(\hat{\mathfrak{g}})$ has a

filtration induced from the filtration of $V_{\kappa_c}$. Then we have an isomorphism of the associated graded

$$\text{gr } \mathfrak{z}(\hat{\mathfrak{g}}) \cong \mathcal{O}(C_{\mathfrak{g}^*,K}). \tag{3.19}$$

Moreover, there is an isomorphism of filtered algebras [45, 46]

$$\mathfrak{z}(\hat{\mathfrak{g}}) \cong \mathcal{O}(\text{Op}_{L\mathfrak{g}}(\mathbb{D})), \tag{3.20}$$

where $\text{Op}_{L\mathfrak{g}}(\mathbb{D})$ is the space of $^L\mathfrak{g}$-opers on $\mathbb{D}$. The center $\mathfrak{z}(\hat{\mathfrak{g}})$ can be further identified with the classical $\mathcal{W}$ algebra $\mathcal{W}_\infty(^L\mathfrak{g})$ [45]. The appearance of the Langlands dual Lie algebra $^L\mathfrak{g}$ here is very interesting. The proof of 3.20 in [45, 46] and the appearance of the Langlands dual Lie algebra is essentially a consequence of T-duality in $2d$. We hope that there could also be an S-duality argument for it.

The algebras $\mathcal{O}(C_{\mathfrak{g}^*,K})$ and $\mathfrak{z}(\hat{\mathfrak{g}})$ also play important roles in the Hitchin integrable system and its quantization. There is a classical Hitchin homomorphism [28, 29]

$$h^{\text{cl}} : \mathcal{O}(C_{\mathfrak{g}^*,K}) \to \Gamma\left(\text{Bun}_G, p^*\mathcal{O}_{T^*\text{Bun}_G}\right), \tag{3.21}$$

and its quantization [29]

$$h : \mathfrak{z}(\hat{\mathfrak{g}}) \to \Gamma\left(\text{Bun}_G, \mathcal{D}'_{T^*\text{Bun}_G}\right). \tag{3.22}$$

This provides an instance of the connection between the twisted $3d$ $\mathcal{N} = 2$ theory and the integrable system that we outlined in the introduction 1.2.

When there is a bare Chern-Simons term, we turn on a differential $\varepsilon\partial_z$ on $\mathfrak{g}[[z,\varepsilon]]$ and make it into a dg Lie algebra. We have a quasi-isomorphism

$$\mathfrak{g}[[z,\varepsilon]]_{\text{deformed}} := (\mathfrak{g}[[z,\varepsilon]], \varepsilon\partial_z) \to \mathfrak{g}. \tag{3.23}$$

This map induces a quasi-isomorphism on Lie algebra cochain

$$C^\bullet(\mathfrak{g},\mathfrak{g}) \to C^\bullet(\mathfrak{g}[[z,\varepsilon]]_{\text{deformed}}, \mathfrak{g}). \tag{3.24}$$

Therefore the perturbative algebra for vector multiplet with a Chern-Simons term becomes trivial in cohomology. We can also see this by noting that, turning on the Chern-Simons term simply turn on the De Rham differential on $\Omega^\bullet(C_{\mathfrak{g}^*,K})$ [44].

In the presence of chiral multiplets, perturbative operator algebra is given by 3.4. However, we don't have too much knowledge about its cohomology at the moment.

### 3.2.2 Non-perturbative algebra

Now we turn to discuss the full non-perturbative algebra. In this paper, we only consider the special case of $U(1)$ gauge theory since the structure of the phase space is more accessible. The complexified gauge group is $G = \mathbb{C}^\times$, and the phase space for pure gauge theory is $\text{Map}(\mathbb{B}_{\bar{\partial}}, T^*[1]BG)$. Geometrically, the space $\text{Map}(\mathbb{B}_{\bar{\partial}}, T^*[1]BG)$ describes a $\mathbb{C}^\times$ bundle on $\mathbb{B}_{\bar{\partial}}$ together with a coadjoint valued section. The space of holomorphic $\mathbb{C}^\times$ bundles on $\mathbb{B}$ as a set is given by $\mathbb{Z}$. For each integer $m$, we get a $\mathbb{C}^\times$ bundle $P_m$ by defining its transition function on the overlap $\mathbb{D}^\times$ to be $z^m$. Therefore our phase space has a decomposition:

$$\text{EOM}(\mathbb{B}) = \prod_{m \in \mathbb{Z}} \text{EOM}^m(\mathbb{B}). \tag{3.25}$$

Each component $\text{EOM}^n(\mathbb{B})$ describes the space of coadjoint valued sections of $P_m$ module gauge transformation. It contains rich structures as a "derived space". It has a tangent space (complex) $\text{R}\Gamma(\mathbb{B}, P_m \times_{\mathbb{C}^\times} (\mathfrak{gl}_1^* \oplus \mathfrak{gl}_1[1]))$. Computation of its cohomology is similar to the chiral multiplet case, and we find

$$\mathfrak{gl}_1^*[[z]] \oplus \mathfrak{gl}_1[1][[z]] \oplus (\mathfrak{gl}_1^*((z))/\mathfrak{gl}_1^*[[z]])[-1] \oplus (\mathfrak{gl}_1((z))/\mathfrak{gl}_1[[z]]). \tag{3.26}$$

We used the fact that the $\mathbb{C}^\times$ action on $\mathfrak{gl}_1$ is trivial. The symplectic structure is given by the natural pairing between $\mathfrak{gl}_1^*[[z]]$ and $(\mathfrak{gl}_1((z))/\mathfrak{gl}_1[[z]])$, and the pairing between $\mathfrak{gl}_1[1][[z]]$ and $(\mathfrak{gl}_1^*((z))/\mathfrak{gl}_1^*[[z]])[-1]$.

Next, we construct the polarization. To avoid the subtle construction of half homology and half cohomology as in the discussion in Section 3.1, we don't use the Lagrangian fibration $T^*\mathrm{Map}(\mathbb{B}_{\bar{\partial}}, BG) \to \mathrm{Map}(\mathbb{B}_{\bar{\partial}}, BG)$. Instead, we take polarized sections to be constant along $(\mathfrak{gl}_1^*((z))/\mathfrak{gl}_1^*[[z]])[-1] \oplus (\mathfrak{gl}_1((z))/\mathfrak{gl}_1[[z]])$ directions as in Section 3.1. Using this polarization, we find the Hilbert space to be the space of functions on

$$\prod_{m \in \mathbb{Z}} J_\infty \mathfrak{gl}_1^* / J_\infty \mathbb{C}^\times \,, \tag{3.27}$$

where, as before, the quotient is understood in the derived sense. It is easy to see that for $m = 0$ we reproduce the perturbative algebra.

For theories with chiral multiplet, the analysis is similar. To simplify the notation we consider the case when the superpotential is zero. The phase space, as before, has $\mathbb{Z}$ disconnected components labeled by $\mathbb{C}^\times$ bundle $P_m$. For each component labeled by $P_m$, the tangent complex is given by

$$\mathrm{R}\Gamma(\mathbb{B}, P_m \times_{\mathbb{C}^\times} (T^*[1]V \oplus \mathfrak{gl}_1^* \oplus \mathfrak{gl}_1[1]))\,. \tag{3.28}$$

As an illustration, we first compute $\mathrm{R}\Gamma(\mathbb{B}, P_m \times_{\mathbb{C}^\times} V)$. The $\mathbb{C}^\times$ action on $V$ is no longer trivial. We decompose $V$ into weight space $V = \oplus_{w \in \mathbb{Z}} V_w$. Then $\mathrm{R}\Gamma(\mathbb{B}, P_m \times_{\mathbb{C}^\times} V)$ decomposes into

$$\oplus_w \mathrm{R}\Gamma(\mathbb{B}, P_m \times_{\mathbb{C}^\times} V_w)\,. \tag{3.29}$$

Each summand is computed by the following Čech complex

$$0 \to V_w[[z]] \oplus V_w[[z]] \xrightarrow{d_w} V_w((z)) \to 0\,, \tag{3.30}$$

where the differential $d$ is given by $d_w(f(z), g(z)) = f(z) - z^{wm} g(z)$ for any $(f(z), g(z)) \in V_w[[z]] \oplus V_w[[z]]$. We find that

$$H^\bullet(\mathbb{B}, P_n \times_{\mathbb{C}^\times} V_w) = \begin{cases} V_w[[z]]z^{wm} \oplus (V_w((z))/V_w[[z]])[-1], & wm \geq 0\,, \\ V_w[[z]] \oplus (V_w((z))/V_w[[z]]z^{wm})[-1], & wm < 0\,. \end{cases} \tag{3.31}$$

Similarly,

$$H^\bullet(\mathbb{B}, P_m \times_{\mathbb{C}^\times} V_w^*[1]) = \begin{cases} V_w^*[[z]][1] \oplus (V_w^*((z))/V_w^*[[z]]z^{-wm}), & wm \geq 0\,, \\ V_w^*[[z]][1]z^{-wm} \oplus (V_w^*((z))/V_w^*[[z]]), & wm < 0\,. \end{cases} \tag{3.32}$$

The symplectic pairing is given by the pairing between $H^0(\mathbb{B}, P_m \times_{\mathbb{C}^\times} V_w)$ and $H^0(\mathbb{B}, P_m \times_{\mathbb{C}^\times} V_w^*[1])$ and the pairing between $H^1(\mathbb{B}, P_m \times_{\mathbb{C}^\times} V_w)$ and $H^{-1}(\mathbb{B}, P_m \times_{\mathbb{C}^\times} V_w^*[1])$. Our previous experience suggests that we should choose the polarization to be the Lagrangian fibration over

$$\Gamma(\mathbb{B}, P_m \times_{\mathbb{C}^\times} T^*[1]V_w) = \begin{cases} V_w[[z]]z^{wm} \oplus V_w^*[[z]][1], & wm \geq 0\,, \\ V_w[[z]] \oplus V_w^*[[z]]z^{-wm}[1], & wm < 0\,. \end{cases} \tag{3.33}$$

Summing over weight, and combining with the gauge fields, we find that the local operator algebra is given by functions on the space

$$\prod_{m \in \mathbb{Z}} \Gamma(\mathbb{B}, P_m \times_{\mathbb{C}^\times} T^*[1]V \oplus \mathfrak{gl}_1^*) / J_\infty \mathbb{C}^\times \,. \tag{3.34}$$

This construction easily generalizes to the case of an arbitrary abelian gauge theory. For example, when the gauge group is $G = U(1)^r$, and we have chiral multiplet in a representation $V$ of $G$, then local operators can be constructed from the following space

$$\prod_{m \in \mathbb{Z}^r} \Gamma(\mathbb{B}, P_m \times_{\mathbb{C}^\times} T^*[1]V \oplus (\mathfrak{gl}_1^*)^r)/(J_\infty \mathbb{C}^\times)^r. \tag{3.35}$$

For non abelian gauge group, we expect that, for pure gauge theory, the bulk algebra can be given by derived $J_\infty G$ invariants of the WZW vacuum module $\mathrm{WZW}_k[\mathfrak{g}]$. However, it is not clear what the full algebra will look like for the most general cases.

## 3.3 Superconformal index and operator counting

An important tool in studying physical $3d$ $\mathcal{N} = 2$ theories is the superconformal index. It computes the following partition function for a theory defined on $S^2 \times S^1$ [17,47]

$$I(t_a; x) = \mathrm{Tr}\left((-1)^F e^{-\beta(E-R-j_3)} x^{E+j_3} \prod_a t_a^{F_a}\right). \tag{3.36}$$

Here $E$ is the energy, $R$ is the R-charge, $j_3$ is the third component of the angular momentum, and $F_a$'s run over the global flavor symmetry. Standard physical arguments tell us that only states with $E = R + j_3$ contribute to the index. Therefore we can simplify the index as

$$I(t_a; x) = \mathrm{Tr}\left((-1)^F q^{\frac{R}{2}+j_3} \prod_a t_a^{F_a}\right), \tag{3.37}$$

where $q = x^2$. This index is usually computed by localization technique [16,18].

In this paper, we study the holomorphic twisted version of physical $3d$ $\mathcal{N} = 2$ theories. For any such theory we consider its local operator algebra Obs. This algebra comes equipped with twisted spin grading $J$, fermionic grading $F$, and other gradings $F_a$ associated with some global symmetry. Then we can define the character

$$\chi(\mathrm{Obs}) = \mathrm{Tr}_{\mathrm{Obs}}\left((-1)^F q^J \prod_a t_a^{F_a}\right). \tag{3.38}$$

The twisting procedure identifies the twisted spin with $J = \frac{R}{2} + j_3$ and keeps other flavor symmetry unchanged. It turns out that this character defined for the holomorphic twisted theory coincides with the superconformal index of the physical theory. On the one hand, this provides us with an alternative method to compute the superconformal index by counting local operators. On the other hand, we can think of the space of local operators as providing a "categorification" of the index.

For example, for a free chiral multiplet, the space of local operators is $\mathbb{C}[\{\partial^n \phi, \partial^n \psi\}_{n \geq 0}]$. This theory has a flavor symmetry that rotates the fields $\phi$ and $\psi$. Suppose the spin and flavor charge of the fields are given as follows

|     | $\phi$ | $\psi$ |
| --- | --- | --- |
| $T$ | 1 | -1 |
| $J$ | 0 | 1 |

Then the operator $\partial^n \phi$ contributes $q^n x$ to the index and $\partial^n \psi$ contributes $q^{n+1} x^{-1}$ to the index. We can easily write down the index of a free chiral

$$\chi(\mathrm{Obs}_{\text{single chiral}}) = \frac{(qx^{-1}, q)_\infty}{(x, q)_\infty}, \tag{3.39}$$

where we used the $q$-Pochhammer symbols

$$(x; q)_\infty = \prod_{n \geq 0} (1 - q^n x). \tag{3.40}$$

### 3.3.1 Chern-Simons line bundle twist

In this section, we consider theory with a Chern-Simons term. In the presence of a Chern-Simons term, the Hilbert space is no longer the space functions on the polarized phase space, but the space of sections of a line bundle $\mathcal{L}^{\otimes k}$, with $k$ being the Chern-Simons coefficient. $\mathcal{L}$ is also called the determinant line bundle. Details of the construction of this line bundle for an arbitrary gauge group are given in [48,49]. Here, we present its construction for the abelian group $\mathbb{C}^{\times}$.

A $G$ bundle on the formal bubble $\mathbb{B}$ can be defined by a transition function on $\mathbb{D}$ module gauge transformations on the two copies of $\mathbb{D}$. Therefore we can regard $\mathrm{Map}(\mathbb{B}, BG)$ as a double quotient

$$G[[z]]\backslash G((z))/G[[z]] = G[[z]]\backslash \mathrm{Gr}_G. \tag{3.41}$$

The affine Grassmannian $\mathrm{Gr}_G$ describes $G$ bundles on the disk $\mathbb{D}$ trivialized on the punctured disk $\mathbb{D}^{\times}$.

Let $R = \mathbb{C}_{\mathrm{fund}}$ be the fundamental representation of $G = \mathbb{C}^{\times}$. For a point in $\mathrm{Map}(\mathbb{B}, BG)$ represented by a $P \in \mathrm{Gr}_{\mathbb{C}^{\times}}$, we get an associated bundle $R_P$ on $\mathbb{D}$. For $P$ equivalent to the $P_m$ that we introduced in 3.2.2, we can describe the space of sections of $R_P$ as

$$R[[z]]z^m, \quad \text{for } m \in \mathbb{Z}. \tag{3.42}$$

We define a map $F : \Gamma(\mathbb{D}, R_P) \to R[[z]]$ by the following composition of maps

$$\Gamma(\mathbb{D}, R_P) \hookrightarrow \Gamma(\mathbb{D}^{\times}, R_P) = R((z)) \to R[[z]]. \tag{3.43}$$

Then we define the fiber of the Chern-Simons line bundle at $P$ to be the following

$$\mathcal{L}_P = \det(\ker F) \otimes \det(\mathrm{coker} F)^{-1}. \tag{3.44}$$

We are interested in the character of this space under the action of gauge group $\mathbb{C}^{\times}$ and spin rotation $\mathbb{C}_q^{\times}$. Suppose $P = P_m$ and $m < 0$. We have $\Gamma(\mathbb{D}, R_P) = R[[z]]z^{-m}$. The map $F$ is injective and has cokernel

$$\mathrm{coker} F = \bigoplus_{0 \le i \le -m} Rz^i. \tag{3.45}$$

We can compute the character

$$\chi(\mathrm{coker} F) = \prod_{i=0}^{m-1} \chi(\det Rz^i) = \prod_{i=0}^{-m-1} sq^{-i} = s^m q^{-\frac{(m+1)m}{2}}. \tag{3.46}$$

For $m \ge 0$, the map $F : R[[z]]z^{-m} \to R[[z]]$ is surjective and has kernel

$$\ker F = \bigoplus_{-m \le i \le -1} Rz^i. \tag{3.47}$$

It has character

$$\chi(\ker F) = \prod_{i=-1}^{-m} sq^{-i} = s^{-m} q^{\frac{(m+1)m}{2}}. \tag{3.48}$$

We find that the character of $\mathcal{L}_P$ is

$$s^{-m} q^{\frac{(m+1)m}{2}} \quad \text{for } m \in \mathbb{Z}. \tag{3.49}$$

### 3.3.2  Berezinian and Half-form twist

The final step in geometric quantization is the so called half-form correction. Here we briefly explain the idea of it. Suppose that in the quantization problem, the polarization is given by a Lagrangian fibration over $X$. If we consider the space of polarized sections as the Hilbert space $\mathcal{H}$ of quantum states, we meet the problem of defining the inner product on $\mathcal{H}$. The inner product should be defined via integration, but the integration of polarized sections over the whole phase space diverges, and there is no natural choice of a measure on $X$. So we are unable to define an inner product on the space of polarized sections. To remedy this situation, we consider the following half-form bundle

$$(\det \Omega^1 X)^{\frac{1}{2}}. \tag{3.50}$$

Suppose $X$ has dimension $n$. Then $\det \Omega^1 X$ has the familiar form $\wedge^n \Omega^1 X = \Omega^n X$ as top degree differential form. We tensor the quantum line bundle with the half-form bundle. Then two sections can be paired and integrated over $X$. This gives us the notion of inner product on the Hilbert space.

In our situation, we get a supermanifold $\mathcal{X}$ after polarization. Differential forms of top degree no longer make sense on $\mathcal{X}$, as the one forms in the Grassmannian odd directions are Grassmannian even variables. The right definition of integration over a supermanifold $\mathcal{X}$ is provided by the Berezinian integration, where the integration map is defined on the space of Berezinian, which replaces the determinant of one form

$$\int : \operatorname{Ber} \Omega^1 \mathcal{X} \to \mathbb{C}. \tag{3.51}$$

We provide the definition and properties of the Berezinian in Appendix B.

The half-form bundle in our case becomes

$$\left( \operatorname{Ber} \Omega^1 \mathcal{X} \right)^{\frac{1}{2}}. \tag{3.52}$$

In the case of free chiral multiples, we get $J_\infty T^*[1]V$ after polarization. The above discussion suggests that we consider the following line bundle

$$\sqrt{\operatorname{Ber} \Omega^1 (J_\infty T^*[1]V)} = \sqrt{\operatorname{Ber}(T^*[1]V[[z]])^*} \otimes \mathcal{O}_{J_\infty T^*[1]V}. \tag{3.53}$$

The space $(J_\infty T^*[1]V)^*$ is infinite dimensional and its Berezinian is not well defined. Its character is not well defined as well. The power of gauge fugacity $s$ diverges if we try to compute the character. The superconformal index of a free chiral in physical computation also coincides with 3.39 without any correction, which suggests that we should ignore the half form correction in this case.

It becomes more interesting when the chiral multiplet is coupled with a $U(1)$ vector multiplet. In this case, we have $\Gamma(\mathbb{B}, P_m \times_{\mathbb{C}^\times} T^*[1]V)$ as the base of fibration for the monopole charge $m$ sector. Although it is again an infinite dimensional space, we get a finite and well defined object by dividing the infinite part we found before. Here, we consider the following "regularized" half form correction

$$L_h(m) = \left( \frac{\operatorname{Ber} \Gamma(\mathbb{B}, P_m \times_{\mathbb{C}^\times} T^*[1]V_w)^*}{\operatorname{Ber}(T^*[1]V[[z]])^*} \right)^{\frac{1}{2}} = \left( \frac{\operatorname{Ber} \Gamma(\mathbb{B}, P_m \times_{\mathbb{C}^\times} T^*[1]V_w)}{\operatorname{Ber}(T^*[1]V[[z]])} \right)^{-\frac{1}{2}}. \tag{3.54}$$

Using the properties of Berezinian that

$$\operatorname{Ber}(V_1 \oplus V_2) = \operatorname{Ber}(V_1) \otimes \operatorname{Ber}(V_2), \tag{3.55}$$

we can rewrite 3.54 as the Berezinian of a finite dimensional space. We substitute 3.33 into the expression and found that each weight space $V_w$ contributes to the half form bundle by the following

$$L_{h,w}(m) = \begin{cases} (\text{Ber}(\oplus_{i=0}^{wm-1} V_w z^i))^{\frac{1}{2}}, & wm \geq 0, \\ (\text{Ber}(\oplus_{i=0}^{-wm-1} V_w^*[1] z^i))^{\frac{1}{2}}, & wm < 0. \end{cases} \tag{3.56}$$

We have

$$L_h(m) = \bigotimes_w L_{h,w}(m). \tag{3.57}$$

For $wm \geq 0$, the space $V_w z^i$ has degree 0, so the Berezinian becomes the usual determinant. We can compute the character of the determinant by taking the product of the characters of a basis of it. We have

$$\chi(\text{Ber}(V_w z^i)) = s^{w \dim V_w} q^{-i \dim V_w}, \tag{3.58}$$

where $s$ is the gauge $\mathbb{C}^\times$ fugacity. For the character of the Berezinian of $V_w^*[1] z^i$, we consider the $\mathbb{C}^\times \times \mathbb{C}_q^\times$ action on $V_w^*[1] z^i$ which induces an action on the Berezinian. The action on $V_w^*[1] z^i$ is given by a diagonal matrix $\text{diag}(s^{-w} q^{-i-1}, s^{-w} q^{-i-1}, \ldots, s^{-w} q^{-i-1})$ that sits completely in degree 1. The Berezinian of this diagonal matrix is $(s^{-w} q^{-i-1})^{-\dim V_w}$, we get

$$\chi(\text{Ber}(V_w^*[1] z^i)) = (-1)^{\dim V_w} s^{w \dim V_w} q^{(i+1) \dim V_w}, \tag{3.59}$$

where we also include a $(-1)$ factor coming from the $\mathbb{Z}_2$ grading of the Berezinian B. We temporarily ignore the issue of possibly other flavor symmetry acting on $V$. We find that for $wm \geq 0$

$$\chi(L_{h,w}(m)) = \prod_{i=0}^{wm-1} \chi(\text{Ber}(V_w z^i)^{\frac{1}{2}}) \tag{3.60}$$
$$= s^{\frac{1}{2} mw^2 \dim V_w} q^{\frac{-wm(wm-1)}{4} \dim V_w},$$

and for $wm < 0$

$$\chi(L_{h,w}(m)) = \prod_{i=0}^{-wm-1} \chi(\text{Ber}(V_w^*[1] z^i)) \tag{3.61}$$
$$= (-1)^{mw \dim V_w} s^{-\frac{1}{2} mw^2 \dim V_w} q^{\frac{wm(wm-1)}{4} \dim V_w}.$$

We can get the character of the half form bundle by

$$\chi(L_h(m)) = \prod_w \chi(L_{h,w}(m)). \tag{3.62}$$

In summary, for theory with Chern-SImons level $k$, the space of local operators can be identified with sections of the line bundle $\mathcal{L}_h \otimes \mathcal{L}_{CS}^{\otimes k}$. In fact, this line bundle contains information about "one loop correction" in the original physical theory. For example, we can consider the gauge $\mathbb{C}^\times$ character of this bundle restricted to the monopole charge $m = \pm 1$ sectors

$$\chi_{\mathbb{C}^\times}(\mathcal{L}_h \otimes \mathcal{L}_{CS}^{\otimes k}(\pm 1)) = s^{\mp k + \sum_w \frac{1}{2} w|w| \dim V_w}. \tag{3.63}$$

This corresponds to the $\mathbb{C}^\times$ gauge charge of the two "bare" monopole operators, and physically comes from the one loop effective Chern-Simons level. This is compatible with results in physics literature [50]. Similarly, the quantum correction of the spin of the monopole operator is also contained in this construction.

Using the decomposition 3.33, we see that the local operator algebra can be decomposed into different monopole sectors

$$\mathrm{Obs} = \bigoplus_{m \in \mathbb{Z}} \mathrm{Obs}^m. \tag{3.64}$$

Each sector $\mathrm{Obs}^m$ consists of a family of dressed monopole operators computed by the cohomology of some complex. The "dressing" is provided by some polynomial of the fields, carrying a certain $\mathbb{C}^\times$ gauge charge that cancels the gauge charge of the line bundle 3.63 on that monopole component. We will provide some explicit examples in the next section.

# 4 Examples

In this section, we apply the general discussion above to some specific examples. We explicitly write down the chain complexes of the operator algebras. Unfortunately, we are unable to compute the whole cohomologies of these complexes. We only compute the cohomologies of operators of small spins as an illustration. On the other hand, the character can be easily computed by counting operators directly from the cochain complex.

## 4.1 XYZ model

We first consider the XYZ model, which is one of the most important examples in this paper. This theory has three chiral multiplets denoted $(\mathbf{X}, \mathbf{\Psi_X})$, $(\mathbf{Y}, \mathbf{\Psi_Y})$, $(\mathbf{Z}, \mathbf{\Psi_Z})$, and is equipped with a cubic superpotential

$$W = \mathbf{XYZ}. \tag{4.1}$$

Using the notation in Section 3.1, the operator algebra is generated by bosonic operators $\partial^n X, \partial^n Y, \partial^n Z$ for $n \geq 0$ and fermionic operators $\partial^n \psi_X, \partial^n \psi_Y, \partial^n \psi_Z$ for $n \geq 0$. We write down the complex of local operator algebra

$$\mathrm{Obs}_{XYZ} = \left( \mathbb{C}[\{\partial^n X, \partial^n Y, \partial^n Z, \partial^n \psi_X, \partial^n \psi_Y, \partial^n \psi_Z\}_{n \geq 0}], Q \right), \tag{4.2}$$

with the differential $Q$ given as follows[1]

$$
\begin{aligned}
Q\partial^n \psi_X &= \sum_{0 \leq l \leq n} \partial^{n-l} Y \partial^l Z, \\
Q\partial^n \psi_Y &= \sum_{0 \leq l \leq n} \partial^{n-l} Z \partial^l X, \\
Q\partial^n \psi_Z &= \sum_{0 \leq l \leq n} \partial^{n-l} X \partial^l Y.
\end{aligned}
\tag{4.3}
$$

The XYZ model has two flavor symmetry, denoted $A$ and $T$. The spin and flavor symmetry charges of various fields are given as follows

|     | $X$ | $Y$ | $Z$ | $\psi_X$ | $\psi_Y$ | $\psi_Z$ |
|-----|-----|-----|-----|----------|----------|----------|
| $A$ | 2   | $-1$ | $-1$ | $-2$    | 1        | 1        |
| $T$ | 0   | 1   | $-1$ | 0        | $-1$     | 1        |
| $J$ | 0   | $\frac{1}{2}$ | $\frac{1}{2}$ | 1 | $\frac{1}{2}$ | $\frac{1}{2}$ |

It's very hard to compute the cohomology of the whole complex. However, the complex is graded by spin $J$ and flavor symmetry $T$ and so does the cohomology. For each spin $J$ and $T$ charge, the cohomology can be computed by hand. Here, we list part of the cohomology of small spins.

---

[1] More precisely we have $Q\partial^n \psi_X = \sum_{0 \leq l \leq n} \binom{n}{l} \partial^{n-l} Y \partial^l Z$. However we can always get rid of this coefficient by a field redefinition $\partial^n \psi_X \to \frac{1}{n!} \partial^n \psi_X$ etc.



| | $J=0$ | $J=1$ | $J=2$ | ... |
|---|---|---|---|---|
| $T=0$ | $\{X^n\}_{n\geq0}$ | $\{X^n\partial X, X^n(X\psi_X -Y\psi_Y)\}_{n\geq0}$, $Y\psi_Y -Z\psi_Z$ | $\{X^n\partial^2 X, X^n(\partial X)^2,$ $X^n\partial(X\psi_X -Y\psi_Y),$ $X^n(\partial X\psi_X -Y\partial\psi_Y +\partial Z\psi_Z)\}_{n\geq0},$ $\partial YZ - Y\partial Z,\ \ \partial(Y\psi_Y - Z\psi_Z),$ $X\partial(Y\psi_Y -Z\psi_Z)$ | |
| $T=2$ | | $Y^2$ | $Y^2(Z\psi_Z -Y\psi_Y),\ Y\partial Y$ | |
| ... | | | | |

| | $J=\frac{1}{2}$ | $J=\frac{3}{2}$ | $J=\frac{5}{2}$ | ... |
|---|---|---|---|---|
| $T=1$ | $Y$ | $\partial XY,\qquad \partial Y,$ $Y(Y\psi_Y -Z\psi_Z)$ | $(\partial X)^2 Y, \partial^2 XY, Y\partial(Y\psi_Y -Z\psi_Z),$ $Y\partial(X\psi_X -Y\psi_Y),\ \partial^2 Y$ | |
| $T=-1$ | $Z$ | $\partial XZ,\qquad \partial Z,$ $Z(Y\psi_Y -Z\psi_Z)$ | $(\partial X)^2 Z, \partial^2 XZ, Z\partial(Y\psi_Y -Z\psi_Z),$ $Z\partial(X\psi_X -Y\psi_Z),\ \partial^2 Z$ | |
| ... | | | | |

Although the whole cohomology is hard to compute, its ghost number zero part is easy to find. It can be identified with functions on the infinite jet of the critical locus of $W$

$$\mathbb{C}[\{\partial^n X, \partial^n Y, \partial^n Z\}_{n\geq0}]/\left\langle\{\partial^n(XY), \partial^n(YZ), \partial^n(ZX)\}_{n\geq0}\right\rangle. \tag{4.4}$$

Similarly, for chiral multiplets with an arbitrary superpotential, the ghost number zero part of the cohomology is

$$\mathbb{C}[J_\infty\mathrm{Crit}(W)] = \mathbb{C}[\{\partial^n\phi_i\}_{n\geq0,i}]/\left\langle\{\partial^n\left(\frac{\delta W}{\delta\phi_i}\right)\}_{n\geq0,i}\right\rangle. \tag{4.5}$$

The character of the XYZ model can be computed as if there is no superpotential, and is simply the product of the characters of the three chiral multiplets. We have

$$\chi(\mathrm{Obs}_{XYZ}) = \frac{(a^{-2}q,q)_\infty}{(a^2,q)_\infty}\frac{(ax^{-1}q^{\frac{1}{2}},q)_\infty}{(a^{-1}xq^{\frac{1}{2}},q)_\infty}\frac{(axq^{\frac{1}{2}},q)_\infty}{(a^{-1}x^{-1}q^{\frac{1}{2}},q)_\infty}. \tag{4.6}$$

## 4.2 $U(1)$ pure gauge theory

As our first example of gauge theory, we consider pure abelian $G = \mathbb{C}^\times$ vector multiplet without matter. The perturbative algebra is very simple. We have

$$\mathrm{Obs}_{U(1)}^{\mathrm{per}} = \mathbb{C}\left[J_\infty\mathfrak{gl}_1/J_\infty\mathbb{C}^\times\right]. \tag{4.7}$$

As the adjoint action of $\mathbb{C}^\times$ on $\mathfrak{gl}_1$ is trivial, all operators are $\mathbb{C}^\times$ invariant. So the operator algebra consists of the field $b$ and all its derivatives and derivatives of the field $c$. Explicitly this gives us

$$\mathrm{Obs}_{U(1)}^{\mathrm{per}} = \mathbb{C}\left[\{\partial^{n+1}c, \partial^n b\}_{n\geq0}\right]. \tag{4.8}$$

Non perturbative algebra is analyzed in Section 3.2.2. The phase space has connected components labeled by $\pi_1(\mathbb{C}^*) = \mathbb{Z}$. For each connected component, we see from 3.33 that the operator algebra is a copy of $\mathrm{Obs}^{\mathrm{per}}$. Therefore we get

$$\mathrm{Obs}_{U(1)} = \bigoplus_{m\in\mathbb{Z}}\mathbb{C}\left[\{\partial^{n+1}c, \partial^n b\}_{n\geq0}\right] = \mathbb{C}\left[v, v^{-1}, \{\partial^{n+1}c, \partial^n b\}_{n\geq0}\right]. \tag{4.9}$$

We can also write it as $\mathrm{Obs}_{U(1)} = \mathbb{C}[J_\infty T^*[1]\mathbb{C}^\times]$, which is the same as the bulk algebra of a free chiral valued in $\mathbb{C}^\times$.

The character of this theory is trivial because $\partial^{n+1}c$ and $\partial^n b$ contribute $-q^{n+1}$ and $q^{n+1}$ to the character respectively, and they cancel with each other.

### 4.3   $N_f = \tilde{N}_f = 1$ **SQED**

The $\mathcal{N} = 2$ SQED has one $G = \mathbb{C}^\times$ vector multiplet $(\mathbf{B}, \mathbf{A})$ and two chiral multiplets $(\mathbf{\Phi}_+, \mathbf{\Psi}_+)$, $(\mathbf{\Phi}_-, \mathbf{\Psi}_-)$ with gauge charge $+1, -1$, respectively. There is no superpotential and Chern Simons term. The global symmetry group of SQED comprises a $U(1)_A$ axial symmetry, $U(1)_T$ topological symmetry, and the usual $U(1)_R$ symmetry. The charges of various fields under gauge symmetry, flavor symmetries and spin are

|       | $\phi_+$ | $\phi_-$ | $\psi_+$ | $\psi_-$ | $c$ | $b$ |
|-------|:--------:|:--------:|:--------:|:--------:|:---:|:---:|
| Gauge | 1        | $-1$     | $-1$     | 1        | 0   | 0   |
| $A$   | 1        | 1        | $-1$     | $-1$     | 0   | 0   |
| $T$   | 0        | 0        | 0        | 0        | 0   | 0   |
| $J$   | 0        | 0        | 1        | 1        | 0   | 0   |

It's very easy to write down the local operator algebra in the perturbative sector. We have

$$\text{Obs}_{\text{SQED}}^{\text{per}} = \left( \mathbb{C}[\{\partial^{n+1}c, \partial^n b, \partial^n \phi_\pm, \partial^n \psi_\pm\}_{n \geq 0}]^{\mathbb{C}^\times}, Q \right), \tag{4.10}$$

where the differential $Q$ is given by

$$
\begin{aligned}
Q\partial^n b &= \sum_{0 \leq l \leq n} \partial^l \phi_+ \partial^{n-l} \psi_+ - \partial^l \phi_- \partial^{n-l} \psi_-, \\
Q\partial^n \phi_\pm &= \pm \sum_{1 \leq l \leq n} \partial^l c \, \partial^{n-l} \phi_\pm, \\
Q\partial^n \psi_\pm &= \mp \sum_{1 \leq l \leq n} \partial^l c \, \partial^{n-l} \psi_\pm.
\end{aligned}
\tag{4.11}
$$

As before, we decompose the cohomology by spin charges and do the computation of cohomology for small spins

|         | $J = 0$ | $J = 1$ | $J = 2$ | ... |
|---------|---------|---------|---------|-----|
| $T = 0$ | $\{(\phi_+\phi_-)^n\}_{n\geq 0}$ | $\{(\phi_+\phi_-)^n\partial(\phi_+\phi_-),$ $(\phi_+\phi_-)^n(\phi_+\psi_+$ $+\phi_-\psi_-)\}_{n\geq 0}, \partial c$ | $\{(\phi_+\phi_-)^n\partial^2(\phi_+\phi_-),$ $(\phi_+\phi_-)^n(\partial(\phi_+\phi_-))^2,$ $(\phi_+\phi_-)^n(\partial(\phi_+\psi_+ + \phi_-\psi_-)),$ $(\phi_+\phi_-)^n(b\partial c \ + \ \partial\phi_+\psi_+ - \phi_-\partial\psi_-)\}_{n\geq 0}, \ \psi_+\psi_-, \ \partial^2 c,$ $\phi_+\phi_-\partial^2 c$ | |

As we have discussed in Section 3.2.2, for operators at a non zero monopole sector $T = m$, we need to consider the following space

$$\Gamma\left(\mathbb{B}, P_m \times_{\mathbb{C}^\times} (T^*[1]V)\right). \tag{4.12}$$

Here $V = \mathbb{C}^2$ and can be decomposed into $\mathbb{C}_+ \oplus \mathbb{C}_-$ on which the $\mathbb{C}^\times$ weights are $+1, -1$ respectively. Note that operators are functions on the space of fields. So operators from functions on $\mathbb{C}_+[[z]]$ have $-1$ gauge charge and operators from functions on $\mathbb{C}_-[[z]]$ have $+1$ gauge charge. The computation of 4.12 is already explained in 3.2.2 and we find that

$$\Gamma\left(\mathbb{B}, P_m \times_{\mathbb{C}^\times} (T^*[1]V)\right) = \begin{cases} z^m\mathbb{C}[[z]] \oplus \mathbb{C}[[z]] \oplus \mathbb{C}[[z]][1] \oplus z^m\mathbb{C}[[z]][1], & m \geq 0, \\ \mathbb{C}[[z]] \oplus z^{-m}\mathbb{C}[[z]] \oplus z^{-m}\mathbb{C}[[z]][1] \oplus \mathbb{C}[[z]][1], & m < 0. \end{cases} \tag{4.13}$$

In summary, the local operator algebra at monopole charge $m$ sector is given by[2]

$$
\text{Obs}_{\text{SQED}}^m = \begin{cases} (\mathbb{C}[\{\partial^{n+1}c, \partial^n b, \partial^n \phi_+, \partial^{m+n}\phi_-, \partial^{-m+n}\psi_+, \partial^n \psi_-\}_{n \geq 0}]^{\mathbb{C}^\times}, Q), & \text{for } m \geq 0, \\ (\mathbb{C}[\{\partial^{n+1}c, \partial^n b, \partial^{-m+n}\phi_+, \partial^n \phi_-, \partial^n \psi_+, \partial^{-m+n}\psi_-\}_{n \geq 0}]^{\mathbb{C}^\times}, Q), & \text{for } m < 0. \end{cases} \tag{4.14}
$$

The differential $Q$ is obtained from the differential 4.11 by discarding all terms not in the complex. The full local operator algebra is

$$
\text{Obs}_{\text{SQED}} = \bigoplus_{m \in \mathbb{Z}} \text{Obs}_{\text{SQED}}^m. \tag{4.15}
$$

We give some examples of operators of non-zero monopole charges.

|  | $J = \frac{1}{2}$ | $J = \frac{3}{2}$ | $J = \frac{5}{2}$ | ... |
|---|---|---|---|---|
| $T = 1$ | $v_+$ | $v_+ \phi_+ \partial \phi_-$, $v_+ \partial c$, $v_+ b$ | $v_+(\phi_+ \partial \phi_-)^2$, $v_+ \partial(\phi_+ \partial \phi_-)$, $v_+(\phi_+ \partial \psi_+ - \partial \phi_- \psi_-)$, $v_+ b^2$, $v_+ \partial^2 c$ |  |
| $T = -1$ | $v_-$ | $v_- \partial \phi_+ \phi_-$, $v_- \partial c$, $v_- b$ | $v_-(\partial \phi_+ \phi_-)^2$, $v_- \partial(\partial \phi_+ \phi_-)$, $v_-(\partial \phi_+ \psi_+ - \phi_- \partial \psi_-)$, $v_+ b^2$, $v_+ \partial^2 c$ |  |

To compute the character for each monopole sector, we need to take the half form correction into account. The gauge and spin fugacities are already considered in 3.3.2, so we only need to consider the flavor $\mathbb{C}_A^\times$ fugacity here. When $m \geq 0$ we have

$$
L_h(m) = \left(\text{Ber}(\oplus_{i=0}^{m-1} \mathbb{C}_+ z^i)\right)^{\frac{1}{2}} \otimes \left(\text{Ber}(\oplus_{i=0}^{m-1} \mathbb{C}_-^*[1]z^i)\right)^{\frac{1}{2}}. \tag{4.16}
$$

$\mathbb{C}_+$ has $A$ charge $-1$ and $\mathbb{C}_-^*$ has $A$ charge $+1$. Combining with spin and gauge fugacity 3.3.2 gives us the character

$$
\begin{aligned}
\chi(L_h(m)) &= (-1)^m a^{-\frac{1}{2}m} s^{\frac{1}{2}m} q^{\frac{-m(m-1)}{4}} a^{-\frac{1}{2}m} s^{-\frac{1}{2}m} q^{\frac{m(m+1)}{4}} \\
&= (-1)^m a^{-m} q^{\frac{1}{2}m}.
\end{aligned} \tag{4.17}
$$

For $m < 0$ we have

$$
L_h(m) = (\text{Ber}(\oplus_{i=0}^{-m-1} \mathbb{C}_- z^i))^{\frac{1}{2}} \otimes (\text{Ber}(\oplus_{i=0}^{-m-1} \mathbb{C}_+^*[1]z^i))^{\frac{1}{2}}. \tag{4.18}
$$

$\mathbb{C}_-$ has $A$ charge $-1$ and $\mathbb{C}_+^*$ has $A$ charge $+1$. The character is

$$
\begin{aligned}
\chi(L_h(m)) &= (-1)^m a^{\frac{1}{2}m} s^{-\frac{1}{2}m} q^{\frac{m(m-1)}{4}} a^{\frac{1}{2}m} s^{\frac{1}{2}m} q^{\frac{-m(m+1)}{4}} \\
&= (-1)^m a^m q^{-\frac{1}{2}m}.
\end{aligned} \tag{4.19}
$$

Together we have

$$
\chi(L_h(m)) = (-1)^m a^{-|m|} q^{\frac{|m|}{2}}. \tag{4.20}
$$

We can now compute the full character of the algebra. Contributions from $\partial^{n+1}c$ and $\partial^n b$ cancel. $\partial^n \phi_+$ contributes $saq^n$, $\partial^n \phi_-$ contributes $s^{-1}aq^n$, $\partial^n \psi_+$ contributes $-s^{-1}a^{-1}q^{n+1}$ and $\partial^n \psi_-$ contributes $-sa^{-1}q^{n+1}$. Therefore, for $m \geq 0$ we have

$$
\chi(\text{Obs}_{\text{SQED}}^m) = (-1)^m a^{-|m|} q^{\frac{|m|}{2}} \int \frac{ds}{2\pi i s} \frac{(sa^{-1}q; q)_\infty (s^{-1}a^{-1}q^{1+m}; q)_\infty}{(sa; q)_\infty (s^{-1}aq^m; q)_\infty}. \tag{4.21}
$$

---

[2]More precisely, the local operator algebra at monopole charge $m$ sector should be written as $\left(L(m)[\{\partial^{n+1}c, \dots\}_{n \geq 0}]\right)^{\mathbb{C}^\times}$, where $L(m) \cong \mathbb{C}$ correspond to the line bundle of 3.3.1,3.3.2 and carry both $\mathbb{C}^\times$ gauge charge and spin charge. We will keep using the notation 4.14 in this paper, but keep in mind the various quantum number of $L(m)$.

Similarly for $m < 0$ we have

$$\chi(\mathrm{Obs}^m_{\mathrm{SQED}}) = (-1)^m a^{-|m|} q^{\frac{|m|}{2}} \int \frac{ds}{2\pi i s} \frac{(sa^{-1}q^{1-m};q)_\infty (s^{-1}a^{-1}q;q)_\infty}{(saq^{-m};q)_\infty (s^{-1}a;q)_\infty}. \tag{4.22}$$

We can simplify our expression by making a change of the integration variable $s \to sq^{\frac{m}{2}}$. This gives us

$$\chi(\mathrm{Obs}^m_{\mathrm{SQED}}) = (-1)^m a^{-|m|} q^{\frac{|m|}{2}} \int \frac{ds}{2\pi i s} \frac{(sa^{-1}q^{1+\frac{|m|}{2}};q)_\infty (s^{-1}a^{-1}q^{1+\frac{|m|}{2}};q)_\infty}{(saq^{\frac{|m|}{2}};q)_\infty (s^{-1}aq^{\frac{|m|}{2}};q)_\infty}. \tag{4.23}$$

The index of the full local operator algebra is the sum

$$\chi(\mathrm{Obs}_{\mathrm{SQED}}) = \sum_{m \in \mathbb{Z}} x^m \chi(\mathrm{Obs}^m_{\mathrm{SQED}}) = \sum_{m \in \mathbb{Z}} x^m a^{-|m|} q^{\frac{|m|}{2}} \int \frac{ds}{2\pi i s} \frac{(s^\pm a^{-1}q^{1+\frac{|m|}{2}};q)_\infty}{(s^\pm aq^{\frac{|m|}{2}};q)_\infty}. \tag{4.24}$$

Here $x$ is the fugacity of topological symmetry and we absorbed the phase factor $(-1)^m$ into $x^m$.

## 4.4  $U(1)_{-\frac{1}{2}}$ + a chiral

In this section, we consider a $G = \mathbb{C}^\times$ vector multiplet $(\mathbf{A}, \mathbf{B})$ at Chern Simons level $-\frac{1}{2}$ coupled to a chiral multiplet $(\mathbf{\Phi}, \mathbf{\Xi})$ of charge 1 under the gauge group.

For the operator algebra at the monopole zero sector, we get the dg algebra

$$\left( \mathbb{C}[\{\partial^n \varphi, \partial^n \xi, \partial^n b, \partial^{n+1} c\}_{n \geq 0}]^{\mathbb{C}^\times}, Q \right), \tag{4.25}$$

with differential $Q$ given by

$$\begin{aligned}
Q\partial^n \varphi &= \sum_{1 \leq l \leq n} \partial^m c \, \partial^{n-l} \varphi, \\
Q\partial^n \xi &= -\sum_{1 \leq l \leq n} \partial^l c \, \partial^{n-l}, \\
Q\partial^n b &= \sum_{0 \leq l \leq n} \partial^l \varphi \, \partial^{n-l} \xi.
\end{aligned} \tag{4.26}$$

We list the cohomology for small spins

|  | $J = 1$ | $J = 2$ | $J = 3$ | ... |
|---|---|---|---|---|
| $T = 0$ | $\partial c$ | $\partial^2 c,\ \xi\partial\varphi + b\partial c$ | $\partial^3 c,\ \partial c\partial^2 c,\ b\xi\partial\varphi + \frac{1}{2}b^2\partial c,$ $\xi\partial^2\varphi - \varphi\partial^2\xi + 2b\partial^2 c + \partial c\partial b$ | |

For monopole charge $m > 0$, the operator algebras can be computed by the following complexes

$$\mathrm{Obs}^m = (\mathbb{C}[\{\partial^n \varphi, \partial^{m+n}\xi, \partial^n b, \partial^{n+1}c\}_{n \geq 0}]^{\mathbb{C}^\times}, Q), \quad \text{for } m > 0.$$

The differential is the truncation of 4.26. As before, we give some examples of operators here

|  | $J = 0$ | $J = 1$ | $J = 2$ | ... |
|---|---|---|---|---|
| $T = 1$ | $v_+$ | $v_+\partial c,\ v_+ b$ | $v_+ b\partial c,\ v_+ b^2,\ v_+\partial^2 c$ | |
| $T = 2$ | $v_+^2$ | $v_+^2\partial c,\ v_+^2 b$ | $v_+^2 b\partial c,\ v_+^2 b^2,\ v_+^2\partial^2 c,\ v_+^2\partial b$ | |

According to 3.3.1 3.3.2, the half form correction and the Chern-Simons term contribute a factor

$$s^{-\frac{1}{2}m}q^{\frac{m(m+1)}{4}}\left(s^{-m}q^{\frac{(m+1)m}{2}}\right)^{-\frac{1}{2}}=1\,.\qquad(4.27)$$

Then we have the character

$$\chi(\mathrm{Obs}^m)=\int\frac{ds}{2\pi is}\frac{(z^{-1}q;q)_\infty}{(sq^m;q)_\infty}\,,\quad m\geq0\,.\qquad(4.28)$$

By making change of the integration variable, $s\to sq^{\frac{m}{2}}$, we get

$$\chi(\mathrm{Obs}^m)=\int\frac{ds}{2\pi is}\frac{(s^{-1}q^{1+\frac{m}{2}};q)_\infty}{(sq^{\frac{m}{2}};q)_\infty}\,,\quad m\geq0\,.\qquad(4.29)$$

For the monopole charge $m$ sector with $m<0$, the half form and line bundle twist contribute a factor

$$(-1)^m s^{\frac{1}{2}m}q^{\frac{-m(m+1)}{4}}\left(s^{-m}q^{-\frac{(m+1)m}{2}}\right)^{-\frac{1}{2}}=(-s)^m q^{\frac{-m(m+1)}{2}}\,.\qquad(4.30)$$

After the change of variable $s\to sq^{\frac{m}{2}}$ as we have done before, we find that the character for $m<0$ is given by

$$\chi(\mathrm{Obs}^m)=\int\frac{ds}{2\pi is}(-s)^m q^{\frac{-m}{2}}\frac{(s^{-1}q^{1-\frac{m}{2}};q)_\infty}{(sq^{-\frac{m}{2}};q)_\infty}\,,\quad m<0\,.\qquad(4.31)$$

We can summarize the expression for $m\geq0$ and $m<0$ into one expression

$$\chi(\mathrm{Obs})=\sum_{m\in\mathbb{Z}}x^m\int\frac{ds}{2\pi is}(-q^{\frac{1}{2}})^{-\frac{1}{2}(m-|m|)}s^{\frac{1}{2}(m-|m|)}\frac{(s^{-1}q^{1+\frac{|m|}{2}};q)_\infty}{(sq^{\frac{|m|}{2}};q)_\infty}\,.\qquad(4.32)$$

It is interesting to note that the character here contains a relative phase factor $(-1)^{-\frac{1}{2}(m-|m|)}$ that only contributes when $m<0$. This phase factor cannot be derived from the usual localization computation of the index. However, it is predicted in [19] to incorporate various mirror dualities for the index. In our setup, this phase factor naturally comes from the fermionic grading of the Berezinian.

A closely related theory is the so called $U(1)_{\frac{1}{2}}+$ chiral theory, where we have a $G=\mathbb{C}^\times$ vector multiplet at Chern Simons level $\frac{1}{2}$ coupled to a chiral multiplet of charge 1 under the gauge group. This theory, together with the $U(1)_{-\frac{1}{2}}+$ chiral theory and the free chiral theory, consist of the simplest example of 3d $\mathcal{N}=2$ mirror "triality" [23].

$$\text{free chiral}\leftrightarrow U(1)_{\frac{1}{2}}+\text{ chiral }\leftrightarrow U(1)_{-\frac{1}{2}}+\text{ chiral}\,.\qquad(4.33)$$

## 4.5 Bulk algebras and dualities

In this section, we consider local operator algebras of different theories related by infrared dualities. It was argued in [12] that the $Q$ cohomology of bulk local operator algebra is constant along the RG flow. Therefore for two IR dual theories $\mathcal{T},\mathcal{T}'$ (two theories that flow to the same IR, also known as mirror dual theories [51, 52]), the corresponding local operator algebras $\mathrm{Obs}_\mathcal{T},\mathrm{Obs}_{\mathcal{T}'}$ should have the same $Q$ cohomology.

$$H^\bullet(\mathrm{Obs}_\mathcal{T},Q_\mathcal{T})\approx H^\bullet(\mathrm{Obs}_{\mathcal{T}'},Q_{\mathcal{T}'})\,.\qquad(4.34)$$

The mirror map always mixes the monopole operators and perturbative operators on the two sides. We should be able to observe this in cohomology. An even stronger statement is that

the two complexes $(\mathrm{Obs}_{\mathcal{T}}, Q_{\mathcal{T}}), (\mathrm{Obs}_{\mathcal{T}'}, Q_{\mathcal{T}'})$ are quasi-isomorphic. If true, this will provide us with a very strong and non-trivial test of $3d$ mirror symmetry. Although we are unable to verify this statement at the moment, we provide some evidence by comparing the cohomology of operators of small spins for mirror dual theories.

We have computed some operators for the $U(1)_{-\frac{1}{2}}$ + chiral theory. From what we have found in Section 4.4, we can observe an exact match of cohomologies of local operators for the two theories. Some examples are listed below

| $T = 0$ | $J = 1$ | $J = 2$ | | $J = 3$ | | ... |
|---|---|---|---|---|---|---|
| free chiral | $\phi\psi$ | $\partial(\phi\psi)$ | $\phi\partial\psi - \partial\phi\psi$ | $\partial^2(\phi\psi)$ | $\phi^2\psi\partial\psi$ | |
| $U(1)_{-\frac{1}{2}}$+ chiral | $\partial c$ | $\partial^2 c$ | $\xi\partial\varphi + b\partial c$ | $\partial^3 c$ | $\partial c\partial^2 c$ | ... |

| $T = 0$ | $J = 3$ | | ... |
|---|---|---|---|
| free chiral | $\partial\phi\partial\psi$ | $\partial^2\phi\psi - \phi\partial^2\psi$ | |
| $U(1)_{-\frac{1}{2}}$+ chiral | $b\xi\partial\varphi + \frac{1}{2}b^2\partial c$ | $\xi\partial^2\varphi - \varphi\partial^2\xi + 2b\partial^2 c + \partial c\partial b$ | ... |

| $T = 1$ | $J = 0$ | $J = 1$ | | $J = 2$ | | ... |
|---|---|---|---|---|---|---|
| free chiral | $\phi$ | $\phi^2\psi$ | $\partial\phi$ | $\phi\partial\phi\psi$ | $\partial^2\phi$ | $\phi\partial(\phi\psi)$ |
| $U(1)_{-\frac{1}{2}}$+ chiral | $v_+$ | $v_+\partial c$ | $v_+ b$ | $v_+ b\partial c$ | $v_+ b^2$, | $v_+\partial^2 c$ | ... |

The mirror duality indeed maps monopole operators on one side to perturbative operators on the other side. The identification $\phi \longleftrightarrow v_+$ is compatible with the studies of the corresponding physical theories [23].

Another classical example is given by the duality between SQED and XYZ model [52]. By comparing the cohomology of local operators with different quantum numbers, we can find an identification of operators under the duality. For the $T$ charge 0 sector, we have

| $T = 0$ | $J = 1$ | $J = 2$ | | | ... |
|---|---|---|---|---|---|
| XYZ | $X^n$ | $X^n\partial X$ | $X^n(X\psi_X - Y\psi_Y)$ | $Y\psi_Y - Z\psi_Z$ | |
| SQED | $(\phi_+\phi_-)^n$ | $(\phi_+\phi_-)^n\partial(\phi_+\phi_-)$ | $(\phi_+\phi_-)^n(\phi_+\psi_+ + \phi_-\psi_-)$ | $\partial c$ | ... |

| $T = 0$ | $J = 3$ | |
|---|---|---|
| | XYZ | SQED |
| $A = -2$ | $\partial Y Z - Y\partial Z$ | $\psi_+\psi_-$ |
| $A = 0$ | $\partial(Y\psi_Y - Z\psi_Z)$ | $\partial^2 c$ |
| $A = 2$ | $X\partial(Y\psi_Y - Z\psi_Z)$ | $(\phi_+\phi_-)\partial^2 c$ |
| $A = 2n$ | $X^n\partial(X\psi_X - Y\psi_Y)$ | $(\phi_+\phi_-)^n\partial(\phi_+\psi_+ - \phi_-\psi_-)$ |
| | $X^n(\partial X\psi_X - Y\partial\psi_Y + \partial Z\psi_Z)$ | $(\phi_+\phi_-)^n(b\partial c + \partial\phi_+\psi_+ - \partial\phi_-\psi_-)$ |
| $A = 2n + 2$ | $X^n\partial^2 X$ | $(\phi_+\phi_-)^n\partial^2(\phi_+\phi_-)$ |
| $A = 2n + 4$ | $X^n(\partial X)^2$ | $(\phi_+\phi_-)^n(\partial(\phi_+\phi_-))^2$ |

For the nonzero $T$ charge sectors, we also find the following identifications

| $T = 1$ | $J = \frac{1}{2}$ | $J = \frac{3}{2}$ | | | $J = \frac{5}{2}$ | | ... |
|---|---|---|---|---|---|---|---|
| XYZ | $Y$ | $\partial Y$ | $\partial X Y$ | $Y(Y\psi_Y - Z\psi_Z)$ | $\partial^2 Y$ | $(\partial X)^2 Y$ | |
| SQED | $v_+$ | $v_+ b$ | $v_+\phi_+\partial\phi_-$ | $v_+\partial c$ | $v_+ b^2$ | $v_+(\phi_+\partial\phi_-)^2$ | ... |

| $T = 1$ | $J = \frac{5}{2}$ | | | ... |
|---|---|---|---|---|
| XYZ | $\partial^2 X Y$ | $Y\partial(X\psi_X - Y\psi_Y)$ | $Y\partial(Y\psi_Y - Z\psi_Z)$ | |
| SQED | $v_+\partial(\phi_+\partial\phi_-)$ | $v_+(\phi_+\partial\psi_+ - \partial\phi_-\psi_-)$ | $v_+\partial^2 c$ | ... |

With the above identification of operators for the two theories, we can study how the Poisson bracket behaves under the mirror map. For example, in the XYZ model, we have the following bracket

$$\{X, Y\partial Z - Z\partial Y\} = 0. \tag{4.35}$$

The corresponding bracket in the SQED is

$$\{\phi_+\phi_-, \psi_+\psi_-\} = \phi_-\psi_- - \phi_+\psi_+ = -Qb. \tag{4.36}$$

This observation suggests that we expect the Poisson bracket to be preserved under the isomorphism only at the level of cohomology. If we find a quasi-isomorphism between the two complexes $(\text{Obs}_{\text{XYZ}}, Q)$ and $(\text{Obs}_{\text{SQED}}, Q)$, it will not preserve the Poisson bracket (or $\lambda$-bracket) at the cochain level. This inconsistency can be resolved by realizing that the bulk algebra actually has a higher analog of chiral algebra structure at the chain level. We will discuss more of this structure in Section 6.4. The upshot is that the quasi-isomorphism between the two complexes should be accompanied by an infinite series of higher maps between them so that the full algebraic structure of the bulk algebras is preserved. Finding all the "higher" morphisms between the algebras will be interesting.

# 5 Review of boundary conditions and boundary algebras

Half-BPS boundary conditions of physical 3d $\mathcal{N} = 2$ theory preserving 2d $\mathcal{N} = (0, 2)$ supersymmetry [53–56] are automatically compatible with the holomorphic twist, therefore leading to boundary conditions of twisted theory [12]. Here we briefly review some of the results.

There are four basic classes of boundary conditions for the vector and chiral multiplet.

- Dirichlet boundary condition for the chiral multiplet, abbreviated D, by imposing $\mathbf{\Phi} = 0$ on the boundary.

- Neumann boundary condition for the chiral multiplet, abbreviated N, by imposing $\mathbf{\Psi} = 0$ on the boundary.

- Dirichlet boundary condition for the vector multiplet, abbreviated $\mathcal{D}$, by imposing $\mathbf{A} = 0$ on the boundary.

- Neumann boundary condition for the vector multiplet, abbreviated $\mathcal{N}$, by imposing $\mathbf{B} = 0$ on the boundary.

There could be more complicated boundary conditions, but in this paper, we only study combinations of the above four.

## 5.1 Boundary algebras for chiral multiplets

We consider $n$ chiral multiplet with a superpotential $W$. Suppose the boundary condition is chosen such that the first $k$ chiral multiplets are given Neumann boundary condition, and the last $n-k$ chiral multiplets are given Dirichlet boundary condition. Namely, we set

$$\mathbf{\Psi}_i = 0, \ i = 1, \dots, k, \qquad \mathbf{\Phi}_i = 0, \ i = k+1, \dots, n \tag{5.1}$$

at the boundary.

By passing to cohomology, the boundary algebra is generated by bosonic operators $\{\phi_i(z), i = 1, \dots, k\}$ and fermionic operatos $\{\psi_j(z), j = k+1 \dots, n\}$. There are two situations to be discussed depending on the superpotential. First, if the superpotential vanishes at the boundary

$$W|_\partial = 0\,, \tag{5.2}$$

then the boundary chiral algebra is specified by OPE

$$\psi_j \psi_l = \frac{\hbar}{z} \partial_j \partial_l W|_\partial\,, \tag{5.3}$$

and BRST differential

$$Q\psi_j = \partial_j W|_\partial\,. \tag{5.4}$$

Another possibility is that the superpotential does not vanish at the boundary. In this case, this boundary condition is anomalous due to the superpotential. We can easily see this by considering the classical master equation:

$$\{S_{BV}, S_{BV}\} = 2\left\{\int \langle \Psi^i, \hat{d}\Phi_i \rangle, \int W(\Phi)\right\} = 2\int \hat{d}W(\Phi) = -2\int_\Sigma W\left(\Phi^\partial\right)\,. \tag{5.5}$$

This term measures the failure of the classical master equation to hold in the presence of the boundary, and we see that it is proportional to $W|_\partial$. Here, we can add boundary fermion to cancel this anomaly. We introduce $2d$ boundary fermion in the BV formalism, and we use superfield notation compatible with the $3d$ theories. We have fields

$$\Gamma_\alpha \in \Omega^{k_\alpha, \bullet}(\Sigma),\ \ \Gamma^\alpha \in \Omega^{1-k_\alpha, \bullet}(\Sigma)\,. \tag{5.6}$$

They are equipped with the following BV bracket

$$\{\Gamma_\alpha, \Gamma^\beta\} = \delta_{\alpha\beta}\delta(x-y)d\text{Vol}\,. \tag{5.7}$$

The BV action functional is given by

$$S = \sum_\alpha \int_\Sigma \Gamma_\alpha \wedge \bar{\partial}\Gamma^\alpha\,. \tag{5.8}$$

This BV action gives $\bar{\partial}$ as the BRST operator. Now suppose that we have polynomials $E_\alpha(\Phi^\partial), J^\alpha(\Phi^\partial)$, which are of charge $k_\alpha, 1-k_\alpha$ respectively under the $U(1)_R$ symmetry and satisfy:

$$E_\alpha(\Phi^\partial)J^\alpha(\Phi^\partial) = W(\Phi^\partial)\,. \tag{5.9}$$

Note that we can always find such a factorization of superpotential by adding enough boundary fermion. We can add the following boundary term to the Lagrangian:

$$S^\partial = \sum_\alpha \int_\Sigma \Gamma_\alpha \wedge \bar{\partial}\Gamma^\alpha + \Gamma^\alpha E_\alpha(\Phi^\partial) + \Gamma_\alpha J^\alpha(\Phi^\partial)\,. \tag{5.10}$$

This boundary term contains the BV action of the $2d$ fermion and the coupling terms on the boundary. Then we can show that the combined system $S^{\text{bulk}} + S^\partial$ satisfies the classical master equation.

For the boundary chiral algebra, we now have the new fields $(\Gamma_\alpha, \Gamma^\alpha)$. Since the cohomology of the complex $(\Omega^{k,\bullet}, \bar{\partial})$ consists of holomorphic $k$-forms, we only need to consider the bottom component of the fields just like what we have done in the $3d$ case. Therefore the boundary algebras have new operators generated by $\Gamma_\alpha(z), \Gamma^\alpha(z)$. The original OPE 5.3 and BRST differential 5.4 do not change, but we have the following new OPE

$$\Gamma_\alpha(z)\widetilde{\Gamma}^\alpha(0) = \frac{\delta_{\alpha\beta}}{z}\,, \tag{5.11}$$

and new BRST differential

$$QΓ_α = E_α(Φ_i),$$
$$QΓ^α = J^α(Φ_i). \tag{5.12}$$

We can flip the boundary condition of a bulk chiral from Neumann to Dirichlet by introducing a boundary fermi multiplet. We start with the boundary condition with the first $m$ chiral multiplets given Neumann boundary condition, and the last $n - m$ chiral multiplets given Dirichlet boundary condition. Suppose the superpotential vanishes at the boundary. We couple this system with a boundary fermi multiplet $(Γ(z), \widetilde{Γ}(z))$ by a $E$ term $E = X$ and no $J$ term as in 5.10. The vertex algebra of the coupled system is argued [12] to be equivalent to that of another boundary condition where the first $m - 1$ chiral multiplets are given Neumann boundary condition, and the last $n - m + 1$ chiral multiplets are given Dirichlet boundary condition. This isomorphism is only expected to be true at the level of cohomology.

We need to mention that if we only care about the vertex algebra structure at the level of cohomology, then the dg vertex algebra model proposed in this section is enough. However, this model does not fully characterize the algebraic structure of boundary algebra with general superpotential. This is related to the fact that the superpotential might have non zero higher derivatives (of order 3 or higher). That information should be contained in the structure of boundary algebra but is lost in our model. In this case, the boundary chiral algebra should form an $A_∞$ analog of chiral algebra, where we have higher operations coming from the higher derivatives of the superpotential. It should produce the familiar $A_∞$ algebra upon reduction on a circle. These higher operations might sound unfamiliar but will be important for the conjectural bulk-boundary relation to work as we will see later.

## 5.2  Boundary algebras for vector multiplets

First, we consider Neumann boundary condition for the gauge fields, which set $\mathbf{B} = 0$ at the boundary. The boundary algebra is generated by $\mathbf{A} \in \mathcal{A}^• ⊗ \mathfrak{g}[1]$ and by passing to cohomology the boundary algebra is generated by the ghost fields $c^a$. The OPE structure is trivial but we have a nontrivial BRST operator.

$$Qc^a = f^a_{bc} c^b c^c. \tag{5.13}$$

As we have discussed in Section 3.2, When the gauge group is compact, we should only introduce ghosts for non-constant gauge transformations and impose $G$ invariance by hand by only considering $G$ invariant operators. In summary, the boundary algebra is:

$$C^•(z\mathfrak{g}[z])^G. \tag{5.14}$$

Explicitly, The boundary algebra is generated by $G$ invariant combinations of $∂_z c^a, ∂_z^2 c^a, \ldots$ having trivial OPE and equipped with the BRST operator

$$Q∂_z^n c^a = \sum_{\substack{l+k=n \\ l,k>0}} f^a_{bc} ∂_z^l c^b ∂_z^k c^c. \tag{5.15}$$

Now we consider Dirichlet boundary conditions for the gauge field. This sets $\mathbf{A} = 0$ at the boundary. Naively, the boundary algebra is generated by fields $B_a$, the bottom component of $\mathbf{B} \in \mathcal{A}^{1,•} ⊗ \mathfrak{g}^*$. The BRST differential is trivial and the OPE is given by

$$B_a(z)B_b(0) \sim \frac{f^c_{ab}}{z} B_c + \frac{k}{z^2} δ_{ab} - \frac{h^∨}{z^2} δ_{ab}, \tag{5.16}$$

where $h^∨$ is the dual coxter number. We denote this vertex algebra by $V_{k-h^∨}(\mathfrak{g})$. This is exactly the vertex algebra associated with the vacuum representation of level $k - h^∨$ of the affine Kac-Moody algebra $\widehat{\mathfrak{g}}$.

There are further non-perturbative corrections to the boundary algebra with Dirichlet boundary conditions. To describe the correct space of local operators on the boundary, we should again use the operator-state correspondence by constructing the Hilbert space associated with the hemi-sphere ending on the boundary. A systematic analysis is performed in [12], which shows that the boundary algebra is computed as Dolbeault homology of affine Grassmannian

$$H_\bullet\left(\text{Gr}_G, \mathcal{L}^{\otimes(h-k)}\right) = H^\bullet\left(\text{Gr}, \mathcal{L}^{\otimes(k-h)}\right)^* . \tag{5.17}$$

This space is also known [57, 58] to be isomorphic to $L_{k-h}$, the integrable highest weight representation of $\widehat{\mathfrak{g}}_{k-h}$. It has been proved in [59] that $L_{k,0}$ possesses the structure of a vertex operator algebra. This is exactly the vertex operator algebra associated with WZW model.

## 5.3 Boundary algebra with vector and chiral multiplets

Now suppose we have both vector and chiral multiplets. The boundary condition could be any combination of Neumann and Dirichlet conditions. In this paper, we will focus on the case when the vector multiples take Neumann boundary conditions, which, as we have seen, has a much simpler structure. The boundary conditions for the chiral multiplets can be arbitrary. As in Section 5.1, we give Neumann boundary conditions to the first $k$ chiral multiplets and Dirichlet boundary conditions to the last $n-k$ chiral multiplets. We also add boundary fermions $(\Gamma^\alpha, \widetilde{\Gamma}_\alpha)$ with $E_\alpha$ and $J^\alpha$ terms to cancel the boundary anomaly. Including the gauge fields, the boundary algebra is generated by

$$\phi_i(z), \psi_j(z), \Gamma^\alpha(z), \widetilde{\Gamma}_\alpha(z), \quad i = 1, \dots, k, \quad j = k+1, \dots, n, \tag{5.18}$$

and their $z$ derivatives, as well as the $z$ derivatives of ghost $c_a(z)$. The nontrivial OPE is the same as before:

$$\psi_j\psi_l = \frac{\hbar}{z}\partial_j\partial_l W|_\partial , \quad \Gamma_\alpha(z)\widetilde{\Gamma}^\alpha(0) = \frac{\delta_{\alpha\beta}}{z} . \tag{5.19}$$

The BRST transformation is more complicated. Schematically, they can be written as

$$\begin{aligned}
Q\phi &= c \cdot \phi , \quad Q\psi = dW|_\partial + c \cdot \psi , \\
Q\Gamma &= c \cdot \Gamma + E , \quad Q\widetilde{\Gamma} = c \cdot \widetilde{\Gamma} + J .
\end{aligned} \tag{5.20}$$

Due to the non-trivial OPE, the algebra is no longer a polynomial algebra. Hence we need to pay extra attention to the BRST operator in this case. When computing the normal ordered product of operators, some unexpected terms may appear in the BRST transformation. We will see this in one of our examples.

In the presence of gauge fields, we should also be careful about the boundary gauge anomalies. For the boundary theory to be consistent at the quantum level, all gauge anomalies must be canceled. Details of boundary anomalies and their cancellation are provided in [56].

# 6 Self-Ext and bulk local operator algebra

It is conjectured in [12] that, the algebra of bulk local operators can be computed by the self-Ext of the vacuum module of the boundary VOA. We explain, following [12, 21], why this might be true. First, by bringing a bulk local operator $\mathcal{O} \in \text{Obs}$ to the boundary, we get an action of bulk operators Obs on boundary vertex algebra $\mathcal{V}_\partial[\mathcal{B}]$. This can be encoded as a homomorphism

$$\text{Obs} \to \text{End}\left(\mathcal{V}_\partial[\mathcal{B}]\right) . \tag{6.1}$$

On the other hand, the algebra of boundary charges $\oint \mathcal{V}_\partial[\mathcal{B}]$ acts on the space of local boundary operators via boundary OPE, and gives us a homomorphism

$$\oint \mathcal{V}_\partial[\mathcal{B}] \to \mathrm{End}\left(\mathcal{V}_\partial[\mathcal{B}]\right). \tag{6.2}$$

Since the actions of bulk operator and boundary charges on $\mathcal{V}_\partial[\mathcal{B}]$ are defined by operator product in $\mathbb{R}$ and $\mathbb{C}$ direction respectively, they must commute with each other. Therefore the image of Obs in $\mathrm{End}(\mathcal{V}_\partial[\mathcal{B}])$ lies in $\mathrm{End}_{\oint \mathcal{V}_\partial[\mathcal{B}]}(\mathcal{V}_\partial[\mathcal{B}])$, and we get a "bulk boundary" map

$$\beta : \mathrm{Obs} \to \mathrm{End}_{\oint \mathcal{V}_\partial[\mathcal{B}]}(\mathcal{V}_\partial[\mathcal{B}]). \tag{6.3}$$

The algebra $\mathrm{End}_{\oint \mathcal{V}_\partial[\mathcal{B}]}(\mathcal{V}_\partial[\mathcal{B}])$ is also known as the center $Z(\mathcal{V}_\partial[\mathcal{B}])$ of the VOA $\mathcal{V}_\partial[\mathcal{B}]$, so we also write $\beta : \mathrm{Obs} \to Z(\mathcal{V}_\partial[\mathcal{B}])$. It was argued in [12] that this center is neither injective nor surjective in general. However, we would expect this "bulk boundary" map to contain more information if we consider its derived generalization. So to say, we conjecture that there is an isomorphism

$$\beta_{\mathrm{der}} : \mathrm{Obs} \to Z_{\mathrm{der}}(\mathcal{V}_\partial[\mathcal{B}]), \tag{6.4}$$

for boundary condition $\mathcal{B}$ that is large enough (namely, when $\mathcal{B}$ is a generator of the category of boundary conditions). Here, the center is replaced by the derived center, and is computed by $\mathrm{Ext}^\bullet_{\oint \mathcal{V}_\partial[\mathcal{B}]}(\mathcal{V}_\partial[\mathcal{B}], \mathcal{V}_\partial[\mathcal{B}])$.

To explain this conjecture, we recall the $2d$ TQFT version of this statement. For a $2d$ TQFT, the boundary conditions form a category $\mathcal{C}$ (usually a dg category). For any boundary conditions $\mathcal{B}$, $\mathrm{Hom}_\mathcal{C}(\mathcal{B}, \mathcal{B}) = \mathrm{End}_\mathcal{C}(\mathcal{B})$ has the structure of a (dg) algebra. We call that the boundary condition $\mathcal{B}$ is large enough (or a generator of the category) if the category of boundary conditions is equivalent to the category of (dg) module of $\mathrm{End}_\mathcal{C}(\mathcal{B})$. On the one hand, in the axiomatic approach of TQFT [20], we take it as a definition that the bulk operator algebra is given by the derived center of the category $\mathcal{C}$, which is computed by the Hochschild cohomology of $A_\partial[\mathcal{B}] = \mathrm{End}_\mathcal{C}(\mathcal{B})$ for large enough $\mathcal{B}$. On the other hand, in the physical approach of TQFT, the bulk operators can be identified as closed string states and can be computed directly. The isomorphism between the bulk operator algebra and Hochschild cohomology of boundary algebra $A_\partial[\mathcal{B}]$ is considered in various cases [60–62].

The analogous statement should hold in $3d$ TQFT. For A and B twist of $3d$ $N = 4$ theory, this was used in [21,22] to compute Higgs and Coulomb branches operators. In the following sections, we perform the calculation for various $3d$ holomorphic twist theories. The expected relation is verified for "Landau-Ginzburg" model with arbitrary superpotential. For theories with vector multiplets, it is much harder to compute the Ext, due to the rather complicated structure of the boundary chiral algebra. At the moment, we are only able to check some examples with abelian gauge group and in Neumann boundary condition. Even for these simple examples, it is fascinating to see that bulk monopole operators show up in the self-Ext of boundary algebra that is purely perturbative.

## 6.1 Chiral multiplet

In this section, we illustrate the above ideas by looking at theories with only chiral multiplets. Although much of the analysis can be done in an abstract way by properties of $R\mathrm{Hom}$, we still perform the calculation in a very concrete manner by writing down explicitly the Koszul resolution.

Imposing a boundary condition usually "freezes" half of the bulk degrees of freedom. It's very interesting to see how the "frozen" half of degrees of freedom naturally arises from the Koszul resolution and eventually gives us the right bulk algebra. It's also very interesting to see how the various terms in the differential of bulk algebra are obtained from the Ext calculation.

### 6.1.1 Single chiral multiplet with D b.c.

We start by considering the simplest example, a single chiral multiplet $(\boldsymbol{\Phi}, \boldsymbol{\Psi})$ without any superpotential. We consider Dirichlet boundary condition. According to our previous discussion, the boundary algebra is generated by a field $\psi(z)$. There is no BRST operator and non-trivial OPE. The algebra of charges is generated by

$$\psi_n = \oint z^{n-1}\psi(z), \; n \in \mathbb{Z}. \tag{6.5}$$

All these modes commute because there is no OPE. We have

$$\oint \mathcal{V}_\partial[D] = \mathbb{C}[\{\psi_n\}_{n\in\mathbb{Z}}]. \tag{6.6}$$

The vacuum module $\mathcal{M}_{vac}$ is generated by a vacuum vector $|0\rangle$ annihilated by charges $\psi_n$ for $n \geq 0$. Equivalently we can write

$$\mathcal{M}_{vac} = \mathbb{C}[\{\psi_n\}_{n<0}]. \tag{6.7}$$

This vacuum module has a free resolution by adjoining $\oint \mathcal{V}_\partial[D]$ infinite many bosonic variables $\lambda_n, n \geq 0$, which is also known as the Koszul resolution [63]. We have

$$\widetilde{\mathcal{M}}_{vac} = \oint \mathcal{V}_\partial[D] \otimes \mathbb{C}[\{\lambda_n\}_{n\geq 0}]. \tag{6.8}$$

The differential $d$ sends $\lambda_n \to \psi_n$, for $n \geq 0$. The self-Ext is computed as maps of $\oint \mathcal{V}_\partial[D]$ module from $\widetilde{\mathcal{M}}_{vac}$ to $\mathcal{M}_{vac}$. We have[3]

$$\begin{aligned}
\mathrm{Hom}_{\oint \mathcal{V}_\partial[D]}(\widetilde{\mathcal{M}}_{vac}, \mathcal{M}_{vac}) &= \mathrm{Hom}_{\mathbb{C}}\left(\mathbb{C}[\{\lambda_n\}_{n\geq 0}], \mathcal{M}_{vac}\right) \\
&= \mathcal{M}_{vac}[\{\lambda_n^*\}_{n\geq 0}].
\end{aligned} \tag{6.9}$$

This complex has no differential, hence the cohomology is the complex itself. If we change our notation

$$\psi_{-n-1} \to \partial^n\psi, \; \lambda_n^* \to \partial^n\phi, \tag{6.10}$$

the self-Ext can be written as

$$\mathbb{C}\left[\{\partial^n\psi, \partial^n\phi\}_{n\geq 0}\right]. \tag{6.11}$$

This is exactly the bulk algebra for a single free chiral.

We can consider a complex mass deformation of the single free chiral by adding a superpotential:

$$W(\boldsymbol{\Phi}) = \frac{1}{2}M\boldsymbol{\Phi}^2. \tag{6.12}$$

With this deformation, the boundary chiral algebra in D b.c. has the following OPE

$$\psi(z)\psi(0) \sim \frac{M}{z}. \tag{6.13}$$

The algebra of charges now has the following anti commutation[4] relation

$$[\psi_n, \psi_m] = M\delta_{n+m,-1}. \tag{6.14}$$

---

[3]We use restricted dual throughout this section.

[4]In this paper we write both commuting and anti commuting brackets by $[-,-]$. With this notation, the bracket is defined by $[f, g] = fg - (-1)^{|f||g|}gf$, where $|a|$ is the cohomological degree of $a$.

Then the associated algebra of charges is no longer the polynomial algebra 6.6, but a Clifford algebra generated by relations 6.14. However, we can still identify the vacuum module as 6.7 and the resolution 6.8 is still valid as a resolution of the vacuum module. But we need to be careful about the differential here. The $\psi_n$ in the differential $d\lambda_n = \psi_n$ is now understood as a multiplication by $\psi_n$. The self-Ext is computed by the complex 6.9, but now with a non vanishing differential. Given a $f \in \mathrm{Hom}_{\mathbb{C}}(\mathbb{C}[\{\lambda_n\}_{n\geq 0}], \mathcal{M}_{vac})$. By definition, its differential is computed by

$$
\begin{aligned}
df(\lambda_{n_1} \dots \lambda_{n_k}) &= \sum f(\lambda_{n_1} \dots d\lambda_{n_j} \dots \lambda_{n_k}) \\
&= \sum \psi_{n_j} f\left(\lambda_{n_1} \dots \hat{\lambda}_{n_j} \dots \lambda_{n_k}\right), \qquad \text{for } n_j \geq 0.
\end{aligned}
\tag{6.15}
$$

$\psi_{n_j}$ acts on $\mathcal{M}_{vac}$ via the relation 6.14. We found that it is identified with $\partial_{\psi_{-1-n_j}}$. We further identify $\mathrm{Hom}_{\mathbb{C}}(\mathbb{C}[\{\lambda_n\}_{n\geq 0}], \mathcal{M}_{vac}) = \mathcal{M}_{vac}[\{\lambda_n^*\}_{n\geq 0}]$. Then the differential on $\mathcal{M}_{vac}[\{\lambda_n^*\}_{n\geq 0}]$ is given by

$$
d = M \sum_{n\geq 0} \lambda_n^* \partial_{\psi_{-1-n}}.
\tag{6.16}
$$

By the change of variable 6.10, the self-Ext is then computed by the complex

$$
\left(\mathbb{C}\left[\{\partial^n \psi, \partial^n \phi\}_{n\geq 0}\right], Q\right),
\tag{6.17}
$$

with differential $Q\partial^n\psi = M\partial^n\phi$. This is exactly the cochain complex computing the bulk algebra of a massive chiral. It has cohomology $H^{\bullet}(\mathbb{C}[\{\partial^n\psi, \partial^n\phi\}_{n\geq 0}], Q) = \mathbb{C}$.

### 6.1.2 XYZ model with NDD b.c.

Let's consider the XYZ model with NDD boundary condition. In this case, the boundary algebra is generated by $X(z), \psi_Y(z), \psi_Z(z)$. According to our discussion in Section 5.1, this algebra has no BRST differential, but has a nontrivial boundary OPE

$$
\psi_Y(z)\psi_Z(0) \sim \frac{1}{z} X(0).
\tag{6.18}
$$

The algebra of charges is generated by

$$
X_n = \oint z^n X(z)dz, \quad \psi_{Y,n} = \oint z^n \psi_Y(z)dz, \quad \psi_{Z,n} = \oint z^n \psi_Z(z)dz,
\tag{6.19}
$$

with (anti)commutation relations inherited from the OPE

$$
[\psi_{Y,n}, \psi_{Z,m}] = X_{n+m}.
\tag{6.20}
$$

If we define a super Lie algebra $\mathfrak{g}_{NDD} = \mathrm{span}_{\mathbb{C}}\{X, \psi_Y, \psi_Z\}$ with one bosonic basis $X$ and two fermionic basis $\psi_Y, \psi_Z$ and only one nontrivial commutator $[\psi_Y, \psi_Z] = X$. Then we observe that the algebra of boundary charges is nothing but the universal enveloping algebra

$$
\oint \mathcal{V}_{\partial} = U\left(\mathfrak{g}_{NDD}((z))\right).
\tag{6.21}
$$

The vacuum module $\mathcal{M}_{vac}$ is generated by a vacuum vector $|0\rangle$ which is annihilated by charges $X_n, \psi_{Y,n}, \psi_{Z,n}$ for $n \geq 0$. In other words

$$
\mathcal{M}_{vac} = U(\mathfrak{g}_{NDD}((z))) \otimes_{U(\mathfrak{g}_{NDD}[z])} \mathbb{C}.
\tag{6.22}
$$

Here $\mathbb{C}$ is the trivial one dimensional representation on which $U(\mathfrak{g}_{XYZ}[[z]])$ acts by zero. We recall the well known Chevalley-Eilenberg cochain complex that provides a resolution of $\mathbb{C}$

$$U(\mathfrak{g}_{NDD}[[z]]) \otimes \bigwedge{}^{\bullet} \mathfrak{g}_{NDD}[[z]] \to \mathbb{C}. \tag{6.23}$$

By tensoring with $\mathcal{M}_{vac}$ we find a free resolution of $\mathcal{M}_{vac}$ as a $U(\mathfrak{g}_{NDD}((z)))$ module,

$$\widetilde{\mathcal{M}}_{vac} = U(\mathfrak{g}_{NDD}((z))) \otimes \bigwedge{}^{\bullet} \mathfrak{g}_{NDD}[[z]]. \tag{6.24}$$

Explicitly we adjoin to $\oint \mathcal{V}_{\partial}$ infinite many fermionic variables $\eta_{X,n}$ and bosonic variables $y_n, z_n$ for $n \geq 0$. The Chevalley-Eilenberg differential can be written as

$$d = \sum_{n \geq 0} X_n \partial_{\eta_{X,n}} + \psi_{Y,n} \partial_{y_n} + \psi_{Z,n} \partial_{z_n} + \sum_{n,m \geq 0} \eta_{X,n+m} \partial_{y_n} \partial_{z_m}. \tag{6.25}$$

Here by $\psi_{Y,n}$, we mean a left multiplication by $\psi_{Y,n}$, and similarly for $X_n, \psi_{Z,n}$.

The $R\mathrm{Hom}$ of $\mathcal{M}_{vac}$ to itself is then the complex of maps of $\oint \mathcal{V}_{\partial}$-module from $\widetilde{\mathcal{M}}_{vac}$ to $\mathcal{M}_{vac}$:

$$\begin{aligned} R\mathrm{Hom}_{\oint \mathcal{V}_{\partial}}(\mathcal{M}_{vac}, \mathcal{M}_{vac}) &= \mathrm{Hom}_{\oint \mathcal{V}_{\partial}}(\widetilde{\mathcal{M}}_{vac}, \mathcal{M}_{vac}) \\ &= \mathrm{Hom}_{\mathbb{C}}(\bigwedge{}^{*} \mathfrak{g}_{NDD}[[z]], \mathcal{M}_{vac}). \end{aligned} \tag{6.26}$$

This complex is $\mathcal{M}_{vac} \otimes \mathbb{C}[\{\eta_{X,n}^*, y_n^*, z_n^*\}_{n \geq 0}]$, where the generators $\eta_{X,n}^*, y_n^*, z_n^*$ are linear dual of $\eta_{X,n}, y_n, z_n$, respectively. The differential on this complex is inherited from the Chevalley-Eilenberg differential 6.25. It is given by

$$d = \sum_{n,m \geq 0} y_n^* z_m^* \frac{\partial}{\partial \eta_{X,n+m}^*} + \sum_{n \geq 0} y_n^* \psi_{Y,n} + z_n^* \psi_{Z,n}. \tag{6.27}$$

Since all the negative modes in the boundary charges commute with each other, we can write the vacuum module as a polynomial algebra

$$\mathbb{C}[\{X_n, \psi_{Y,n}, \psi_{Z,n}\}_{n<0}].$$

Then left multiplication by $\psi_{Y,n}$ for $n \geq 0$ is identified with the operator $\sum_{-n < m < 0} X_{n+m} \partial_{\psi_{Z,m}}$ and left multiplication by $\psi_{Z,n}$ for $n \geq 0$ is identified with $\sum_{-n < m < 0} X_{n+m} \partial_{\psi_{Y,m}}$. Hence the differential becomes

$$d = \sum_{n,m \geq 0} y_n^* z_m^* \partial_{\eta_{X,n+m}} + \sum_{n \geq 0, -n < m < 0} X_{n+m} y_n^* \partial_{\psi_{Z,m}} + X_{n+m} z_n^* \partial_{\psi_{Y,m}}. \tag{6.28}$$

To make things clear we can change the notation $\partial^n \psi_X = \eta_{X,n}^*, \partial^n Y = y_n^*, \partial^n Z = z_n^*$ and $\partial^n X = X_{-n-1}, \partial^n \psi_Y = \psi_{Y,-n-1}, \partial^n \psi_Z = \psi_{Z,-n-1}$ for $n \geq 0$. After this change of notation, the complex computing self-$R\mathrm{Hom}$ is generated by $\partial^n X, \partial^n Y, \partial^n Z, \partial^n \psi_X, \partial^n \psi_Y, \partial^n \psi_Z$ for $n \geq 0$. One can easily check that the differential is the same as 4.3.

$$Q\partial^n \psi_X = \sum_{0 \leq l \leq n} \partial^{n-l} Y \partial^l Z, \quad Q\partial^n \psi_Y = \sum_{0 \leq l \leq n} \partial^{n-l} Z \partial^l X, \quad Q\partial^n \psi_Z = \sum_{0 \leq l \leq n} \partial^{n-l} X \partial^l Y. \tag{6.29}$$

This is indeed the bulk algebra of XYZ model.

### 6.1.3 XYZ model with NNN b.c.

If we choose Neumann boundary condition for all the bulk chiral multiplets, the superpotential $W = \mathbf{XYZ}$ does not vanish at the boundary. As we discussed in Section 5.1, we must add boundary fermions with appropriate $E$ and $J$ terms to factorize the superpotential. In our case, a convenient choice is by adding one boundary Fermi multiplet $(\mathbf{\Gamma}(z), \widetilde{\mathbf{\Gamma}}(z))$ and taking $E = \mathbf{X}$, $J = \mathbf{YZ}$. With this choice, the boundary chiral algebra is generated by $X(z), Y(z), Z(z), \Gamma(z), \widetilde{\Gamma}(z)$ with boundary SUSY/BRST differential

$$Q\Gamma(z) = X(z), \quad Q\widetilde{\Gamma}(z) = YZ(z), \tag{6.30}$$

and a nontrivial OPE

$$\Gamma(z)\widetilde{\Gamma}(0) \sim \frac{1}{z}. \tag{6.31}$$

The algebra of charges is generated by $\{X_n, Y_n, Z_n, \Gamma_n, \widetilde{\Gamma}_n\}_{n \in \mathbb{Z}}$ where

$$X_n = \oint z^n X(z)dz, \tag{6.32}$$

and similarly for $Y_n, Z_n, \Gamma_n, \widetilde{\Gamma}_n$. The OPE gives us the following anti-commutation relation on the algebra of charges

$$[\Gamma_n, \widetilde{\Gamma}_m] = \delta_{n+m,-1}. \tag{6.33}$$

The boundary SUSY/BRST differential becomes the following differential

$$d\Gamma_n = X_n, \quad d\widetilde{\Gamma}_n = \sum_{m+l=n-1} Y_m Z_l. \tag{6.34}$$

The (differential graded) vacuum module $\mathcal{M}_{vac}$ is generated by a vacuum vector $|0\rangle$ annihilated by $X_n, Y_n, Z_n, \Gamma_n, \widetilde{\Gamma}_n$ for $n \geq 0$. A free resolution is obtained by adjoining to the algebra $\oint \mathcal{V}_\partial$ infinite many fermionic variables $\{\eta_{X,n}, \eta_{Y,n}, \eta_{Z,n}\}_{n \geq 0}$ and bosonic variables $\{\sigma_n, \widetilde{\sigma}_n\}_{n \geq 0}$. Based on our previous construction, a naive choice of Koszul differential is

$$d\eta_{X,n} = X_n, \ d\eta_{Y,n} = Y_n, \ d\eta_{Z,n} = Z_n, \ d\sigma_n = \Gamma_n, \ d\widetilde{\sigma}_n = \widetilde{\Gamma}_n, \tag{6.35}$$

where, as before, $X_n$ represents left multiplication by $X_n$ and so on. However, this differential is not compatible with the original differential 6.34 on $\oint \mathcal{V}_\partial$. For example, $d^2\sigma_n = d\Gamma_n = X_n \neq 0$. To remedy this problem we modify our differential on the variable $\{\sigma_n, \widetilde{\sigma}_n\}_{n \geq 0}$ as follows

$$\begin{aligned}
d\sigma_n &= \Gamma_n - \eta_{X,n}, \\
d\widetilde{\sigma}_n &= \widetilde{\Gamma}_n - \sum_{0 \leq m < n} (Y_{n-1-m}\eta_{Z,m} + Z_{n-1-m}\eta_{Y,m}) \\
&\quad - \sum_{m \geq n} (Y_{n-1-m}\eta_{Z,m} + Z_{n-1-m}\eta_{Y,m}).
\end{aligned} \tag{6.36}$$

One can easily check that this modified $d$ satisfies $d^2 = 0$ and is a valid differential. We denote this complex by $(\widetilde{\mathcal{M}}_{vac}, d)$. We can also prove that this complex is quasi-isomorphic to the vacuum module. To show this we consider a double complex whose horizontal differential $d_h$ is given by 6.36 only, and whose vertical differential is $d_v = d - d_h$. Consider the associated spectral sequence whose first page has differential $d_h$. Then we note that the complex $(\widetilde{\mathcal{M}}_{vac}, d_h)$ is actually the standard Koszul resolution with respect to a regular sequence. The cohomology of the first page then sits completely in degree 0, and we have

$$H^0(\widetilde{\mathcal{M}}_{vac}, d_h) \cong \mathcal{M}_{vac}. \tag{6.37}$$

The vertical differential on $\mathcal{M}_{vac}$ is exactly the boundary SUSY/BRST differential on the vacuum module. Therefore the total complex $(\widetilde{\mathcal{M}}_{vac}, d)$ is the resolution of $\mathcal{M}_{vac}$ that we are looking for.

After obtaining the right resolution, we can proceed to compute the self-$R$Hom. The complex is given by $\mathcal{M}_{vac} \otimes \mathbb{C}[\{\eta_{X,n}^*, \eta_{Y,n}^*, \eta_{Z,n}^*, \sigma_n^*, \widetilde{\sigma}_n^*\}_{n \geq 0}]$ with differential

$$d = \sum_{n<0} X_n \partial_{\Gamma_n} + \sum_{n,m<0} Y_n Z_m \partial_{\widetilde{\Gamma}_{n+m+1}}$$
$$- \sum_{n \geq 0} \left( \sigma_n^* \partial_{\eta_{X,n}^*} + \sum_{0 \leq m \leq n} \widetilde{\sigma}_m^* Z_{m-n-1} \partial_{\eta_{Y,n}^*} + \widetilde{\sigma}_m^* Y_{m-n-1} \partial_{\eta_{Z,n}^*} - \sigma_n^* \Gamma_n - \widetilde{\sigma}_n^* \widetilde{\Gamma}_n \right). \tag{6.38}$$

As before, we can identify the vacuum module with the polynomial algebra

$$\mathbb{C}\left[ \{X_n, Y_n, Z_n, \Gamma_n, \widetilde{\Gamma}_n\} \right]_{n<0}. \tag{6.39}$$

Hence left multiplication by $\Gamma_n$ is identified with $\partial_{\widetilde{\Gamma}_{-n-1}}$ for $n \geq 0$ and left multiplication by $\widetilde{\Gamma}_n$ is identified with $\partial_{\Gamma_{-n-1}}$ for $n \geq 0$. After this identification, we can rewrite the differential as

$$d = \sum_{n<0} \widetilde{\sigma}_{-n-1}^* \partial_{\Gamma_n} + \sigma_{-n-1}^* \partial_{\widetilde{\Gamma}_n} + \sum_{n<0} X_n \partial_{\Gamma_n} + \sum_{n,m<0} Y_n Z_m \partial_{\widetilde{\Gamma}_{n+m+1}}$$
$$- \sum_{n \geq 0} \left( \sigma_n^* \partial_{\eta_{X,n}^*} + \sum_{0 \leq m \leq n} \widetilde{\sigma}_m^* Z_{m-n-1} \partial_{\eta_{Y,n}^*} + \widetilde{\sigma}_m^* Y_{m-n-1} \partial_{\eta_{Z,n}^*} \right). \tag{6.40}$$

This is a very large complex, but we can use a spectral sequence to eliminate the variable $\sigma_n, \widetilde{\sigma}_n$ for $n \geq 0$ and $\Gamma_n, \widetilde{\Gamma}_n$ for $n < 0$. We prove in Appendix C that this complex has the same cohomology as a complex freely generated by $\{X_{-n-1}, Y_{-n-1}, Z_{-n-1}, \eta_{X,n}^*, \eta_{Y,n}^*, \eta_{Z,n}^*\}_{n \geq 0}$ with differential

$$d = \sum_{n \geq 0} \sum_{0 \leq m \leq n} Y_{-m-1} Z_{m-n-1} \partial_{\eta_{X,n}^*} + X_{-m-1} Z_{m-n-1} \partial_{\eta_{Y,n}^*} + X_{-m-1} Y_{m-n-1} \partial_{\eta_{Z,n}^*}. \tag{6.41}$$

We can observe that, by a change of notation, this is indeed the complex computing bulk algebra of the XYZ model.

### 6.1.4 XYZ model with DDD b.c.

In this section, we choose Dirichlet boundary conditions for all three chiral multiplets in the XYZ model. If we naively work with the vertex algebra model, we are left with fields $\psi_X(z), \psi_Y(z), \psi_Z(z)$ at the boundary. The boundary chiral algebra neglecting all higher operations is trivial. There are no boundary differential and non trivial OPE. Thus the self-Ext computation will give us the algebra of three free chiral multiplets, which is not correct. To give the right bulk algebra we need to encode the superpotential $\partial_X \partial_Y \partial_Z W = 1$ into the structure of boundary algebra. To do this we use the flipping technique, which provides us a dg vertex model for the yet known $A_\infty$ vertex algebra at the boundary.

We couple a boundary fermi multiplet $(\Gamma(z), \widetilde{\Gamma}(z))$ with the NDD boundary condition boundary algebra $(X(z), \psi_Y(z), \psi_Z(z))$ with coupling $E = X$. This dg vertex algebra has a boundary differential

$$Q\Gamma(z) = X(z), \quad Q\widetilde{\Gamma}(z) = 0, \tag{6.42}$$

and boundary OPE

$$\psi_Y(z)\psi_Z(0) \sim \frac{1}{z}X(0),$$
$$\Gamma(z)\widetilde{\Gamma}(0) \sim \frac{1}{z}. \tag{6.43}$$

This dg vertex algebra is expected to provide us with the right model of the "$A_\infty$" boundary algebra with DDD b.c. The algebra of charges of this dg vertex algebra can be described as follows. We define $\mathfrak{g}_{NDDF} := \mathrm{span}_{\mathbb{C}}\{\Gamma, \tilde{\Gamma}, X, \psi_Y, \psi_Z\}$ and

$$\hat{\mathfrak{g}}_{NDDF} := \mathfrak{g}_{NDDF}((z)) \otimes \mathbb{C}K. \tag{6.44}$$

We have the Lie bracket

$$\begin{aligned}
[\psi_Y \otimes z^n, \psi_Z \otimes z^m] &= X \otimes z^{n+m}, \\
[\Gamma \otimes z^n, \Gamma \otimes z^m] &= K\delta_{n+m,-1},
\end{aligned} \tag{6.45}$$

and a differential

$$d\Gamma \otimes z^n = X \otimes z^n. \tag{6.46}$$

Then the algebra of charges of the boundary algebra can be written as

$$\oint \mathcal{V}_\partial[DDD] \cong \oint \mathcal{V}_\partial[NDD + \mathrm{fermi}] = U(\hat{\mathfrak{g}}_{NDDF})/(K-1). \tag{6.47}$$

To make the relation between $\hat{\mathfrak{g}}_{NDDF}$ and the DDD boundary algebra more clear, we can compute the cohomology of $(\hat{\mathfrak{g}}_{NDDF}, d)$. It is easy to find that

$$H^\bullet(\hat{\mathfrak{g}}_{NDDF}, d) = \hat{\mathfrak{g}}_{DDD} := \mathfrak{g}_{DDD}((z)) \otimes \mathbb{C}K, \tag{6.48}$$

where $\mathfrak{g}_{DDD} = \mathrm{span}_{\mathbb{C}}\{\tilde{\Gamma}, \psi_Y, \psi_Z\}$. We can identify $\tilde{\Gamma}$ with $\psi_X$ in the boundary algebra of DDD boundary condition. The bracket 6.45 on $\hat{\mathfrak{g}}_{NDDF}$ induced a trivial bracket on $\hat{\mathfrak{g}}_{DDD}$. This is expected because the vertex algebra of DDD boundary condition has trivial OPE. However, interesting information is encoded in the higher operations. Homotopy transfer theorem [64] tells us that there is an $L_\infty$ structure on $\hat{\mathfrak{g}}_{DDD}$ such that there is an $L_\infty$ quasi-isomorphism between $(\hat{\mathfrak{g}}_{DDD}, \{l_k\}_{k\geq 3})$ and the dg Lie algebra $(\hat{\mathfrak{g}}_{NDDF}, d, [-,-])$. We can explicitly construct the higher operations $\{l_k\}_{k\geq 3}$ through the trees of [65] or by using homological perturbation theory [66]. Consider the following data of a homotopy retract

$$h \overset{\curvearrowright}{\bigcirc} (\hat{\mathfrak{g}}_{NDDF}, d) \underset{i}{\overset{p}{\rightleftarrows}} (\hat{\mathfrak{g}}_{DDD}, d = 0), \tag{6.49}$$

where $i$ and $p$ are the natural inclusion and projection respectively, and $h$ is a homotopy satisfying

$$dh + hd = 1 - p \circ i. \tag{6.50}$$

Explicitly, $h$ is given by $h(X \otimes z^n) = \Gamma \otimes z^n$ and maps all other elements to zero. The operations $l_3$ of $\hat{\mathfrak{g}}_{DDD}$ can be constructed via the following tree



We find the following $l_3$ on $\hat{\mathfrak{g}}_{DDD}$

$$\begin{aligned}
l_3(\psi_Y \otimes z^n, \psi_Z \otimes z^m, \psi_X \otimes z^l) &= \left[h([\psi_Y \otimes z^n, \psi_Y \otimes z^m]), \tilde{\Gamma} \otimes z^l\right] \\
&= K\delta_{n+m+l,-1}.
\end{aligned} \tag{6.51}$$

Indeed, this looks like a "triple OPE" $\psi_Y \psi_Z \psi_X \sim \partial_X \partial_Y \partial_Z W$ that encodes the 3rd derivative of $W$ at the boundary.

We can proceed to compute self-Ext in two ways. We can either work directly with the $L_\infty$ algebra $\hat{\mathfrak{g}}_{DDD}$ or we can work with the dg Lie algebra $\hat{\mathfrak{g}}_{NDDF}$. Calculation of self-Ext using $\oint \mathcal{V}_\partial[NDD + \text{fermi}]$ is very similar to the calculation in the previous sections and we omit them. Here we briefly comment on how to perform the calculation using $\hat{\mathfrak{g}}_{DDD}$. Resolution of the vacuum module will be given by adjoining bosonic elements $\{x_n, y_n, z_n\}_{n\geq 0}$ with a version of Chevalley-Eilenberg differential. Then the self-Ext can be computed by a complex $\mathbb{C}[\{x^*_{-n-1}, y^*_{-n-1}, z^*_{-n-1}, \psi_{X,n}, \psi_{Y,n}, \psi_{Z,n}\}_{n<0}]$ with a differential of the form $d = \sum_{\text{perm}} \sum_{n,m} y_n^* z_m^* l_3(\psi_{Y,n}, \psi_{Z,m}, -)$, which can be identified with the bulk algebra of XYZ model.

### 6.1.5 Chiral multiplets with arbitrary superpotential

In this section, we generalize the above calculation of the XYZ model to chiral multiplets with an arbitrary superpotential. The choice of boundary conditions requires some explanation. Although we can perform the self-Ext computation in any boundary condition, our description of the boundary algebra in Section 5.1 is not enough for this purpose in general. "Higher operations" in the chiral algebra related to higher order derivatives of $W$ are omitted, but they turn out to be crucial in the self-Ext computation as we can see in the last section.

There are two ways to overcome this problem. Just like for every $L_\infty/A_\infty$ algebra we can find a quasi-isomorphic dg Lie/Associative algebra, for every $A_\infty$ analog of vertex algebra we wish to find a quasi-isomorphic dg vertex algebra. For the boundary algebra studied in this paper, their quasi-isomorphic dg vertex algebra model can be found through the procedure of "filliping" boundary conditions. We can turn Dirichlet boundary conditions into Neumann boundary conditions until the $W|_\partial$ is at most quadratic as we did in the last section. Alternatively, we can directly work with a boundary condition that the dg vertex algebra model 5.1 suffice. In the following, we work with the second method.

If we give Neumann b.c. for all the chiral multiplets, no higher operations are present in our description of the boundary algebra. We only need to pay attention to the boundary anomaly introduced by the superpotential. Suppose the boundary anomaly can be canceled by a collection of boundary fermions $\{\Gamma^\alpha, \widetilde{\Gamma}_\alpha\}$ and appropriate E and J terms $E^\alpha(\boldsymbol{\Phi}|_\partial), J_\alpha(\boldsymbol{\Phi}|_\partial)$. Since we have chosen Neumann b.c. for all the chiral multiplets, $\boldsymbol{\Phi}|_\partial = \boldsymbol{\Phi}$.

The boundary chiral algebra, including boundary fermions, consists of fields $\phi^i(z), \Gamma^\alpha(z), \widetilde{\Gamma}_\alpha(z)$. Boundary BRST differential is given by

$$Q\Gamma^\alpha(z) = E^\alpha(\phi)(z), \quad Q\widetilde{\Gamma}_\alpha(z) = J_\alpha(\phi)(z). \tag{6.52}$$

Nontrivial OPE's are

$$\Gamma^\alpha(z)\widetilde{\Gamma}_\alpha(0) = \frac{1}{z}, \quad \text{for all } \alpha. \tag{6.53}$$

The boundary algebra of charges is generated by $\{\phi_{i,n}, \Gamma^\alpha_n, \widetilde{\Gamma}_{\alpha,n}\}_{n\in\mathbb{Z}}$. OPE's give us the (anti)commutation relations

$$[\Gamma^\alpha_n, \widetilde{\Gamma}_{\alpha,m}] = \delta_{n+m,-1}. \tag{6.54}$$

The BRST transformations give us the differential

$$d\Gamma^\alpha_n = E^\alpha_n, \quad d\widetilde{\Gamma}_{\alpha,n} = J_{\alpha,n}. \tag{6.55}$$

Here the symbols $E^\alpha_n, J_{\alpha,n}$ need some explanations. Suppose the E terms polynomials are given by

$$E^\alpha(\phi) = \sum_{\{i_k\}} a_{\{i_k\}} \phi_{i_1} \phi_{i_2} \cdots \phi_{i_l}, \tag{6.56}$$

for some constant $a_{\{i_k\}}$ coefficients. Then we define

$$E_n^\alpha := \sum_{\{i_k\}} a_{\{i_k\}} \sum_{\substack{n_i \in \mathbb{Z} \\ n_1+n_2+\cdots+n_l=n+1-l}} \phi_{i_1,n_1}\phi_{i_2,n_2}\cdots\phi_{i_l,n_l}. \tag{6.57}$$

We define $J_{\alpha,n}$ in a similar way.

The vacuum module is generated by a vacuum vector $|0\rangle$ annihilated by $\phi_{i,n}, \Gamma_n^\alpha, \widetilde{\Gamma}_{\alpha,n}$ for $n \geq 0$. Motivated by our previous example, we obtain a Koszul type resolution of the vacuum module by adjoining to the algebra $\oint \mathcal{V}_\partial$ infinite many fermionic variables $\{\eta_{i,n}\}_{n\geq0}$ and bosonic variables $\{\sigma_n^\alpha, \widetilde{\sigma}_{\alpha,n}\}_{n\geq0}$. The nontrivial part is to construct an appropriate differential making the complex quasi-isomorphic to the vacuum module. We first define an operator $h : \mathbb{C}[\{\phi_{i,n}\}_{n\in\mathbb{Z}}] \to \oplus_{j,m\geq0}\mathbb{C}[\{\phi_{i,n}\}_{n\in\mathbb{Z}}]\eta_{j,m}$ as follows

$$h\left(\phi_{i_1,n_1}\cdots\phi_{i_l,n_l}\right) = \frac{1}{\displaystyle\sum_{\substack{j=1,\ldots,l \\ \text{with } n_j\geq0}}1} \sum_{\substack{j=1,\ldots,l \\ \text{with } n_j\geq0}} \phi_{i_1,n_1}\cdots\hat{\phi}_{i_j,n_j}\cdots\phi_{i_l,n_l}\eta_{i_j,n_j}, \tag{6.58}$$

if at least one $n_j \geq 0$, and $h\left(\phi_{i_1,n_1}\cdots\phi_{i_l,n_l}\right) = 0$ if $n_j < 0$ for all $j$.

We define the differential as follows

$$\begin{aligned} d\eta_{i,n} &= \phi_{i,n}, \\ d\sigma_n^\alpha &= \Gamma_n^\alpha - h(E_n^\alpha), \quad d\widetilde{\sigma}_{\alpha,n} = \widetilde{\Gamma}_{\alpha,n} - h(J_{\alpha,n}). \end{aligned} \tag{6.59}$$

For $n \geq 0$, at least one $n_j$ in the sequence $(n_1,\ldots,n_l) \in \mathbb{Z}^l$ satisfying $n_1+n_2+\cdots+n_l = n+1-l$ will be greater than 0. Thus we have

$$\begin{aligned} dh\left(\phi_{i_1,n_1}\cdots\phi_{i_l,n_l}\right) &= \frac{1}{\displaystyle\sum_{\substack{j=1,\ldots,l \\ \text{with } n_j\geq0}}1}\left(\sum_{\substack{j=1,\ldots,l \\ \text{with } n_j\geq0}}1\right)\phi_{i_j,n_j}\cdots\phi_{i_l,n_l} \\ &= \phi_{i_j,n_j}\cdots\phi_{i_l,n_l}. \end{aligned} \tag{6.60}$$

This implies that $dh(E_n^\alpha) = E_n^\alpha$ and $dh(J_{\alpha,n}) = J_{\alpha,n}$ for $n \geq 0$. It follows that $d^2 = 0$, so this is a well defined differential. Denote this complex by$(\widetilde{\mathcal{M}}_{vac}, d)$. We also need to prove that $(\widetilde{\mathcal{M}}_{vac}, d)$ has the same cohomology as the vacuum module. We consider a double complex whose horizontal differential is given by 6.59 only and whose vertical differential is given by $d - d_h$. As in our previous example, the cohomology with respect to the horizontal differential is isomorphic to $\mathcal{M}_{vac}$ and sit completely in cohomological degree 0. $d_v$ restricted on $\mathcal{M}_{vac}$ is the same as the BRST differential on $\mathcal{M}_{vac}$. Hence $(\widetilde{\mathcal{M}}_{vac}, d)$ is quasi-isomorphic to the vacuum module.

We find that the self-$R$Hom can be computed by the complex $\mathcal{M}_{vac} \otimes \mathbb{C}[\{\eta_{i,n}^*, \sigma_n^{\alpha*}, \widetilde{\sigma}_{\alpha,n}^*\}]$ with differential induced from $d$ of $\widetilde{\mathcal{M}}_{vac}$. Equivalently, we have the polynomial algebra $\mathbb{C}[\{\phi_{i,-n-1}, \Gamma_{-n-1}^\alpha, \widetilde{\Gamma}_{\alpha,-n-1}, \eta_{i,n}^*, \sigma_n^{\alpha*}, \widetilde{\sigma}_{\alpha,n}^*\}_{n\geq0}]$ with some differential. After appropriately identifying various terms in the differential, we find that

$$\begin{aligned} d = &\sum_{n<0,\alpha} \widetilde{\sigma}_{\alpha,-n-1}^*\partial_{\Gamma_n^\alpha} + \sigma_{-n-1}^\alpha\partial_{\widetilde{\Gamma}_{\alpha,n}} + E_n^\alpha\Big|_{\substack{\phi_{j,k}=0 \\ \text{for } k\geq0}}\partial_{\Gamma_n^\alpha} + J_{\alpha,n}\Big|_{\substack{\phi_{j,k}=0 \\ \text{for } k\geq0}}\partial_{\widetilde{\Gamma}_{\alpha,n}} \\ &+ \sum_{n\geq0,m\geq0,\alpha}\left(\sigma_m^{\alpha*}\langle\eta_{i,n}^*, h(E_m^\alpha)\rangle + \widetilde{\sigma}_{\alpha,m}^*\langle\eta_{i,n}^*, h(J_{\alpha,m})\rangle\right)\Big|_{\substack{\phi_{j,k}=0 \\ \text{for } k\geq0}}\partial_{\eta_{i,n}^*}, \end{aligned} \tag{6.61}$$

where we used the dual pairing $\langle \eta^*_{i,n}, \eta_{j,m} \rangle = \delta_{ij}\delta_{nm}$. We prove in C that this complex is quasi-isomorphic to the complex $\mathbb{C}[\{\phi_{i,-n-1}, \eta^*_{i,n}\}_{n\geq 0}]$ with differential given by

$$d = \sum_{n\geq 0, m\geq 0, \alpha} \left( J_{\alpha,-m-1}\langle \eta^*_{i,n}, h(E^\alpha_m) \rangle + E^\alpha_{-m-1}\langle \eta^*_{i,n}, h(J_{\alpha,m}) \rangle \right) \Big|_{\substack{\phi_{j,k}=0 \\ \text{for } k>0}} \partial_{\eta^*_{i,n}}. \tag{6.62}$$

This differential can be further identified with (see C for a proof)

$$d = \sum_{n\geq 0} \left( \frac{\delta W(\phi)}{\delta \phi_i} \right)_{-n-1} \Big|_{\substack{\phi_{j,k}=0 \\ \text{for } k>0}} \partial_{\eta^*_{i,n}}. \tag{6.63}$$

By a change of notation $\phi_{i,-n-1} \to \partial^n \phi_i$, $\eta^*_{i,n} \to \partial^n \psi_i$, we see that this complex is exactly the same as the bulk algebra of chiral multiplets with superpotential $W$ that we wrote down in Section 3.1.

## 6.2 Koszul duality in boundary chiral algebra

In our previous examples of the XYZ model, the computations in different boundary conditions all give us the same result. The boundary algebras for distinct boundary conditions can be very different. It is not obvious a priori that computations in different boundary conditions are the same. This can be explained as follows. We expect the Ext computation to give us the same bulk algebra, as long as we work with boundary conditions that are large enough. This assumption that the boundary condition is large enough guarantees that the categories of modules of the boundary algebras are always equivalent to the category of boundary conditions. Although the self-Ext's are computed in different ways in terms of the boundary algebras, we are essentially working in the same category. We explain in this section that for two boundary conditions that are complementary to each other, this equivalence of categories specializes to the notion of Koszul duality.

Suppose our boundary conditions consist of a dg category $\mathcal{C}$. Consider two generators $\mathcal{B}, \mathcal{B}^!$ of the category. We call the two boundary conditions complimentary to each other if they satisfy

$$\text{Hom}_{\mathcal{C}}(\mathcal{B}, \mathcal{B}^!) \approx \mathbb{C}. \tag{6.64}$$

Physically, this corresponds to the following situation. We consider the theory to be placed at $[0,1]\times\mathbb{C}$. We give boundary conditions $\mathcal{B}$ and $\mathcal{B}^!$ on the two sides of the interval respectively. Then the condition 6.64 is equivalent to the bulk theory on $[0,1]\times\mathbb{C}$ being trivial. More explicitly, if we try to solve the Equation of motion on the interval with the boundary condition that two sets of complementary fields are set to zero on the two sides, then we only get a trivial solution.

Denote the two boundary algebras by $A_\partial = \text{End}_{\mathcal{C}}(\mathcal{B})$ and $A^!_\partial = \text{End}_{\mathcal{C}}(\mathcal{B}^!)$. The condition that $\mathcal{B}, \mathcal{B}^!$ are generators guarantees that the two algebras $A_\partial, A^!_\partial$ generate the whole commutant of each other. Therefore $A^!_\partial = \text{End}_{A_\partial}(\mathbb{C})$, $A_\partial = \text{End}_{A^!_\partial}(\mathbb{C})$. This turns out to be the definition for two Koszul dual algebras [67]. This is only a very rough argument. In fact, the boundary algebra is not associative algebra but instead vertex algebra. The notion of Koszul duality for vertex algebra has not yet been developed.[5] In this paper, by Koszul duality, we mean the Koszul duality of the corresponding algebra of charges. This is enough for our purpose. Because we are interested in the Ext computation, which only concerns the category of modules of the vertex algebra, which is equivalent to the category of modules of the associated algebra of charges [69].

---

[5]For a physical approach to the definition of Koszul duality of vertex algebra, see [68].

For instance, we consider a free single chiral without any superpotential. The boundary algebra for Dirichlet boundary condition has been worked out in Section 5.1, we have

$$\oint V_\partial[D] = \bigwedge{}^\bullet(\mathbb{C}((z))).$$  (6.65)

For Neumann boundary condition, the boundary algebra is generated by a boson $\phi(z)$. There is non boundary BRST operator and non trivial OPE, therefore the corresponding algebra of charges is

$$\oint V_\partial[N] = \mathrm{Sym}\,(\mathbb{C}((z))).$$  (6.66)

By identifying $\mathbb{C}((z))$ with the dual of itself via the residue pairing, we see that $\oint V_\partial[D] = (\oint V_\partial[N])^!$. This is the classical example of Koszul duality between symmetric algebra and exterior algebra $(\mathrm{Sym}(V))^! = \wedge^\bullet(V^*)$.

We can also consider the XYZ model. We already studied the NDD boundary condition. Using the notation of Section 6.1.2, the algebra of charges is the universal enveloping algebra

$$\oint \mathcal{V}_\partial[NDD] = U(\mathfrak{g}_{NDD}((z))).$$  (6.67)

The boundary condition complimentary to the NDD is the DNN boundary condition. The boundary algebra for DNN boundary condition is generated by fields $\psi_X(z), Y(z), Z(z)$. There is no nontrivial OPE, but we have a boundary BRST differential

$$Q\psi_X = YZ.$$  (6.68)

We expand the fields into charges $\{\psi_{X,n}, Y_n, Z_n\}$. The boundary BRST operator gives us the following differential on the algebra of charges

$$d\psi_{X,n} = \sum_{m+l=n-1} Y_m Z_l.$$  (6.69)

We find that this algebra is exactly the Chevalley-Eilenberg algebra of the loop algebra $\mathfrak{g}_{NDD}((z))$

$$\oint \mathcal{V}_\partial[NDD] = C^\bullet(\mathfrak{g}_{NDD}((z))).$$  (6.70)

We see that the boundary algebras for the DNN and the NDD boundary condition are indeed Koszul dual $C^\bullet(\mathfrak{g}_{NDD}((z))) = U(\mathfrak{g}_{NDD}((z)))^!$. This is another classical example of Koszul duality between the Universal enveloping algebra and the Chevalley-Eilenberg algebra of a Lie algebra.

It is a well-known fact that the derived categories of modules for two Koszul dual algebras are equivalent [67, 70]. This provides an explanation of why we expect to obtain the same results from the Ext computation from two complementary boundary conditions.

## 6.3  Including gauge fields

We expect that the boundary-bulk relation should work for a general holomorphic twisted $\mathcal{N} = 2$ theory. However, such a statement is harder to verify in the presence of gauge fields, due to the rather complicated structure of the boundary algebra in this case.

If we choose Dirichlet boundary condition for gauge fields. The description of the boundary chiral algebra is conjectural [12], and its vertex algebra structure is not clear. If we consider the case without chiral multiplet, the boundary monopole correction makes the boundary chiral algebra the WZW vacuum module, whose vertex algebra structure is given in [59]. We leave

the analysis of self-Ext in this case to future work. However, if we are satisfied with perturbative results, the computation can be simplified because the perturbative boundary algebra $V_k(\mathfrak{g})$ has a relatively simple structure. We provide relevant results in the next section.

Things can get easier if we choose Neumann boundary condition for gauge fields. In this case, we have better control of the vertex algebra structure, due to the absence of boundary monopole operators and the fact that most OPE's are trivial. However, imposing $G$ invariant causes difficulties in finding a resolution of the vacuum module directly. We expect that there are cases when we have a better description of the cohomology of the boundary chiral algebra. For example, theory with abelian gauge group or theory with $U(N)$ gauge group when $N$ is very large. In the following section, we provide some examples with abelian gauge group. Large $N$ gauge theories are also very interesting, as some of them may have gravity dual [68, 71]. We leave its discussion to future work.

However, we have to be very careful about the self-Ext computation, as they not always give us the correct bulk algebra. This is explained in [21], as we only expect Neumann boundary condition to work when the theory is a CFT. We provide a simple example when the Ext computation fails for Neumann boundary condition in Section 6.3.3.

Excluding those "Non-CFT" theories, the bulk boundary relation provided by the self-Ext of Neumann condition boundary algebra is very nontrivial. As we have seen, the boundary algebra for Neumann boundary condition is purely perturbative, and the bulk algebra contains non-perturbative objects. This bulk boundary relation extracts non-perturbative information merely from perturbative information on the boundary, and this amazing phenomenon only comes as a result of the general structure of field theory.

### 6.3.1 Perturbative analysis for vector multiplet

Although we don't have a good understanding of the non-perturbative result, the perturbative part is more accessible. Here we consider Dirichlet boundary condition for a pure gauge theory and focus on the perturbative algebra. At level $k$, the vacuum module is

$$V_k = U(\widehat{\mathfrak{g}}) \otimes_{U(\mathfrak{g}[[z]]\oplus\mathbb{C}K)} \mathbb{C}_k \,, \tag{6.71}$$

where $\mathbb{C}_k$ is the one dimensional representation on which $\mathfrak{g}[[z]]$ acts by 0 and $K$ acts as multiplication by $k$. The mode algebra is given by $U_k(\widehat{\mathfrak{g}}) = U(\widehat{\mathfrak{g}})/(K-k)$. Computation of $R\mathrm{Hom}$ follows the standard argument as in Section 6.1.2. We have

$$
\begin{aligned}
R\mathrm{Hom}(V_k, V_k) &= \mathrm{Hom}_{U_k(\widehat{\mathfrak{g}})}(U_k(\widehat{\mathfrak{g}}) \otimes C^{\bullet}(\mathfrak{g}[[z]]), V_k) \\
&= C^{\bullet}(\mathfrak{g}[[z]], V_k) \,.
\end{aligned}
\tag{6.72}
$$

Results on Lie algebra cohomology [63] imply that we have an isomorphism

$$H^{\bullet}(\mathfrak{g}[[z]], V_k) \approx H^{\bullet}(\mathfrak{g}[[z]], \mathfrak{g}, V_k) \otimes H^{\bullet}(\mathfrak{g}) \,. \tag{6.73}$$

The part $H^{\bullet}(\mathfrak{g})$ should be discarded since it corresponds to the constant gauge mode.[6] This matches our previous analysis in Section 3.2.1. We focus on the relative Lie algebra cohomology $H^{\bullet}(\mathfrak{g}[[z]], \mathfrak{g}, V_k)$.

Computations in [44] tell us that for $k \neq k_c$, the cohomology $H^i(\mathfrak{g}[[z]], \mathfrak{g}, V_k)$ vanishes for $i > 0$. For $i = 0$, $H^0(\mathfrak{g}[[z]], \mathfrak{g}, V_k) = \mathbb{C}$. From these results, we see that for $k \neq k_c$ (when the bare Chern-Simons level $\neq 0$), the Ext calculation gives us an operator algebra that only

---

[6]An interesting result in [44] is that if we compute the self-Extension in a suitable category, namely the category of Harish-Chandra modules $HC(\widehat{\mathfrak{g}}_\kappa, G[[t]])$, then we automatically get relative Lie algebra cohomology instead of Lie algebra cohomology.

contains the identity operator in cohomology (if we discard the $H^\bullet(\mathfrak{g})$ part). This coincides with our previous analysis in 3.2.1.

As we have discussed in Section 3.4, the local operator algebra drastically changes when the level $k$ is equal to the critical level $k_c$. We expect a similar phenomenon to happen here. For $k = k_c$, we have the following results [44]

**Theorem 6.1** *For $k = k_c$, we have an isomorphism*

$$H^\bullet(\mathfrak{g}[[z]], \mathfrak{g}, V_{k_c}) \approx \Omega^\bullet\left(\mathrm{Op}_{{}^L\mathfrak{g}}(\mathbb{D})\right), \tag{6.74}$$

*where $\Omega^\bullet(\mathrm{Op}_{{}^L\mathfrak{g}}(\mathbb{D}))$ is the algebra of differential forms on the space of ${}^L\mathfrak{g}$-opers on $\mathbb{D}$.*

the $H^0$ part is exactly the center $\mathfrak{z}(\hat{\mathfrak{g}})$ we mentioned in Section 3.2.1. The associated graded of $H^\bullet(\mathfrak{g}[[z]], \mathfrak{g}, V_{k_c})$ is the algebra we proposed for the perturbative bulk algebra in 3.2.1. The self-Ext construction naturally provides us with a quantization.

### 6.3.2 $U(1)$ **gauge theory at level 1**

Non-perturbatively, WZW models are complicated to describe in general. We have a simple example by making use of boson-fermion correspondence. Let's consider the $U(1)_1$ WZW model, which, according to boson-fermion correspondence, is isomorphic to a fermionic vertex algebra [72]. This fermion vertex algebra is exactly what we used in canceling boundary anomaly in Section 5.1. We have fields $(\Gamma(z), \widetilde{\Gamma}(z))$ and OPE

$$\Gamma(z)\widetilde{\Gamma}(0) \sim \frac{1}{z}. \tag{6.75}$$

The algebra of charges is generated by $(\Gamma_n, \widetilde{\Gamma}_n)$, with

$$\Gamma(z) = \sum_{n \in \mathbb{Z}} \Gamma_n z^{-n-1}, \quad \widetilde{\Gamma}(z) = \sum_{n \in \mathbb{Z}} \widetilde{\Gamma}_n z^{-n-1}. \tag{6.76}$$

They have commutation relation

$$[\Gamma_n, \widetilde{\Gamma}_m] = \delta_{n+m,-1}. \tag{6.77}$$

We can consider the vector space $\mathbb{C}((t))[dt]$ with an inner product defined by residue pairing. Let $Cl$ be the Clifford algebra associated to $\mathbb{C}((t))[dt]$. Then the algebra of charges can be identified with $Cl$

The vacuum fermionic fock representation $M_{vac}$ of $Cl$ is generated by a vacuum vector annihilated by $\Gamma_n, \widetilde{\Gamma}_n, n \geq 0$. Equivalently

$$M_{vac} = Cl \otimes_{\bigwedge \mathbb{C}[[t,dt]]} \mathbb{C}, \tag{6.78}$$

where $\mathbb{C}$ is the trivial representation of $\mathbb{C}[[t,dt]]$.

We have a resolution of the vacuum module given by adjoining infinite many even variable $X_n, \widetilde{X}_n$ to the algebra of charges $Cl$, written as $Cl[X_n, \widetilde{X}_n]_{n \geq 0}$. The differential is

$$d = \Gamma_n \frac{\partial}{\partial X_n} + \widetilde{\Gamma}_n \frac{\partial}{\partial \widetilde{X}_n}. \tag{6.79}$$

As before the self-Ext can be computed by the complex $M_{vac}[X_n^*, \widetilde{X}_n^*]_{n \geq 0}$. We can identify the differential

$$d = X_n^* \frac{\partial}{\partial \widetilde{\Gamma}_{-n}} + \widetilde{X}_n^* \frac{\partial}{\partial \Gamma_{-n}}. \tag{6.80}$$

Computing the cohomology we find that $\mathrm{Ext}^\bullet_{Cl}(M_{vac}, M_{vac}) = \mathbb{C}$. This is compatible with our previous observation that the bulk operator algebra has trivial cohomology when the Chern-Simons level is not zero.

### 6.3.3 A failure case

In this section, we provide a simple example where the Ext computation fails for Neumann b.c. Consider a pure $U(1)$ gauge theory without Chern-Simons term. For Neumann boundary condition, there is no boundary anomaly since the gauge group is abelian. The boundary algebra is generated by derivatives of ghost $\partial^n c$. Since $U(1)$ is abelian, there is no BRST operator. $G$ action on the ghost is also trivial, so we get the same algebra after imposing $G$ invariant. This algebra is the same as the boundary chiral algebra for a free chiral with Dirichlet boundary condition (Section 6.1.1).

The Ext computation is the same as in Section 6.1.1 and we get

$$\mathrm{Ext}^{\bullet} = \mathbb{C}[J_{\infty}T^*[1]\mathbb{C}]. \tag{6.81}$$

This is not the same as the bulk algebra $\mathbb{C}[J_{\infty}T^*[1]\mathbb{C}^{\times}]$ of the $U(1)$ gauge theory we found in Section 4.2.

This failure case also provides us with evidence why the self-Ext calculation only works when the underlying field theory is a CFT. The operator algebra takes the form of functions on the infinite jet space of some derived stack $\mathcal{X}$. The self-Ext calculation of a given boundary condition can only know about the infinitesimal neighborhood of the corresponding isotropic subspace of $\mathcal{X}$. For example, in Neumann b.c. of the $U(1)$ gauge theory, the self-Ext calculation only knows about the infinitesimal neighborhood of a fiber of $T^*[1]\mathbb{C}^{\times} \to \mathbb{C}^{\times}$. Therefore the self-Ext only reproduces $J_{\infty}T^*[1]\mathbb{C}$ instead of $J_{\infty}T^*[1]\mathbb{C}^{\times}$. However, when the theory is a CFT, $\mathcal{X}$ becomes a "cone". The tangent cone around the origin is the cone $\mathcal{X}$ itself. Therefore everything is encoded in a neighborhood of the cone point. Only in this case do we expect Ext calculation to reproduce the whole operator algebra. This indeed happens for our previous analysis of chiral multiplets.

### 6.3.4 $U(1)_{-\frac{1}{2}}+$ chiral with $(\mathcal{N},N)$ b.c.

With the $(\mathcal{N},N)$ boundary condition, we have boundary fields $\varphi(z)$ and derivatives of $c(z)$. This boundary condition has gauge anomaly [56]. To cancel it we have to add a boundary fermion $\Gamma(z),\widetilde{\Gamma}(z)$ of gauge charge $(+1,-1)$ and T charge $(+1,-1)$. The chiral algebra is generated by gauge invariant combination of $\varphi(z), \Gamma(z), \widetilde{\Gamma}(z)$ and derivative of $c(z)$, with OPE

$$\Gamma(z)\widetilde{\Gamma}(0) \sim \frac{1}{z}, \tag{6.82}$$

and BRST differential

$$Q\varphi = c\varphi, \quad Q\Gamma = c\Gamma, \quad Q\widetilde{\Gamma} = -c\widetilde{\Gamma}. \tag{6.83}$$

It's not easy to perform the Ext calculation with this VOA directly. However, its BRST cohomology has a very simple description. We first prove that the BRST cohomology of the above boundary algebra is equivalent to the boundary algebra of a single chiral with Dirichlet b.c., namely the VOA generated by a single field $\psi(z)$ with no nontrivial OPE. The equivalence is provided by a map

$$\rho: \ \psi(z) \to (\varphi\widetilde{\Gamma})(z). \tag{6.84}$$

It is clear that $\varphi\widetilde{\Gamma}$ has gauge charge 0. Since $\varphi$ has trivial OPE with $\widetilde{\Gamma}$, we have

$$Q(\varphi\Gamma) = c\varphi\Gamma + \varphi(-c)\widetilde{\Gamma} = 0. \tag{6.85}$$

Therefore, $\rho$ is indeed a map to the cohomology of the $U(1)_{-\frac{1}{2}}+$ chiral boundary algebra. The non-trivial part is to prove that $\rho$ is actually an isomorphism on the cohomology.

First, we show that the cohomology of the $U(1)_{-\frac{1}{2}}+$ chiral boundary operators are all in cohomological degree 0. From the boundary fermion we can construct the following $U(1)$ current

$$b(z) := \; : \Gamma\widetilde{\Gamma} : (z). \tag{6.86}$$

Denote $b_n$ the modes of $b(z)$, they satisfy $U(1)$ current relation $[b_n, b_m] = n\delta_{n,-m}$. Let $M$ be the vacuum module of the $U(1)_{-\frac{1}{2}}+$ chiral boundary algebra, and let $M_0$ be the sub-module of the kernel of all the non-negative modes $b_n, n \geq 0$. Then the whole vacuum module $M$ is generated by applying operators $b_n, n < 0$ to $M_0$

$$M = M_0[b_{-1}, b_{-2}, \dots]. \tag{6.87}$$

Note that the ghost $c(z)$ has trivial OPE with all other operators. Hence the vacuum module is simply a tensor product of the exterior algebra of ghost modes and the rest. We denote $N_0$ the sub-module of states that are in the kernel of $b_n, n \geq 0$ and do not contain any ghosts. Then we have

$$M = N_0[c_{-2}, c_{-3}, \dots, b_{-1}, b_{-2}, \dots]. \tag{6.88}$$

Let's study the action of BRST operators on the modes $b_n$. We use the definition of normal product to rewrite $b(0)$ as

$$b(0) = \; : \Gamma\widetilde{\Gamma} : (0) = \lim_{z \to 0} \Gamma(z)\widetilde{\Gamma}(0) - \frac{1}{z}. \tag{6.89}$$

Then we have

$$\begin{aligned}
Q b(0) &= \lim_{z \to 0}(Q\Gamma(z)\widetilde{\Gamma}(0) - \Gamma(z)Q\widetilde{\Gamma}(0)) \\
&= \lim_{z \to 0}(c(z)\Gamma(z)\widetilde{\Gamma}(0) - c(0)\Gamma(z)\widetilde{\Gamma}(0)) \\
&= \lim_{z \to 0}(c(z) - c(0))\frac{1}{z} \\
&= \partial c(0).
\end{aligned} \tag{6.90}$$

Therefore we found that the BRST operator sends $b_n$ to $c_{n-1}$ for $n < 0$. The full BRST operator can be complicated, but we can consider a spectral sequence whose differential at the first page is given by $Q_0 b_n = c_{n-1}, Q_0 c_{n-1} = 0$. Then at the first page, all ghosts $c_{n-1}$ cancel with $b_n$, and the cohomology sits completely in degree 0. Because of this, it's impossible to have further differential, the spectral sequence converges here and the cohomology is isomorphic to $N_0$.

Then we show that $N_0$ and the boundary algebra of a free chiral are isomorphic as a double graded vector space. On the free chiral side, the grading is given by spin and flavor charge $T$. Since there is only one fermionic operator $\psi$ of T charge $-1$, the coefficients in the index is the dimension of the space of operators of given quantum numbers. On the $U(1)_{-\frac{1}{2}}+$ chiral side, $N_0$ is built from operators $\varphi, \Gamma, \widetilde{\Gamma}$. We note that the bosonic operators have $T$ charge 0 and fermionic operators have $T$ charge $\pm 1$. Therefore no cancellation can happen when computing the index, and the coefficients in the index are also the dimensions of the spaces of operators of certain quantum numbers. It was shown in [19] that the indices of the two algebra are the same, so the two algebra is indeed isomorphic as double graded vector space.

Finally, to prove that $\rho$ is an isomorphism of vertex algebra, we only need to show that it is injective. This is easy because $\rho$ sends $\psi$ to $\varphi\widetilde{\Gamma}$, and there is no relation among operators built from $\varphi\widetilde{\Gamma}$ and its derivatives. Therefore the kernel of the map $\rho$ must be trivial.

Now we have proved that the cohomology of the $U(1)_{-\frac{1}{2}}+$ chiral boundary algebra is generated by a single operator $\psi(z)$ with trivial OPE. The self-Ext for this VOA is computed,

which gives us the bulk algebra of a single free chiral

$$\mathbb{C}\big[\{\partial^n\phi,\partial^n\psi\}_{n\geq 0}\big]. \tag{6.91}$$

This is expected to be quasi-isomorphic to the bulk algebra of $U(1)_{-\frac{1}{2}}+$ chiral theory by mirror duality. Moreover, this algebra contains not only perturbative operators but also non perturbative monopole operators.

We mention that it is very important to introduce the boundary fermion. They not only cancel the boundary gauge anomaly but also are crucial for the boundary bulk relation to work. We can easily figure out what the boundary algebra looks like without the fermion. $c$ is gauge neutral and $\phi$ has gauge charge 1. Therefore gauge invariant operators are generated by derivatives of $c(z)$. However, $c$ has $T$ charges 0. So we won't even be able to get the right bulk algebra as graded vector space.

### 6.3.5   SQED with $(\mathcal{N}, N, N)$ b.c.

With the $(\mathcal{N}, N, N)$ boundary condition, we have boundary fields $\phi_+(z)$, $\phi_-(z)$ and derivatives of $c(z)$. Like our previous example, this boundary condition also has gauge anomaly. To cancel this gauge anomaly we add boundary fermion $\Gamma(z), \widetilde{\Gamma}(z)$ of gauge charge $(+1, -1)$. This boundary algebra has OPE

$$\Gamma(z)\widetilde{\Gamma}(0) \sim \frac{1}{z}, \tag{6.92}$$

and BRST differential

$$\begin{aligned} Q\phi_+ &= c\phi_+, \ \ Q\phi_- = c\phi_-, \\ Q\Gamma &= c\Gamma, \ \ Q\widetilde{\Gamma} = -c\widetilde{\Gamma}. \end{aligned} \tag{6.93}$$

Performing Ext computation directly with this algebra could be difficult. However, just like our previous example, the BRST cohomology of this algebra has a much better description. It was proved in [12] that the above dg vertex algebra is equivalent to the boundary algebra of XYZ model with NDD boundary condition via a map

$$\begin{aligned} X &\to \phi_+\phi_-, \\ \rho: \ \psi_Y &\to \Gamma\phi_-, \\ \psi_Z &\to \widetilde{\Gamma}\phi_+. \end{aligned} \tag{6.94}$$

The self-Ext computation for the XYZ model is performed in the previous section and we obtained the bulk algebra of the XYZ model. By the SQED/XYZ mirror duality, this algebra should be quasi-isomorphic to the bulk algebra of the SQED model. Moreover, monopole operators of SQED automatically emerge from this algebra.

## 6.4   Algebraic structure from Ext

We have seen examples where the self-Ext computation reproduces the right vector space of the bulk algebra. We might hope that more structure of the bulk algebra could be revealed from this construction. More precisely, we conjecture that there is a shifted Poisson vertex algebra structure on $\text{Ext}^\bullet_{V-\text{mod}}(V, V)$ for a vertex algebra $V$. To motivate this conjecture we consider again the $2d$ TQFT. We have mentioned in Section 6 that the bulk algebra of a $2d$ TQFT can be computed by the Hochschild cohomology $HH^\bullet(A)$. It is known since the pioneering work of Gerstenhaber [33] that the Hochschild cohomology $HH^\bullet(A)$ has the structure of a Gerstenhaber algebra. We have a Hochschild cup product defined by

$$(f \cup g)(a_1 \otimes \cdots \otimes a_{n+m}) = f(a_1 \otimes \cdots \otimes a_n)g(a_{n+1} \otimes \cdots \otimes a_{n+m}), \tag{6.95}$$

for $f \in C^n(A,A)$ and $g \in C^m(A,A)$, and a Gerstenhaber bracket defined by

$$\{f,g\} = f \circ g - (-1)^{(m-1)(n-1)} g \circ f \,, \tag{6.96}$$

where

$$f \circ g(a_1 \otimes \cdots \otimes a_{n+m-1}) = \sum_{i=1}^{n} (-1)^{(i-1)(m-1)} f(a_1 \otimes \ldots a_{i-1} \otimes g(a_i \otimes \ldots a_{i+m-1}) \otimes \ldots a_{n+m-1}) \,. \tag{6.97}$$

This corresponds to the well-known Gerstenhaber algebra structure of the bulk operators of a $2d$ TQFT [73,74].

For any $d$ dimensional TQFT of cohomology type, we expect the bulk algebra to have the structure of a $d$-shifted Poisson algebra (which becomes Gerstenhaber algebra in $d = 2$). This structure consists of a graded product and a $d$-shifted Poisson bracket with some compatibility conditions. The field theory explanation for the two operations is explored in [38]. The product is also known as OPE of local operators, which comes from collisions of two operators

$$O_1 O_2(y) = \lim_{x \to y} O_1(x) O_2(y) \,. \tag{6.98}$$

The bracket is constructed by integrating the $(d-1)$-th descent of a operator around a small $(d-1)$-sphere surrounding the other operator

$$\{O_1, O_2\} = \oint_{S_y^{d-1}} O_1(x) O_2^{(d-1)}(y) \,. \tag{6.99}$$

They are also called the primary and secondary products.

For $3d$ $\mathcal{N} = 2$ theory with holomorphic topological twist, the algebraic structure of the bulk local operators is studied in [39], and can be summarized by saying that the space of local operators possesses the structure of shifted Poisson vertex algebra. A Poisson vertex algebra consist of a graded commutative product and $\lambda$ bracket, and correspond to the primary and secondary products respectively in the field theory context. We can combine the story of bulk algebra with our bulk-boundary construction. Since we can reproduce the space of bulk local operators from boundary algebra, we should also find the algebraic structures of Poisson vertex algebra in the self-Ext construction.

### 6.4.1 The (graded) commutative product

In this section, we construct the primary product of bulk local operators from the self-Ext construction and briefly comment on the secondary product.

We note that there is a natural isomorphism between Ext and Hochschild cohomology (see [63])

$$\mathrm{Ext}_A^\bullet(M,N) = HH^\bullet(A, \mathrm{Hom}(M,N)) \,, \tag{6.100}$$

for an arbitrary algebra $A$ and $A$-modules $M$ and $N$. This isomorphism is obtained by using the bar resolution $B_n(A,M) = A^{\otimes(n+2)} \otimes_A M \approx A^{\otimes(n+1)} \otimes M$. In our case the algebra $A$ is the algebra of charges $A = \oint V$ associated to the vertex algebra, and $M = N = V$ is the vacuum module. We have

$$\mathrm{Ext}_A^\bullet(V,V) = HH^\bullet(A, \mathrm{End}(V)) \,. \tag{6.101}$$

We can define an analog of Hochschild cup product by Equation 6.95. For $f \in \mathrm{Hom}(A^{\otimes n}, \mathrm{End}(V))$ and $g \in \mathrm{Hom}(A^{\otimes m}, \mathrm{End}(V))$, we define the cup product

$$(f \cdot g)(a_1 \otimes \cdots \otimes a_{n+m}) = f(a_1 \otimes \cdots \otimes a_n) g(a_{n+1} \otimes \cdots \otimes a_{n+m}) \,, \tag{6.102}$$

by using the composition product on $\text{End}(V)$.

Note that there is another product naturally defined on $\text{Ext}_A(V, V)$, the Yoneda product. Actually, the Yoneda product and Hochschild cup product agree via the isomorphism 6.101. To see this we first recall a definition of Yoneda product suited for our purpose. For any projective resolution $P^\bullet$ of $V$, we have a product on $\text{Hom}_A(P^\bullet, P^\bullet)$ defined via composition of chain maps. The induced composition map on cohomology is defined to be the Yoneda product on Ext. It is also known that this product does not depend on the choice of the projective resolution, so we can use the Bar resolution $B_\bullet(A, V)$ to establish the identification. We have an injective $i : \text{Hom}_A(B_\bullet(A, V), \text{End}(V)) \to \text{End}_A(B_\bullet(A, V))$ sending a $f \in \text{Hom}(A^{\otimes n} \otimes V, V)$ to $i(f) \in \bigoplus_{N \geq n} \text{Hom}_A(A^{\otimes(N+1)} \otimes V, A^{\otimes(N-n+1)} \otimes V)$ defined by

$$i(f)(a \otimes a_1 \otimes \cdots \otimes a_N \otimes m) = a \otimes a_1 \otimes \ldots a_{N-n} \otimes f(a_{N-n+1} \otimes \ldots a_N \otimes m). \tag{6.103}$$

The augmentation $\varepsilon : B_\bullet(A, V) \to V$ induces a quasi-isomorphism $\varepsilon_* : \text{End}_A(B_\bullet(A, V)) \to \text{Hom}_A(B_n(A, V), \text{End}(V))$. We find that $\varepsilon_* \circ i = id$. Hence $i$ is a quasi-isomorphism. We can check that

$$
\begin{aligned}
&i(f) \circ i(g)(a \otimes a_1 \otimes \cdots \otimes a_N \otimes m) \\
&= a \otimes \ldots a_{N-n-m} \otimes f(a_{N-n-m+1} \otimes \cdots \otimes a_{N-m} \otimes g(a_{N-m+1} \otimes \ldots a_N \otimes m)) \\
&= i(f \cdot g)(a \otimes a_1 \otimes \cdots \otimes a_N \otimes m).
\end{aligned} \tag{6.104}
$$

Therefore $i$ is also a morphism of algebra sending the Hochschild cup product to the Yoneda product.

This is a very useful fact because we used Koszul resolution to compute the Ext and it will be more convenient to use the Yoneda product instead of the Hochschild cup product in the computation. We show that this product indeed corresponds to the graded commutative product of the bulk algebra.

$$
O_1 \bullet \quad \leadsto \quad \bullet (O_1 O_2) \qquad \Rightarrow \text{Ext}(V, V) \otimes \text{Ext}(V, V) \to \text{Ext}(V, V)
$$
$$
O_2 \bullet
$$

Figure 1: bulk product and Yoneda product

We consider the free chiral multiplet in Dirichlet boundary condition as an example. Recall that the Ext is computed by the Koszul resolution

$$\widetilde{M}_{vac} = \oint \mathcal{V}_\partial[D] \otimes \mathbb{C}[\{\lambda_n\}_{n \geq 0}] \approx \mathbb{C}\left[\{\psi_n\}_{n \in \mathbb{Z}}, \{\lambda_n\}_{n \geq 0}\right], \tag{6.105}$$

with differential $d\lambda_n = \psi_n$. The Yoneda product is the product on the endomorphism ring $\text{End}_{\oint \mathcal{V}_\partial[D]}(\widetilde{M}_{vac})$. Then we can identify the endomorphism ring $\text{End}_{\oint \mathcal{V}_\partial[D]}(\widetilde{M}_{vac})$ with the ring $\mathbb{C}[\{\psi_n\}_{n \in \mathbb{Z}}, \{\lambda_n, \frac{\partial}{\partial \lambda_n}\}_{n \geq 0}]$, by identifying $\lambda_n^*$ with $\frac{\partial}{\partial \lambda_n}$. This complex has cohomology the bulk algebra of free chiral $\mathbb{C}[\{\psi_n, \lambda_n^*\}_{n \geq 0}]$. We have injective $j : \mathbb{C}[\{\psi_n, \lambda_n^*\}_{n \geq 0}] \to \text{End}_{\oint \mathcal{V}_\partial[D]}(\widetilde{M}_{vac})$ sending $\psi_n \to \psi_n$ and $\lambda_n^* \to \frac{\partial}{\partial \lambda_n}$, and a projection $p : \text{End}_{\oint \mathcal{V}_\partial[D]}(\widetilde{M}_{vac}) \to \mathbb{C}[\{\psi_n, \lambda_n^*\}_{n \geq 0}]$ by sending all $\frac{\partial}{\partial \lambda_n}$ to zero. Then the Yoneda product on $\mathbb{C}[\{\psi_{-n-1}, \lambda_n^*\}_{n \geq 0}]$ can be defined by

$$ab = p(j(a)j(b)), \quad \text{for any } a, b \in \mathbb{C}\left[\{\psi_{-n-1}, \lambda_n^*\}_{n \geq 0}\right]. \tag{6.106}$$

It is easy to check that this is exactly the graded commutative product of the polynomial algebra $\mathbb{C}[\{\psi_{-n-1}, \lambda_n^*\}_{n \geq 0}]$ and induces the desired graded commutative vertex algebra structure on it.

For a chiral multiplet with an arbitrary superpotential, the agreement between the Yoneda product and the graded commutative product of the bulk algebra can be checked similarly. For $V$ the vacuum Verma module $M_k$ of the affine Lie algebra $\hat{\mathfrak{g}}$, a similar statement is proved in [44].

By its identification with the primary product on bulk algebra, the Yoneda product on the self-Ext should be graded commutative. However, this is not obvious a priori. In fact, the Yoneda product on $\mathrm{Ext}_A^\bullet(M, M)$ is not necessarily a commutative product in general. We note that there are extra structures on the category of vertex algebra module and $V$. The category of $V$ module has the structure of a modular tensor category and in particular a monoidal category. The vacuum module is the unit object in this category. We expect that these structures should be enough to guarantee the commutativity of the self-Ext.

In a more general setting, [75] defined a bracket $[-,-]_\mathcal{C}$ on $\mathrm{Ext}_\mathcal{C}^\bullet(\mathbb{1}_\mathcal{C}, \mathbb{1}_\mathcal{C})$ for a strong exact monoidal category $(\mathcal{C}, \otimes_\mathcal{C}, \mathbb{1}_\mathcal{C})$. When we take $\mathcal{C}$ to be the category of $A$ bimodule, $\mathrm{Ext}_\mathcal{C}^\bullet(\mathbb{1}_\mathcal{C}, \mathbb{1}_\mathcal{C}) = HH^\bullet(A, A)$ and the bracket $[-,-]_\mathcal{C}$ agrees with the Gerstenhaber bracket [33] on the Hochschild cohomology. We expect that this construction will lead to the 0-th order component of the $\lambda$ bracket of the Poisson vertex algebra. However, the construction in [75] does not provide us a very explicit formula for the bracket as in the definition of Gerstenhaber bracket 6.96.

For the full Poisson vertex algebra structure, a proper construction of the chain complex replacing Hochschild complex is needed. We conjecture that the chiral deformation complex constructed by D. Tamarkin in [37] is quasi-isomorphic to the self-Ext we are computing. At least for the commutative chiral algebra, the cohomology of the chiral deformation complex is computed in *loc. cit.* and agrees nicely with the bulk algebra of free chiral multiplets. Moreover, D. Tamarkin constructed a shifted Poisson vertex algebra structure (which is called a c-Gerstenhaber algebra structure in *loc. cit.*) on the cohomology of the deformation complex. For the commutative chiral algebra, the bracket computed in *loc. cit.* also agrees with the bracket computed in [39] directly from the bulk algebra of free chiral multiplets. We believe that this is not a coincidence. The shifted Poisson vertex algebra structure on the cohomology of chiral deformation complex should be the same as the algebraic structure of bulk local operators of the corresponding $3d$ theory.

### 6.4.2 A generalization of Deligne's conjecture

The Yoneda product is only one piece of a series of algebraic structures on the self-Ext. Higher multiplication maps can be defined on the self-Ext extending the Yoneda product and making it into an $A_\infty$ algebra. For a projective resolution $P^\bullet$ of $V$, we have a dga algebra $\mathrm{End}_A(P^\bullet)$ whose cohomology is the $\mathrm{Ext}_A^\bullet(V, V)$. As a result of the Homotopy Transfer Theorem for dga algebra [76] (see also [64] for an introduction), there exists an $A_\infty$ algebra structure $\{m_n\}_{n \geq 2}$ on the cohomology $\mathrm{Ext}_A^\bullet(V, V)$, such that there is an $A_\infty$ quasi-isomorphism between $\mathrm{Ext}_A^\bullet(V, V)$ and the dga algebra $\mathrm{End}_A(P^\bullet)$ and the $m_2$ coincides with the Yoneda product. There are various techniques to explicitly construct this $A_\infty$ algebra, for example using homological perturbation theory [66].

The appearance of higher structure here is not a surprise. Higher products exist in a general TQFT of cohomological type. In $2d$ we have the famous Deligne conjecture, which has been verified by several people ( for examples [34,35]). This conjecture states that the Gerstenhaber algebra structure on the Hochschild cohomology actually comes from the structure of Hochschild complex as an algebra over the chain little disc operad. From the TQFT perspective, Deligne's conjecture is very natural because the space of local operators of a $2d$ TQFT at

the chain level has an $E_2$-algebra structure.

The twisted theories studied in this paper can be considered as "holomorphic topological" theories of cohomological type. Higher structures exist at the chain level that contain much richer structures than OPE's in the cohomology. Though we don't have a clear picture of all these higher structures that are present in the bulk algebra, we can try to understand it in a hierarchical manner. For example, we can first understand the OPE's of local operators in the topological $\mathbb{R}$ direction, this structure should be characterized by an $A_\infty$ algebra. Then we study the topological line defects and their OPE structure along the holomorphic $\mathbb{C}$ direction. Alternatively, we can first study OPE's of local operators in the holomorphic direction and then OPE's of holomorphic surface defects along the topological direction.

As we have seen, the Yoneda product correspond to the OPE of bulk operators in the topological $\mathbb{R}$ direction. We believe that the $A_\infty$ extension of the Yoneda product is a piece of the whole structure of the local operator algebras that encode the operator product in the topological $\mathbb{R}$ direction. It has been shown in [12] that the $\lambda$ bracket of the bulk algebra encodes the leading term of a bulk to line defect OPE. There must exist a coherent series of higher order operations that encode all the higher order terms in the line defect OPE's in the holomorphic direction. We believe that these structures of higher order operations should also appear in the self-$R$Hom or the chiral deformation complex of [37]. More generally we expect that

$$\text{End}_{\mathcal{C}}(\mathbb{1}_{\mathcal{C}}, \mathbb{1}_{\mathcal{C}}) \tag{6.107}$$

for a chiral category $\mathcal{C}$ to have this higher analog of chiral algebra structure.

This higher analog of chiral algebra structure is still mysterious to us. However, we can get a variant of this conjecture by imposing some extra structure on the boundary vertex algebra, and the resulting structure on the self-Ext will be more familiar to us. Specifically, as is discussed in [12], when the boundary vertex algebra $V$ has a stress energy tensor (i.e. $V$ is a conformal vertex algebra), the bulk theory will be topological. In this case the bulk local algebra becomes an $E_3$ algebra, and we expect that the self-$R$Hom can be naturally equipped with an $E_3$ algebra structure.

# 7 Discussion & Conclusion

In this paper, we studied the bulk local operators of holomorphic twisted $3d$ $\mathcal{N} = 2$ theories from two different perspectives – the first being a direct bulk analysis and the second using the bulk-boundary relation that relate the self-Ext (or the derived center) of the boundary algebra and the bulk algebra.

From the first perspective, we constructed the perturbative bulk local operators in the most general situations, and the full non perturbative local operators for abelian gauge theories. The non perturbative operators are constructed using state-operator correspondence and geometric quantization. A important construction is the line bundles on phase space, whose characters reproduce one loop correction to the gauge charges of monopole operators. We also analyzed the implication of mirror duality, which predict an isomorphism of the local operator algebras for mirror dual pair. This is non trivial as the perturbative operators and monopole operators are exchanged under the duality. We examined the isomorphism for part of the local operators for some dual pairs. It will be an interesting problem to prove the isomorphism for the whole algebras in some examples.

From the second perspective, we computed the self-Ext of boundary vertex algebras in many examples. We analyzed different boundary conditions and the corresponding boundary algebras for chiral multiplets. The self-Ext computations in these cases all coincide with our direct bulk analysis. Along the way, we discussed an interesting phenomenon that complimen-

tary boundary conditions lead to Koszul dual boundary vertex algebras. Theories with gauge fields turns out to be more subtle, as the bulk-boundary relation can fail for certain cases. Nevertheless, we expect that the bulk-boundary relation to hold when the theory is a CFT. We also examined the bulk-boundary relation for SQED and $U(1)_{\frac{1}{2}}+$ chiral theory. A remarkable fact is that monopole operators can arise form the self-Ext of a perturbative boundary algebra without monopoles. This might provide us with an easier way to access the bulk monopole operators when the direct analysis is hard, especially for non abelian theories with chiral multiplets and superpotential.

In the end, we touched on the algebraic structures of bulk operators from the boundary. We believe that the best context to discuss this is via the chiral deformation complex defined by [37]. It will be important to relate the chiral deformation complex, together with the c-Gerstenhaber structure on it, with the self-Ext and the bulk algebraic structure.

# Acknowledgments

I would like to thank Kevin Costello, Tudor Dimofte, Davide Gaiotto, Justin Hilburn, Si Li, Wenjun Niu, Shanming Ruan, Jingxiang Wu and Gongwang Yan for illuminating discussion. I especially thank Kevin Costello for invaluable conversation and guidance on this work. Research at Perimeter Institute is supported in part by the Government of Canada through the Department of Innovation, Science and Economic Development Canada and by the Province of Ontario through the Ministry of Colleges and Universities.

# A  Classical BV-formalism in finite dimension

In this section, we explain aspects of BV formalism in finite dimension following [32] and introduce various derived schemes appearing as the target in the AKSZ formulation of our theories.

In the Lagrangian approach to physics, we start with a space of fields $V$ (a finite dimensional space in our discussion) and an action functional $S : V \to \mathbb{R}$. Classical physics concerns the solutions of equations of motion for this system. Namely, we are interested in the critical locus of $S$.

$$\mathrm{Crit}(S) = \{\phi \in V : dS(\phi) = 0\}. \tag{A1}$$

The critical locus can also be considered as the intersection of the $\mathrm{graph}(dS) \subset T^*V$ with the zero-section of the cotangent bundle of $V$. In other words, $\mathrm{Crit}(S) = \mathrm{graph}(dS) \times_{T^*V} V$. Functions on $\mathrm{Crit}(S)$ can be written as

$$\mathcal{O}(\mathrm{Crit}(S)) = \mathcal{O}(\mathrm{graph}(dS)) \otimes_{\mathcal{O}(T^*V)} \mathcal{O}(V). \tag{A2}$$

The "derived" philosophy tells us that the naive critical locus could be highly singular (e.g. in the case of not transverse intersection). A better choice is to replace it by its derived version. In particular, in the above situation we replace the tensor product by the derived tensor product:

$$\mathcal{O}(\mathrm{dCrit}(S)) = \mathcal{O}(\mathrm{graph}(dS)) \otimes^{\mathbb{L}}_{\mathcal{O}(T^*V)} \mathcal{O}(V). \tag{A3}$$

This could be taken as a definition of the derived critical locus $\mathrm{dCrit}(S)$. Namely, we define it as a dg scheme whose ring of function is given by the derived tensor product as above. Explicitly, we can take Koszul resolution of $\mathcal{O}(V)$ as a $\mathcal{O}(T^*V)$ module and realize $\mathcal{O}(\mathrm{dCrit}(S))$ as the following complex

$$(\Gamma(V, \wedge^\bullet TV), \vee dS) = (\mathcal{O}(T^*[1]V), \vee dS). \tag{A4}$$

The Polyvector fields $\Gamma(V, \wedge^\bullet TV)$ come equipped with a Poisson bracket of cohomological degree 1, called Schouten-Nijenhuis bracket. For $f, g \in \mathcal{O}(V)$ and $X, Y \in \Gamma(V, TV)$, we have

$$\{X, Y\} = [X, Y], \quad \{X, f\} = Xf, \quad \{f, g\} = 0. \tag{A5}$$

This bracket extends to whole $\Gamma(V, \wedge^\bullet TV)$ by Leibniz rule, and defined a shifted symplectic structure on $T^*[1]V$. Using this bracket, the differential $\vee dS$ on $\mathcal{O}(T^*[-1]V)$ is identified with $\{S, -\}$.

Explicitly, given a basis $\{x_i\}$ of $V^*$ and a basis $\{\theta^i\}$ of $\Gamma(V, TV)[1]$, we can write $\mathcal{O}(T^*[-1]) = \mathbb{R}[x_i, \theta^i]$. Then the differential $\vee dS$ can be written as

$$\vee dS = \sum_i \frac{\partial S}{\partial x_i} \frac{\partial}{\partial \theta^i}. \tag{A6}$$

An important class of field theories involves a gauge group $G$ acting on the space of field $V$. We want to make sense of the quotient space $V/G$ and the space of functions on $V/G$. To avoid discussion on derived stack we consider the quotient $V/\mathfrak{g}$ by the Lie algebra. Again, the naive quotient could be very badly behaved. It is always better to take the derived invariants for the action of $\mathfrak{g}$ on the algebra $\mathcal{O}(V)$ of functions on $V$. This is given by the Chevalley-Eilenberg complex

$$(C^\bullet(\mathfrak{g}, \mathcal{O}(V)), d_{\text{CE}}) = (\mathcal{O}(\mathfrak{g}[1] \oplus V), d_{\text{CE}}). \tag{A7}$$

Explicitly, for $\{t_a\}$ a basis of $\mathfrak{g}$ with respect to which we have structure constant $f_{ab}^c$. $\mathfrak{g}$ act on $V$ by vector field. We denote $X_a = X_{ia}(x)\frac{\partial}{\partial x_i}$ the vector field associated with $t_a$. Write $c^a$ the dual basis of $\mathfrak{g}^*$, then the Chevalley-Eilenberg complex is $\mathbb{R}[x_i, c^a]$ with the following differential

$$d_{\text{CE}} = \sum_a c^a X_a + \sum_{a,b,c} \frac{1}{2} c^a c^b f_{ab}^c \frac{\partial}{\partial c^a}. \tag{A8}$$

Having understood the derived version of $V/\mathfrak{g}$, we can combine our previous discussion to understand the critical locus of $S$ inside $V/\mathfrak{g}$. We model this space by

$$T^*[1](\mathfrak{g}[1] \oplus V) = \mathfrak{g}[1] \oplus V \oplus V^*[-1] \oplus \mathfrak{g}^*[-2]. \tag{A9}$$

$T^*[1](\mathfrak{g}[1] \oplus V)$ is naturally equipped with a shifted Poisson bracket $\{-.-\}$, namely the Schouten-Nijenhuis bracket on $\mathfrak{g}[1] \oplus V$. The function $S$ on $\mathfrak{g}[1] \oplus V$ pulls back to $T^*[1](\mathfrak{g}[1] \oplus V)$ via the natural projection, and we still denote it by $S$. The Chevalley-Eilenberg differential $d_{\text{CE}}$ can be regarded as a vector field on $\mathfrak{g}[1] \oplus V$ and induce a vector field on $T^*[1](\mathfrak{g}[1] \oplus V)$. There exists a function $h_{\text{CE}}$ such that its Hamiltonian vector field is $d_{\text{CE}}$. The differential on $T^*[-1](\mathfrak{g}[1] \oplus V)$ can be expressed as

$$d = \{S + h_{\text{CE}}, -\}. \tag{A10}$$

Explicitly, using our previous notation and writing $b_a$ the corresponding basis of $\mathfrak{g}[2]$, then

$$\mathcal{O}(T^*[-1](\mathfrak{g}[1] \oplus V)) = \mathbb{R}[x_i, \theta^i, c_a, b^a]. \tag{A11}$$

The Poisson bracket is defined through

$$\{x_i, \theta^j\} = \delta_i^j, \quad \{c_a, b^b\} = \delta_a^b. \tag{A12}$$

The function $S + h_{\text{CE}}$ can be written as

$$S + \sum c^a X_{ia} \theta^i + \sum \frac{1}{2} c^a c^b b_c f_{ab}^c. \tag{A13}$$



The differential can be written explicitly as

$$
\begin{aligned}
d = \sum c^a X_a &+ \frac{\partial S}{\partial x_i} \frac{\partial}{\partial \theta^i} + c^a \frac{\partial X_{ja}}{\partial x_i} \theta^j \frac{\partial}{\partial \theta^i} \\
&+ \frac{1}{2} c^a c^b f_{ab}^c \frac{\partial}{\partial c^a} + X_{ia} \theta^i \frac{\partial}{\partial b_a} + c^b b_c f_{ab}^c \frac{\partial}{\partial b^a} \, .
\end{aligned}
\tag{A14}
$$

The derived critical locus of $V/\mathfrak{g}$ can be equivalently described as a derived symplectic reduction

$$
\mathrm{dCrit}(S) /\!\!/ \mathfrak{g} \, .
\tag{A15}
$$

Symplectic reduction consists of two steps, first we take the zero sections of the momentum map and then we take the quotient. Derived symplectic reduction simply performs the above two steps in a derived way. First, the action of $\mathfrak{g}$ on the dg scheme induces a momentum map

$$
\mu : T^*[1]V \to \mathfrak{g}^*[-1] \, .
\tag{A16}
$$

Functions on the homotopy fibre $\mu^{-1}(0)$ is defined by

$$
\mathcal{O}(\mu^{-1}(0)) = \mathcal{O}(T^*[1]V) \otimes_{\mathcal{O}(\mathfrak{g}^*[-1])}^{\mathbb{L}} \mathbb{C} \, .
\tag{A17}
$$

By using Koszul resolution this can be modeled on a complex $\mathcal{O}(T^*[1]V \oplus \mathfrak{g}[-2])$. Explicitly, we use our previous notation of the basis, then this complex can be identified with $R[x_i, \theta^i, b^a]$. It has differential

$$
d = \frac{\partial S}{\partial x_i} \frac{\partial}{\partial \theta^i} + X_{ia} \theta^i \frac{\partial}{\partial b_a} \, .
\tag{A18}
$$

Next, we take the derived quotient of $T^*[1]V \oplus \mathfrak{g}[-2]$ by $\mathfrak{g}$. This is performed similarly to A7. We use Chevalley-Eilenberg complex

$$
C^\bullet(\mathfrak{g}, \mathcal{O}(T^*[1]V \oplus \mathfrak{g}[-2])) \approx \left( \mathbb{R}[x_i, \theta^i, c_a, b^a], d \right) \, .
\tag{A19}
$$

The differential here is exactly the same as A14.

# B  Berezinian and its properties

In this appendix, we introduce the definition of Berezinian following [77] and discuss its properties. Let $A$ be a (super) commutative algebra, $L$ a free module of rank $p|q$ over $A$. We have an isomorphism $L = \mathbb{C}^{p|q} \otimes A = A^{p|q}$. We will be most interested in the case $A = \mathbb{C}$.

**Definition B.1** *Consider the super algebra* $\mathrm{Sym}^\bullet(L^*)$. *We view $A$ as a* $\mathrm{Sym}^\bullet(L^*)$ *module by the augmentation map that projects* $\mathrm{Sym}^{\geq 1}(L^*)$ *to zero.*
   *The Berezinian of $L$ is*

$$
\mathrm{Ber}\, L = \mathrm{Ext}_{\mathrm{Sym}^\bullet(L^*)}^p(A, \mathrm{Sym}^\bullet(L^*)) \, .
\tag{B1}
$$

**Example B.1** *For $L = \mathbb{C}^{p|0}$, we consider the Koszul resolution of $\mathbb{C}$*

$$
\cdots \to \mathrm{Sym}^\bullet(L^*) \otimes \wedge^2 L^* \to \mathrm{Sym}^\bullet(L^*) \otimes L^* \to \mathrm{Sym}^\bullet(L^*) \to \mathbb{C} \, .
\tag{B2}
$$

$\mathrm{Ext}^\bullet$ *can be computed by*

$$
\mathrm{Hom}_{\mathrm{Sym}^\bullet(L^*)}(\mathrm{Sym}^\bullet(L^*) \otimes \wedge^\bullet L^*, \mathrm{Sym}^\bullet(L^*)) \approx \wedge^\bullet L \otimes \mathrm{Sym}^\bullet(L^*) \, ,
\tag{B3}
$$

*where the differential is given by*

$$d(e_{i_1} \wedge \cdots \wedge e_{i_k}) = \sum_{j=1}^{p} e_j^* \otimes e_j \wedge e_{i_1} \wedge \cdots \wedge e_{i_k}. \tag{B4}$$

*We have $H^\bullet(\wedge^\bullet L \otimes \text{Sym}^\bullet(L^*)) = H^p(\wedge^\bullet L \otimes \text{Sym}^\bullet(L^*)) = \wedge^p L$. Moreover,*

$$\text{Ber} \, L = \wedge^p L, \tag{B5}$$

*this gives us the usual definition of the determinant for an ordinary free module.*

*Let $T \in \text{End}_{\mathbb{C}}(L)$ be an endomorphism of $L = \mathbb{C}^p$. The induced action of $T$ on $\text{Ber} \, L = \det L$ is given by multiplication of $\det T$, the determinant of the operator $T$.*

**Example B.2** *For $L = \mathbb{C}^{0|q}$, if we denote a basis of $L^*$ by $\theta_1, \ldots, \theta_q$, then we have $\text{Sym}^\bullet(L^*) = \mathbb{C}[\theta_1, \ldots, \theta_q]$. Note that $\mathbb{C}[\theta_1, \ldots, \theta_q]$ is an injective module over itself. Therefore $\text{Ext}^\bullet = \text{Ext}^0 = \text{Hom}$. And*

$$\text{Ber} \, L = \text{Hom}_{\mathbb{C}[\theta_1,\ldots,\theta_q]}\left(\mathbb{C}, \mathbb{C}[\theta_1, \ldots, \theta_q]\right). \tag{B6}$$

*$\text{Ber} \, L$ is spanned by a basis $e \in \text{Hom}_{\mathbb{C}[\theta_1,\ldots,\theta_q]}(\mathbb{C}, \mathbb{C}[\theta_1, \ldots, \theta_q])$ defined by*

$$e(a) = \theta_1 \theta_2 \cdots \theta_q a, \quad \text{for } a \in \mathbb{C}. \tag{B7}$$

*Moreover, we see from the above formula B7 that the $\mathbb{Z}_2$ grading of $\text{Ber} \, L$ is even if $q$ is even, and odd if $q$ is odd.*

*Let $T$ be an endomorphism of $L = \mathbb{C}^{0|q}$. Suppose the corresponding matrix with respect to the basis is $N$. The action on $L$ induces an action on the dual $L^*$ and hence an action on $\text{Sym}^\bullet(L^*)$ also denoted by $T$. It maps $\theta_1 \theta_2 \cdots \theta_q$ to $(\det N)\theta_1 \theta_2 \cdots \theta_q$. It also induce an action on $\text{Hom}_{\mathbb{C}[\theta_1,\ldots,\theta_q]}(\mathbb{C}, \mathbb{C}[\theta_1, \ldots, \theta_q])$ defined as follows. For any $f \in \text{Hom}_{\mathbb{C}[\theta_1,\ldots,\theta_q]}(\mathbb{C}, \mathbb{C}[\theta_1, \ldots, \theta_q])$, $T f$ is defined such that*

$$
\begin{array}{ccc}
\mathbb{C} & \xrightarrow{\ f\ } & \text{Sym}^\bullet(L^*) \\
{\scriptstyle 1}\big\uparrow & & \big\uparrow{\scriptstyle T} \\
\mathbb{C} & \xrightarrow{\ Tf\ } & \text{Sym}^\bullet(L^*)
\end{array}
\tag{B8}
$$

*We see that $T f$ is defined if and only if $T \in \text{End}_{\mathbb{C}}(L)$ is invertible, and $Te = (\det N)^{-1} e$. This means that*

$$\text{Ber} \, T = (\det N)^{-1}. \tag{B9}$$

More generally we can compute the $\text{Ext}^\bullet_{\text{Sym}^\bullet(L^*)}(A, \text{Sym}^\bullet(L^*))$ for any free module $L$ using the standard Koszul resolution. We have the following results

**Proposition B.1** *Let $L$ be a free module of rank $p|q$ over $\mathbb{C}$. Then we have*

$$
\text{Ext}^n_{\text{Sym}^\bullet(L^*)}(\mathbb{C}, \text{Sym}^\bullet(L^*)) = 
\begin{cases}
\mathbb{C}^{1|0}, & \text{if } n = p \text{ and } q \text{ is even}, \\
\mathbb{C}^{0|1}, & \text{if } n = p \text{ and } q \text{ is odd}, \\
0, & \text{if } n \neq p.
\end{cases}
\tag{B10}
$$

Note that the Koszul resolution for $A$ as a $\text{Sym}^\bullet((L_1 \oplus L_2)^*)$ module is exactly the tensor product of the Koszul resolutions of $A$ as $\text{Sym}^\bullet(L_1^*)$ and $\text{Sym}^\bullet(L_2^*)$ modules respectively. Combining with the Künneth theorem we have

**Corollary B.1** *Let $L_1$, $L_2$ be two free modules over $\mathbb{C}$. we have*

$$\mathrm{Ber}(L_1 \oplus L_2) = \mathrm{Ber}(L_1) \otimes \mathrm{Ber}(L_2). \tag{B11}$$

**Corollary B.2** *Let $T \in$ be an invertible endomorphism of $\mathbb{C}^{p|q}$ with matrix $\begin{pmatrix} K & L \\ M & N \end{pmatrix}$. Then the induced action of $T$ on $\mathbb{C}^{p|q}$ is given by multiplication by the Berezinian defined as follows*

$$\mathrm{Ber}\, T = \det(K - LN^{-1}M)\det(N)^{-1}. \tag{B12}$$

## C Computing some complexes

In this appendix we simplify the complexes appearing in Section 6.1.3 and in Section 6.1.5. First we consider the simpler case, the XYZ model. We recall that the complex we wish to compute is the following

$$\mathbb{C}\Big[\{X_n, Y_n, Z_n, \Gamma_n, \widetilde{\Gamma}_n\}_{n<0}, \{\eta^*_{X,n}, \eta^*_{Y,n}, \eta^*_{Z,n}, \sigma^*_n, \widetilde{\sigma}^*_n\}_{n\geq 0}\Big], \tag{C1}$$

with differential

$$
\begin{aligned}
d = & \sum_{n<0} \widetilde{\sigma}^*_{-n-1}\partial_{\Gamma_n} + \sigma^*_{-n-1}\partial_{\widetilde{\Gamma}_n} + \sum_{n<0} X_n \partial_{\Gamma_n} + \sum_{n,m<0} Y_n Z_m \partial_{\widetilde{\Gamma}_{n+m+1}} \\
& - \sum_{n\geq 0}\left(\sigma^*_n \partial_{\eta^*_{X,n}} + \sum_{0\leq m \leq n} \widetilde{\sigma}^*_m Z_{m-n-1}\partial_{\eta^*_{Y,n}} + \widetilde{\sigma}^*_m Y_{m-n-1}\partial_{\eta^*_{Z,n}}\right).
\end{aligned}
\tag{C2}
$$

By giving $\Gamma_n, \widetilde{\Gamma}_n$ bidegree $(1,0)$ and $\eta^*_{X,n}, \eta^*_{Y,n}, \eta^*_{Z,n}$ bidegree $(0,1)$ and all other elements bidegree $(0,0)$. The complex C1 becomes a double complex $C_{p,q}$ with two differential $d_1 : C_{p,q} \to C_{p-1,q}$ and $d_2 : C_{p,q} \to C_{p,q-1}$ given by

$$d_1 = \sum_{n<0}(\widetilde{\sigma}^*_{-n-1} + X_n)\partial_{\Gamma_n} + \sum_{n<0}\left(\sigma^*_{-n-1} + \sum_{\substack{m,l<0 \\ m+l=n-1}} Y_m Z_l\right)\partial_{\widetilde{\Gamma}_{n+m+1}}, \tag{C3}$$

and

$$d_2 = -\sum_{n\geq 0}\left(\sigma^*_n \partial_{\eta^*_{X,n}} + \sum_{0\leq m \leq n} \widetilde{\sigma}^*_m Z_{m-n-1}\partial_{\eta^*_{Y,n}} + \widetilde{\sigma}^*_m Y_{m-n-1}\partial_{\eta^*_{Z,n}}\right). \tag{C4}$$

We note that $(C_{\bullet,\bullet}, d_1)$ is the standard Koszul resolution with respect to the sequence of elements $\{r_{2n} := \widetilde{\sigma}^*_n + X_{-n-1}\}_{n\geq 0}$ and $\{r_{2n+1} := \sigma^*_n + \sum_{\substack{m,l<0 \\ m+l=-n}} Y_m Z_l\}_{n\geq 0}$. Moreover, it is easy to check that this sequence $\{r_n\}_{n\geq 0}$ is a regular sequence, therefore the cohomology only survives at degree 0. We have

$$H_p\big(C_{\bullet q}, d_1\big) = 0, \quad \text{for } p > 0, \tag{C5}$$

and

$$E^1_{0\bullet} = H_0(C_{\bullet,\bullet}, d_1) = \mathbb{C}\Big[\{X_n, Y_n, Z_n\}_{n<0}, \{\eta^*_{X,n}, \eta^*_{Y,n}, \eta^*_{Z,n}, \sigma^*_n, \widetilde{\sigma}^*_n\}_{n\geq 0}\Big]/(r_1, r_2, \dots). \tag{C6}$$

Note that we have an isomorphism

$$\mathbb{C}\Big[\{X_n, Y_n, Z_n\}_{n<0}, \{\eta^*_{X,n}, \eta^*_{Y,n}, \eta^*_{Z,n}\}_{n\geq 0}\Big] \xrightarrow{\approx} E^1_{0\bullet}. \tag{C7}$$

Under this isomorphism, the differential $d_2$ becomes

$$d_2 = -\sum_{n\geq 0}\left(\sum_{\substack{m,l<0\\m+l=-n}} Y_m Z_l \partial_{\eta^*_{X,n}} + \sum_{0\leq m\leq n} X_{-n-1}Z_{m-n-1}\partial_{\eta^*_{Y,n}} + X_{-n-1}Y_{m-n-1}\partial_{\eta^*_{Z,n}}\right). \qquad \text{(C8)}$$

This is exactly the complex of 4.1.

Now we consider the complex in Section 6.1.5.

$$\left(\mathbb{C}\left[\{\phi_{i,-n-1},\Gamma^\alpha_{-n-1},\widetilde{\Gamma}_{\alpha,-n-1},\eta^*_{i,n},\sigma^{\alpha*}_n,\widetilde{\sigma}^*_{\alpha,n}\}_{n\geq 0}\right],d\right), \qquad \text{(C9)}$$

where the differential is given by

$$\begin{aligned}
d = &\sum_{n<0,\alpha}\widetilde{\sigma}^*_{\alpha,-n-1}\partial_{\Gamma^\alpha_n} + \sigma^\alpha_{-n-1}\partial_{\widetilde{\Gamma}_{\alpha,n}} + E^\alpha_n\Big|_{\substack{\phi_{j,k}=0\\\text{for }k\geq 0}}\partial_{\Gamma^\alpha_n} + J_{\alpha,n}\Big|_{\substack{\phi_{j,k}=0\\\text{for }k\geq 0}}\partial_{\widetilde{\Gamma}_{\alpha,n}}\\
&+ \sum_{n<0,m\geq 0,\alpha}\left(\sigma^{\alpha*}_m\langle\eta^*_{i,n},h(E^\alpha_m)\rangle + \widetilde{\sigma}^*_{\alpha,m}\langle\eta^*_{i,n},h(J_{\alpha,m})\rangle\right)\Big|_{\substack{\phi_{j,k}=0\\\text{for }k\geq 0}}\partial_{\eta^*_{i,n}}.
\end{aligned} \qquad \text{(C10)}$$

As before we give $\{\Gamma^\alpha_{-n-1},\widetilde{\Gamma}_{\alpha,-n-1}\}_{n\geq 0}$ bidegree $(1,0)$ and $\{\eta^*_{i,n}\}_{n\geq 0}$ bidegree $(0,1)$. Then we get a double complex with differential

$$d_1 = \sum_{n<0,\alpha}\widetilde{\sigma}^*_{\alpha,-n-1}\partial_{\Gamma^\alpha_n} + \sigma^\alpha_{-n-1}\partial_{\widetilde{\Gamma}_{\alpha,n}} + E^\alpha_n\Big|_{\substack{\phi_{j,k}=0\\\text{for }k\geq 0}}\partial_{\Gamma^\alpha_n} + J_{\alpha,n}\Big|_{\substack{\phi_{j,k}=0\\\text{for }k\geq 0}}\partial_{\widetilde{\Gamma}_{\alpha,n}}, \qquad \text{(C11)}$$

and

$$d_2 = \sum_{n\geq 0,m\geq 0,\alpha}\left(\sigma^{\alpha*}_m\langle\eta^*_{i,n},h(E^\alpha_m)\rangle + \widetilde{\sigma}^*_{\alpha,m}\langle\eta^*_{i,n},h(J_{\alpha,m})\rangle\right)\Big|_{\substack{\phi_{j,k}=0\\\text{for }k\geq 0}}\partial_{\eta^*_{i,n}}. \qquad \text{(C12)}$$

The $E^0$ page complex is the standard Koszul complex with respect to the regular sequence $r^\alpha_{2n} := \widetilde{\sigma}^*_{\alpha,n} + E^\alpha_{-n-1}|_{\substack{\phi_{j,k}=0\\\text{for }k\geq 0}}$ and $r^\alpha_{2n+1} := \sigma^\alpha_n + J_{\alpha,-n-1}|_{\substack{\phi_{j,k}=0\\\text{for }k\geq 0}}$. We have

$$E^1_{0\bullet} = \mathbb{C}[\{\phi_{i,-n-1},\eta^*_{i,n},\sigma^{\alpha*}_n,\widetilde{\sigma}^*_{\alpha,n}\}_{n\geq 0}]/(r^\alpha_n). \qquad \text{(C13)}$$

Moreover, there is an isomorphism

$$\mathbb{C}[\{\phi_{i,-n-1},\eta^*_{i,n}\}_{n\geq 0}]\xrightarrow{\approx} E^1_{0\bullet}, \qquad \text{(C14)}$$

under which the differential $d_2$ becomes

$$d = \sum_{n\geq 0,m\geq 0,\alpha}\left(J_{\alpha,-m-1}\langle\eta^*_{i,n},h(E^\alpha_m)\rangle + E^\alpha_{-m-1}\langle\eta^*_{i,n},h(J_{\alpha,m})\rangle\right)\Big|_{\substack{\phi_{j,k}=0\\\text{for }k>0}}\partial_{\eta^*_{i,n}}. \qquad \text{(C15)}$$

Now, we further simplify this formula. We recall that for any polynomial $F\in\mathbb{C}[\{\phi_i\}_{i=1\ldots,N_f}]$ we defined a polynomial (see 6.1.5) $(F)_n\in\mathbb{C}[\{\phi_i\}_{i=1\ldots,N_f,n\in\mathbb{Z}}]$, which is defined on monomial by

$$(\phi_{i_1}\phi_{i_2}\ldots\phi_{i_l})_n = \sum_{\substack{n_i\in\mathbb{Z}\\n_1+n_2+\cdots+n_l=n+1-l}}\phi_{i_1,n_1}\phi_{i_2,n_2}\cdots\phi_{i_l,n_l}, \qquad \text{(C16)}$$

and extend linearly to $\mathbb{C}[\{\phi_i\}_{i=1\ldots,N_f}]$. With this definition we immediately find that for any $k$ that $1\leq k\leq l$

$$\begin{aligned}
(\phi_{i_1}\ldots\phi_{i_l})_n &= \sum_{m\in\mathbb{Z}}\sum_{\substack{n_i\in\mathbb{Z}\\n_1+n_2+\cdots+n_k=n-m-k}}\phi_{i_1,n_1}\cdots\phi_{i_k,n_k}\sum_{\substack{n_i\in\mathbb{Z}\\n_{k+1}+n_2+\cdots+n_l=m+k+1-l}}\phi_{i_{k+1},n_{k+1}}\cdots\phi_{i_l,n_l}\\
&= \sum_{m\in\mathbb{Z}}(\phi_{i_1}\ldots\phi_{i_k})_{n-m-1}(\phi_{k+1}\ldots\phi_l)_m.
\end{aligned} \qquad \text{(C17)}$$

By linearity we have $(FG)_n = \sum_{m \in \mathbb{Z}} (F)_{n-m-1} (G)_m$ for any polynomial $F$ and $G$.

Since $W = E^\alpha J_\alpha$, we have $\frac{\delta W}{\delta \phi_i} = \frac{\delta E^\alpha}{\delta \phi_i} J_\alpha + E^\alpha \frac{\delta J_\alpha}{\delta \phi_i}$ and

$$\left( \frac{\delta W}{\delta \phi_i} \right)_{-n-1} = \sum_{m \in \mathbb{Z}} \left( \frac{\delta E^\alpha}{\delta \phi_i} \right)_{m-n-1} (J_\alpha)_{-m-1} + \left( \frac{\delta J_\alpha}{\delta \phi_i} \right)_{m-n-1} (E^\alpha)_{-m-1}. \tag{C18}$$

When we restrict these polynomial by letting $\phi_{j,k} = 0$ for $k \geq 0$ we find that

$$\left( \frac{\delta W}{\delta \phi_i} \right)_{-n-1} \Big|_{\substack{\phi_{j,k}=0 \\ \text{for } k>0}} = \sum_{0 \leq m \leq n} \left( \left( \frac{\delta E^\alpha}{\delta \phi_i} \right)_{m-n-1} (J_\alpha)_{-m-1} + \left( \frac{\delta J_\alpha}{\delta \phi_i} \right)_{m-n-1} (E^\alpha)_{-m-1} \right) \Big|_{\substack{\phi_{j,k}=0 \\ \text{for } k>0}}. \tag{C19}$$

Then we consider the expression $\left( J_{\alpha,-m-1} \langle \eta^*_{i,n}, h(E^\alpha_m) \rangle + E^\alpha_{-m-1} \langle \eta^*_{i,n}, h(J_{\alpha,m}) \rangle \right) \Big|_{\substack{\phi_{j,k}=0 \\ \text{for } k>0}}$. We prove that for any $F \in \mathbb{C}[\{\phi_i\}_{i=1\ldots,N_f}]$ we have

$$\left\langle \eta^*_{i,n}, h((F)_m) \right\rangle \Big|_{\substack{\phi_{j,k}=0 \\ \text{for } k>0}} = \left( \frac{\delta F}{\delta \phi_i} \right)_{m-n-1} \Big|_{\substack{\phi_{j,k}=0 \\ \text{for } k>0}}. \tag{C20}$$

We only need to prove this for monomial

$$\begin{aligned}
&\left\langle \eta^*_{i,n}, h((\phi_{i_1} \ldots \phi_{i_l})_m) \right\rangle \Big|_{\substack{\phi_{j,k}=0 \\ \text{for } k>0}} \\
&= \sum_{\substack{n_i \in \mathbb{Z} \\ n_1+n_2+\cdots+n_l=m+1-l}} \frac{1}{\sum_{\substack{j=1,\ldots,l \\ \text{with } n_j \geq 0}} 1} \sum_{\substack{j=1,\ldots,l \\ \text{with } n_j \geq 0}} \delta_{i,i_j} \delta_{n,n_j} \phi_{i_1,n_1} \cdots \hat{\phi}_{i_j,n_j} \cdots \phi_{i_l,n_l} \Big|_{\substack{\phi_{j,k}=0 \\ \text{for } k>0}} \\
&= \sum_{\substack{j=1,\ldots,l}} \sum_{\substack{n_i \leq 0 \\ n_1+\cdots+\hat{n}_j+\cdots+n_l=m-n+1-l}} \delta_{i,i_j} \phi_{i_1,n_1} \cdots \hat{\phi}_{i_j,n_j} \cdots \phi_{i_l,n_l} \\
&= \left( \frac{\delta(\phi_{i_1} \ldots \phi_{i_l})}{\delta \phi_i} \right)_{m-n-1} \Big|_{\substack{\phi_{j,k}=0 \\ \text{for } k>0}}.
\end{aligned} \tag{C21}$$

Using this we find that

$$\begin{aligned}
d_2 &= \sum_{n \geq 0, 0 \leq m \leq n} \left( (J_\alpha)_{-m-1} \left( \frac{\delta E^\alpha}{\delta \phi_i} \right)_{m-n-1} + (E^\alpha)_{-m-1} \left( \frac{\delta J_\alpha}{\delta \phi_i} \right)_{m-n-1} \right) \Big|_{\substack{\phi_{j,k}=0 \\ \text{for } k>0}} \partial_{\eta^*_{i,n}} \\
&= \left( \frac{\delta W}{\delta \phi_i} \right)_{-n-1} \Big|_{\substack{\phi_{j,k}=0 \\ \text{for } k>0}} \partial_{\eta^*_{i,n}}.
\end{aligned} \tag{C22}$$

This is exactly the complex computing the bulk algebra of chiral mutiplets with an arbitrary superpotential $W$.

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
