# Peer review of "Monopole Operators and Bulk-Boundary Relation in Holomorphic Topological Theories"

_SciPost Physics, doi:SciPost Phys. 14, 153 (2023)_

## Round 2 · Referee Report · Anonymous (Referee 1) · 2022-10-16

Strengths

  1. This paper rigorously generalizes 3d N=4 analysis of bulk-boundary algebra correspondence to 3d N=2. It is nice to see quite explicit self-Ext computation here, which was not easily found in other literature.
  2. At the same time, it attempts to study failure case of the above relation in the case of 3d N=2.

Weaknesses

  1. This paper is obviously written by a mathematician, who has a great interest in physics. Of course, this itself is not a weakness, but it seems that there is a huge need for more friendly explanation if the author cares physicist readers.
  2. I see numerous grammatical errors, such as singular vs plural.

Report

I have no doubt that the paper presents a very nice and correct result, but since this journal is SciPost *Physics*, the author may need to make an effort when presenting the material. I recommend to publish the paper if parts that need further (physicist-friendly) explanation can be supplemented.

Requested changes

The following list collects some(not all) parts that may need some further explanation for general physicist readers.

  1. Equality signs in (2.15), (2.17).
  2. A sentence after (2.18).
  3. In many places, I find an expression "derived". It would be nice to have some more friendly explanation.
  4. It was hard for me to read the paragraph that contains equation (3.11).
  5. Some words that are used by the author frequently may need further explanation, such as "... infinite jet of the ”non derived”..." in front of (4.4).
  6. It was very hard for me to read section 6.4.

  • validity: high
  • significance: good
  • originality: good
  • clarity: ok
  • formatting: good
  • grammar: acceptable

Author:  Keyou Zeng  on 2022-11-10  [id 3001]

(in reply to Report 1 on 2022-10-16)
Category:
remark
answer to question
correction

I would like to thank the referee for his/her useful suggesttions.

  1. I added a few more explanations around (2.15), (2.17), (2.18) .

  2. I added a paragraph in the middle of p.11 (after (3.4)) to explain the word "derived" and address its importance.

  3. I added a few more explanations after (3.11).

  4. I explianed the word "infinite jet" in the paragraphy after (3.3). I deleted the word "non derived" becasue it is not a standard terminology. "infinite jet of the critical locus of W" is defined by (4.4) and (4.5). So I didn't add any further explaination.

  5. I added a few more physical explanations in section 6.4. However, the purpose of this section is to define the bulk algebraic structures (e.g. OPE) abstractly from the boundary algebra. The construction is purely mathematical, not direclty related to QFT style computations or any physical intuitions. So I couldn't make it more physicist-friendly. However, the final answer should coincide with direct computation in QFT. The reader can see the reference [38,39] cited in that section for the QFT conterpart of the mathemtical construction.

---

## Round 2 · Referee Report · Anonymous (Referee 2) · 2022-10-24

Strengths

see report

Weaknesses

see report

Report

This work is centered on the study of 3d N=2 theories and in particular the holomorphic-topologic twist they admit. The focus is given to the structure of the algebra of local operators: in the twisted theory the operators form a chiral algebra which in the paper is explicitly constructed in specific examples of 3d N=2 theories, both in the perturbative and non-perturbative level, focusing on the latter to monopole operators. This analysis is rigorous and is performed in the Batlin-Vilkovisky (BV) formalism, via state-operator correspondence combined with geometric quantisation. Applying the construction initially for the cases of basic building blocks, such as 3d N=2 chiral and vector multiplets (abelian theories are treated), the analysis is performed for specific 3d N=2 theories and mirror dual pairs.
The work extends to theories with boundary conditions preserving 2d $\mathcal{N}=(0,2)$ supersymmetry, where the boundary algebras for specific boundary conditions are discussed. A nice feature is the addition of an analysis on the role of Koszul duality relating boundary conditions.
The core of this analysis is the correspondence between boundary and bulk operator algebras, which is studied through a rigorous self-Ext computation, along with a case where this construction fails to give such a bulk-boundary algebra correspondence. The article closes with a discussion on the higher algebraic structure of the bulk local operator algebra, accessible from the above self-Ext computation.

The article is overall well written, consistently structured and presents a very important set of results, extending recent analyses on this field.

The main idea, the motivation and the general structure of the paper, along with a consistent presentation of the background material, are given with clarity in a very well written introduction. The introduction is followed by a list of open research directions related to the presented analysis (a part of which also motivated this work), as well as by section reviewing the holomorphic twist of the 3d N=2 theories. The main part of the paper is organised in sections including the computations and the supplemented by an extensive appendix, which includes the basic notions of the classical BV formalism in terms of which the analysis of the 3d N=2 theories is performed, the definition and the properties of the Berezinian of a module of a super-commutative algebra and additional explicit computations, supplementing the final section of the paper.
Nevertheless, the paper lacks a discussion session summarizing the results and also a control for minor grammatical errors throughout the paper (e.g various errors of the type "Acknowledgement vs. Acknowledgements"; minor but still important) is needed.

I would highly recommend this paper for publication in SciPost Physics after:

-an addition of a final "Discussion/Conclusions" section, including a clear summary of the main results of the author's work and underlying their importance, as it would be highly useful for the readers and also would add to the clarity and the (already very good) consistency of the paper.

Requested changes

  1. addition of a final "Discussion/Conclusions" section, including a clear summary of the main results of the author's work and underlying their importance, as it would be highly useful for the readers and also would add to the clarity and the (already very good) consistency of the paper.

---

## Round 3 · Referee Report · Anonymous (Referee 2) · 2022-11-23

Report

The author has provided a new , updated version of this work, in which has applied the recommended modifications/additions stated in the previous referee report (indicatively the addition of the colclusions and discussion section, in which the author provides a clear summary of the analysis and results obtained).

I therefore strongly recommend this paper for publication in SciPost Physics.

---

## Round 3 · Referee Report · Anonymous (Referee 1) · 2023-2-14

Report

The author successfully supplemented requested changes. Therefore, I recommend to publish the article.

---

## Editorial Decision

published